# Diffusion Posterior Sampling for Nonlinear Contextual Bandits

## Abstract

We study multi-task nonlinear contextual bandits, where different tasks share the same reward structure but are characterized by distinct model parameters drawn from a common unknown prior distribution. The goal is to leverage information from past tasks to minimize regret on a new task with limited online interactions. Thompson Sampling (TS) is a popular approach for solving contextual bandits, maintaining a posterior over the model parameter that is updated each round using a hand-specified conjugate prior (e.g., Gaussian) and the observed rewards. However, such priors cannot capture the rich cross-task structure in multi-task settings, leading to misspecified posteriors and suboptimal exploration. To address this, we train a diffusion model on data from past tasks to learn a flexible prior distribution over task parameters. In a new bandit task, parameters are estimated via a conditional reverse-diffusion process, where each step combines: (i) an unconditional drift from the diffusion prior, (ii) a likelihood-driven drift from the interaction history, and (iii) a noise term enabling randomized exploration. We instantiate this framework in two ways. **DLTS** integrates history into the diffusion prior at every reverse step to form a conditional posterior, from which approximate samples are drawn. **DPSG** first performs unconditional reverse sampling from the pretrained diffusion prior and then applies a single history-guided gradient correction. Both methods adhere to the same framework but differ in how they incorporate interaction history from the new task: DLTS explicitly constructs the conditional posterior, while DPSG provides a lightweight approximation by coupling unconditional sampling with one corrective step. In theory, we formalize oracle TS (OTS) and its diffusion counterpart (ODTS) and prove they are equivalent when the diffusion prior matches the true prior. We bound the per-round expected regret gap between ODTS and OTS by the cumulative score estimation error across diffusion levels. Our empirical evaluation demonstrates that our proposed methods are competitive with specialized baselines in linear settings and outperform baselines benefiting from the diffusion prior in challenging nonlinear bandit environments.

## 1 Introduction

Sequential decision-making under uncertainty hinges on balancing exploration and exploitation (Lattimore & Szepesvári, 2020). One prominent approach to address this trade-off is Thompson Sampling (TS) (Thompson, 1933). In contextual bandits, TS maintains a posterior over the model parameter that is updated each round using a hand-specified conjugate prior (e.g., Gaussian) and the observed rewards (Agrawal & Goyal, 2013a). In multi-task settings where different tasks share the same reward structure but are characterized by distinct model parameters drawn from a common unknown prior distribution, such simple priors fail to capture rich cross-task structure, leading to misspecified posteriors and suboptimal exploration (Chapelle & Li, 2011). To address this, Hong et al. (2022) proposed to initialize TS with a mixture prior for multi-task bandits. Nevertheless, fixed parametric mixtures still struggle with complex structure—e.g., multimodal task families where the parameter distribution has many distinct peaks (modes) that can be uneven in size or far apart (Vuorio et al., 2019; Finn et al., 2018), and heavy-tailed families with extreme outliers (Bubeck et al., 2013; Forbes & Wraith, 2014). Such patterns are hard to capture with a small, fixed set of components, leading to underfit posteriors. This motivates learning a flexible prior that fully leverages past tasks to enable fast adaptation on a new task with limited interaction.

Deep generative models have achieved remarkable success in producing high-quality synthetic data across modalities (Saharia et al., 2022; Rombach et al., 2022; Liu et al., 2023). These results highlight their ability to model complex, multi-modal distributions. In online decision-making, the policy must incorporate newly collected interaction data and update its strategy frequently. Diffusion models (Sohl-Dickstein et al., 2015; Ho et al., 2020) align well with this need: their reverse process is iterative and naturally supports conditioning on new observations at every round. In score-based diffusion (Song et al., 2021), this conditioning is implemented by augmenting the score with a likelihood term (Chung et al., 2023). Given this intuition, prior work (Hsieh et al., 2023) tackles multi-task multi-armed bandits by training a diffusion model to learn a flexible prior over task parameters and coupling it with Thompson Sampling, with practical steps such as variance calibration to obtain reliable uncertainty in the reverse process (Hsieh et al., 2023). In parallel, diffusion priors have been adapted to online posterior sampling for linear and generalized linear contextual bandits via closed-form updates at each reverse step (Kveton et al., 2024). However, these prior work focus on specialized updates for linear or generalized linear bandits, leaving multi-task nonlinear contextual bandits largely unexplored. The core difficulty is the absence of closed-form diffusion reverse updates for nonlinear reward models, which complicates algorithm design. Without closed forms, sampling error can be large; combined with limited interactions per task and potential inaccuracies in the diffusion learned prior, this can destabilize posterior sampling and misguide exploration. Consequently, a natural question arises:

*Can we design diffusion posterior sampling algorithms for general nonlinear contextual bandits?*

In this paper, we provide an affirmative answer to this question by first providing a diffusion-based posterior sampling framework for multi-task nonlinear contextual bandits. We learn a flexible prior over task parameters with a diffusion model trained on past tasks, and on a new task perform conditional reverse sampling that blends prior drift, likelihood guidance, and stochastic exploration at each step. We instantiate this with two methods: DLTS, which conditions at every reverse step and samples via Langevin Monte Carlo (LMC), and DPSG, which draws an unconditional sample and applies a single likelihood-guided correction. To stabilize learning, we also propose DPSG-MP, a practical DPSG variant that replaces the single correction with a short inner loop of likelihood-gradient updates with projection.

**Our contributions** are summarized as follows:

- **Unified Framework.** We present a unified diffusion-based posterior sampling framework for multi-task nonlinear contextual bandits. In our framework, the algorithm estimates the reward model parameters via a conditional reverse-diffusion process that, at each step, consists of (i) an unconditional drift from the diffusion prior, (ii) a likelihood-driven drift from the interaction history, and (iii) a noise term enabling randomized exploration.
- **Algorithm Design.** We instantiate the unified framework in two ways: (i) Diffusion Langevin Thompson Sampling (DLTS), which explicitly constructs the conditional posterior and then draws approximate samples; and (ii) Diffusion Posterior Sampling with Guidance (DPSG), which couples unconditional sampling with a single history-guided correction step for simplicity and speed. We also propose DPSG with Multi-step Projection (DPSG-MP), a practical DPSG variant to stabilize learning, which improves empirical performance.
- **Theoretical analysis.** We formalize oracle TS (OTS), oracle diffusion TS (ODTS) and prove they are equivalent when the diffusion prior exactly matches the true prior. We also bound the per-round expected regret gap between ODTS and OTS by the cumulative score-estimation error across diffusion levels.
- **Extensive experiments.** We evaluate our algorithms in simulations of both linear and non-linear contextual bandits. In the linear setting, our algorithms achieve performance comparable to baseline algorithms that use closed-form updates, such as LinTS and DiffTS. Furthermore, our methods demonstrate strong performance in the non-linear bandit setting, where the learned diffusion prior provides a significant advantage.

## 2 PRELIMINARIES

**Nonlinear Contextual Bandits** Contextual bandits form a broad class of sequential decision problems where the player chooses based on an observed action set represented by feature vectors (contexts). At each round $t$, the player observes arm set $\mathcal{X}_t \subseteq \mathbb{R}^d$, selects an arm $x_t \in \mathcal{X}_t$ and receives

reward $y_t$ from the environment. Assume that the mean reward for a feature $\boldsymbol{x} \in \mathbb{R}^d$ is generated by an underlying function $f(\boldsymbol{\theta}^*; \boldsymbol{x})$, and the observed reward satisfies $y(\boldsymbol{x}) = f(\boldsymbol{\theta}^*; \boldsymbol{x}) + \varepsilon$, where $\boldsymbol{\theta}^* \in \mathbb{R}^d$ is an unknown parameter shared by all arms and $\varepsilon$ is observation noise. In this work, we consider contextual bandits with nonlinear reward model $f(\boldsymbol{\theta}^*; \boldsymbol{x})$. The goal of a bandit algorithm is to maximize cumulative reward over a horizon $T$, equivalently to minimize the pseudo-regret (Lattimore & Szepesvári, 2020): $R(T) = \sum_{t=1}^{T}(f(\boldsymbol{\theta}^*; \boldsymbol{x}_t^\star) - f(\boldsymbol{\theta}^*; \boldsymbol{x}_t))$, where $\boldsymbol{x}_t^* = \operatorname{argmax}_{\boldsymbol{x} \in \mathcal{X}_t} f(\boldsymbol{\theta}^*; \boldsymbol{x})$ is the arm with the highest expected reward at round $t$.

**Multi-task Bandit Problem**   We consider the multi-task nonlinear contextual bandit problem. All the bandit tasks share the same reward model structure $y(\boldsymbol{x}) = f(\boldsymbol{\theta}^*; \boldsymbol{x}) + \varepsilon$ and the underlying parameter $\boldsymbol{\theta}^*$ varies across tasks but is drawn independently from a common prior distribution $p_0(\boldsymbol{\theta}^*)$. In our problem setup, we have model parameter $\{\boldsymbol{\theta}^{(i)}\}_{i=1}^N$ from previous $N$ bandit tasks, which can be ground-truth (perfect data) or estimation (imperfect data). Our goal is to utilize these data to help solve for a new bandit task $(N + 1)$.

**Diffusion Models**   Score-based diffusion models define a generative process as the reverse of a continuous-time noising process that gradually perturbs data toward a tractable reference distribution. Let $\boldsymbol{\theta}_s \in \mathbb{R}^d$ denote the state at continuous time $s \in [0, S]$, with $\boldsymbol{\theta}_0 \sim p_0$ and $\boldsymbol{\theta}_S \sim \mathcal{N}(\mathbf{0}, \mathbf{I})$. In the variance-preserving (VP) formulation (Song et al., 2021), the forward SDE is: $\mathrm{d}\boldsymbol{\theta}_s = -\frac{1}{2}\beta_s\boldsymbol{\theta}_s\mathrm{d}s + \sqrt{\beta_s}\mathrm{d}\boldsymbol{w}_s$, where $\beta_s > 0$ is a noise schedule and $\boldsymbol{w}_s$ is a standard $d$-dimensional Wiener process. The corresponding reverse-time diffusion that transports $\boldsymbol{\theta}_S \sim \mathcal{N}(\mathbf{0}, \mathbf{I})$ back to $p_0$. The reverse diffusion SDE from $s = S$ down to $s = 0$ is as follows:

$$\mathrm{d}\boldsymbol{\theta}_s = [-\tfrac{1}{2}\beta_s\boldsymbol{\theta}_s - \beta_s\nabla_{\boldsymbol{\theta}}\log p_s(\boldsymbol{\theta}_s)]\mathrm{d}s + \sqrt{\beta_s}\mathrm{d}\bar{\boldsymbol{w}}_s,$$

where $\bar{\boldsymbol{w}}_s$ a standard Wiener process in reverse time and $\nabla_{\boldsymbol{\theta}}\log p_s(\boldsymbol{\theta}_s)$ is the time-dependent score function where $p_s(\boldsymbol{\theta}_s) = \int p_0(\boldsymbol{\theta}_0)p(\boldsymbol{\theta}_s|\boldsymbol{\theta}_0)\mathrm{d}\boldsymbol{\theta}_0$. Since score function $\nabla_{\boldsymbol{\theta}}\log p_s(\boldsymbol{\theta}_s)$ is unknown, a neural network $s_{\boldsymbol{\psi}}(\boldsymbol{\theta}_s, s)$ is trained to approximate it via denoising score matching:

$$\boldsymbol{\psi}^* = \operatorname{argmin}_{\boldsymbol{\psi}} \mathbb{E}_{s \sim \mathcal{U}(\epsilon, 1), \boldsymbol{\theta}_0 \sim p_0, \boldsymbol{\theta}_s \sim p(\boldsymbol{\theta}_s|\boldsymbol{\theta}_0)} \|s_{\boldsymbol{\psi}}(\boldsymbol{\theta}_s, s) - \nabla_{\boldsymbol{\theta}}\log p(\boldsymbol{\theta}_s|\boldsymbol{\theta}_0)\|_2^2,$$

where $\epsilon > 0$ is a small constant and $p(\boldsymbol{\theta}_s|\boldsymbol{\theta}_0)$ is Gaussian under the VP forward process. Plugging $s_{\boldsymbol{\psi}^*}$ into the reverse SDE and discretizing yields a sampler that approximately draws from $p_0$.

To obtain a practical training and sampling procedure, we discretize the time index into $L$ noise levels indexed by $\ell \in \{1, \ldots, L\}$ and adopt discrete VP formulation. We define a discrete noise schedule $\beta_\ell > 0$, and define $\alpha_\ell = 1 - \beta_\ell, \bar{\alpha}_\ell = \prod_{j=1}^{\ell} \alpha_j$. The forward process has the closed form: $\boldsymbol{\theta}_\ell = \sqrt{\bar{\alpha}_\ell}\boldsymbol{\theta}_0 + \sqrt{1 - \bar{\alpha}_\ell}\boldsymbol{z}$ where $\boldsymbol{z} \sim \mathcal{N}(\mathbf{0}, \mathbf{I})$. The reverse process involves iteratively sampling $\boldsymbol{\theta}_{\ell-1}$ from the posterior distribution $p(\boldsymbol{\theta}_{\ell-1}|\boldsymbol{\theta}_\ell)$. Instead of directly approximating the score function $\nabla_{\boldsymbol{\theta}_\ell}\log p_s(\boldsymbol{\theta}_\ell)$, it is common practice to reparameterize the model to predict the added noise $\boldsymbol{z}$ at step $\ell$ by denoiser network $\varepsilon_{\boldsymbol{\phi}^*}$. The reverse update then becomes an ancestral Gaussian sampling step: $\boldsymbol{\theta}_{\ell-1} \sim \mathcal{N}(\boldsymbol{\mu}_\ell, \boldsymbol{\Sigma}_\ell)$, where $\boldsymbol{\mu}_\ell = \frac{1}{\sqrt{\alpha_\ell}}(\boldsymbol{\theta}_\ell - \frac{1-\alpha_\ell}{\sqrt{1-\bar{\alpha}_\ell}}\varepsilon_{\boldsymbol{\phi}^*}(\boldsymbol{\theta}_\ell, \ell))$, $\boldsymbol{\Sigma}_\ell = \frac{1-\bar{\alpha}_{\ell-1}}{1-\bar{\alpha}_\ell}\beta_\ell\mathbf{I}$. This formulation is equivalent to DDPM (Ho et al., 2020) and serves as backbone for our diffusion prior.

# 3   DIFFUSION-BASED POSTERIOR SAMPLING IN MULTI-TASK NONLINEAR CONTEXTUAL BANDIT

In this section, we introduce the diffusion posterior sampling framework for multi-task nonlinear contextual bandits. We begin with the core components of a unified diffusion reverse update, and then instantiate it in two ways, following the prevailing classes of diffusion models in the literature.

## 3.1   A UNIFIED DIFFUSION REVERSE UPDATE FOR DECISION-MAKING

We first recall posterior sampling in contextual bandits. Thompson Sampling (TS) maintains a posterior over reward model parameters, constructed from a hand-specified conjugate prior and the observed rewards. At round $t$, given history $\mathcal{H}_t$ and prior $p(\boldsymbol{\theta})$, TS samples a parameter estimate $\widetilde{\boldsymbol{\theta}}_t \sim p(\boldsymbol{\theta}|\mathcal{H}_t) \propto p(\boldsymbol{\theta})\,p(\mathcal{H}_t|\boldsymbol{\theta})$ and selects the arm $\boldsymbol{x}_t = \operatorname{argmax}_{\boldsymbol{x} \in \mathcal{X}_t} f(\widetilde{\boldsymbol{\theta}}_t; \boldsymbol{x})$. However, such simple priors cannot capture the rich cross-task structure in multi-task settings, leading to misspecified posteriors and suboptimal exploration. To address this, we train a diffusion model on data from

past tasks to learn a flexible prior distribution over task parameters. For a new task, we run a conditional reverse-diffusion process that incorporates the interaction history. At each round, we initialize from noise and perform an $L$-step conditional reverse diffusion to produce a posterior sample (i.e., a parameter estimate) for arm selection.

We unify diffusion-based posterior sampling for decision-making with a single per-level reverse update (omitting coefficients):

$$\underbrace{\boldsymbol{\theta}_{\ell-1}}_{\text{next state}} \leftarrow \underbrace{\boldsymbol{\theta}_{\ell}}_{\text{current state}} + \underbrace{\text{unconditional term}}_{\text{diffusion prior drift}} + \underbrace{\text{likelihood term}}_{\text{data drift}} + \underbrace{\text{noise term}}_{\text{randomized exploration}}, \tag{3.1}$$

The unconditional term is the standard reverse step from a pretrained diffusion model, implemented via either a score network or a denoiser network. The likelihood term incorporates bandit interaction history up to round $t-1$ by defining a loss $L_t$, evaluating it at a diffusion-informed argument $\Phi_\ell(\boldsymbol{\theta}_\ell)$ (the rescaled state or Tweedie estimate), and applying a gradient step $-\eta_\ell \nabla_\theta L_t(\Phi_\ell(\boldsymbol{\theta}_\ell))$. The noise term enables randomized exploration, either as DDPM noise in the diffusion prior drift or Langevin noise from approximate sampling. This decomposition (3.1) separates offline knowledge (the diffusion prior drift) from online adaptation (the data drift).

In the next two subsections, we instantiate (3.1) in two ways. First, **DLTS** follows the TS workflow and performs Langevin updates at each level using the rescaled state, yielding a diffusion-prior approximate sampler for nonlinear reward models. Second, **DPSG** preserves the reverse-diffusion drift and adds a single likelihood drift per level via the Tweedie estimate, resulting in a simple and efficient conditional sampler. Both methods follows the unified diffusion posterior sampling framework in (3.1), while each admits its own interpretation and derivation.

## 3.2 DIFFUSION LANGEVIN THOMPSON SAMPLING

To handle contextual bandits with complex priors, Kveton et al. (2024) proposed DiffTS for linear and generalized linear models with specialized closed-form updates. For nonlinear bandits, such closed forms are unavailable, making algorithm design harder. Motivated by LMC-TS (Xu et al., 2022), an approximate sampling method for nonlinear bandits, we propose **Diffusion Langevin Thompson Sampling (DLTS)** (Algorithm 1). DLTS extends approximate TS to nonlinear rewards (including neural networks) by using a learned diffusion prior.

---

**Algorithm 1** Diffusion Langevin Thompson Sampling (DLTS)

**Input:** Diffusion denoiser $\varepsilon_{\boldsymbol{\phi}^*}(\cdot, \cdot)$, noise schedule $\{\beta_\ell\}_{\ell=1}^L$, learning rate $\{\eta_\ell\}_{\ell=1}^L$.

1: $\alpha_\ell \leftarrow 1 - \beta_\ell$ and $\bar{\alpha}_\ell \leftarrow \prod_{j=1}^\ell \alpha_j$ for all $\ell$
2: **for** $t = 1, 2, \cdots, T$ **do**
3:     Receive contextual vector $\{\boldsymbol{x}_t(a)\}_{a \in \mathcal{A}}$.
4:     $(\boldsymbol{\mu}_{L+1}, \boldsymbol{\Sigma}_{L+1}) \leftarrow (\mathbf{0}, \mathbf{I})$.
5:     **for** $\ell = L+1, L, \cdots, 1$ **do**
6:         $\boldsymbol{\theta}_{\ell,0} \leftarrow \boldsymbol{\mu}_\ell$.
7:         **for** $k = 0, 1, \cdots, K_\ell - 1$ **do**
8:             Sample $\boldsymbol{\xi}_k \sim \mathcal{N}(\mathbf{0}, \mathbf{I})$.
9:             $\boldsymbol{\theta}_{\ell,k+1} \leftarrow \left(1 - \frac{(1-\bar{\alpha}_\ell)\eta_\ell}{(1-\bar{\alpha}_{\ell-1})\beta_\ell}\right)\boldsymbol{\theta}_{\ell,k} + \frac{(1-\bar{\alpha}_\ell)\eta_\ell}{(1-\bar{\alpha}_{\ell-1})\beta_\ell}\boldsymbol{\mu}_\ell - \eta_\ell \nabla_\theta L_t\left(\boldsymbol{\theta}_{\ell,k}/\sqrt{\bar{\alpha}_{\ell-1}}\right) + \sqrt{2\eta_\ell\zeta_\ell^{-1}}\boldsymbol{\xi}_k$.
10:         **end for**
11:         $\boldsymbol{\theta}_{\ell-1} \leftarrow \boldsymbol{\theta}_{\ell,K_\ell}$.
12:         $\boldsymbol{\mu}_{\ell-1} \leftarrow \frac{1}{\sqrt{\alpha_{\ell-1}}}\left(\boldsymbol{\theta}_{\ell-1} - \frac{1-\alpha_{\ell-1}}{\sqrt{1-\bar{\alpha}_{\ell-1}}}\varepsilon_{\boldsymbol{\phi}^*}(\boldsymbol{\theta}_{\ell-1}, \ell-1)\right)$.
13:     **end for**
14:     Let $\widetilde{\boldsymbol{\theta}}_t \leftarrow \boldsymbol{\theta}_0$, take action $\boldsymbol{x}_t = \arg\max_{\boldsymbol{x}_t(a)} f\left(\widetilde{\boldsymbol{\theta}}_t; \boldsymbol{x}_t(a)\right)$.
15:     Receive reward $y_t$ and update history $\mathcal{H}_{t+1} = \{(\boldsymbol{x}_i, y_i)\}_{i=1}^t$.
16: **end for**

---

Our method follows the TS workflow, which builds a posterior over parameters and samples for arm selection, but it differs in three ways: (1) It replaces a hand-crafted prior with a diffusion prior trained on past tasks. (2) Instead of a single posterior update, it runs a reverse diffusion process and injects interaction history at each level, building the conditional posterior gradually. Formally, the reverse process is a Markov chain with the per-level transition in the form:

$$p(\boldsymbol{\theta}_{\ell-1}|\boldsymbol{\theta}_\ell, \mathcal{H}_t) \propto p(\mathcal{H}_t|\boldsymbol{\theta}_{\ell-1})p(\boldsymbol{\theta}_{\ell-1}|\boldsymbol{\theta}_\ell), \quad \ell = L, \ldots, 1, \tag{3.2}$$

where $\boldsymbol{\theta}_\ell \in \mathbb{R}^d$ is the noisy parameter at level $\ell$ and the terminal $\boldsymbol{\theta}_0$ is used as the parameter sample for action selection. (3) It applies iterative LMC updates to approximately sample from (3.2).

After training the diffusion denoiser $\varepsilon_{\boldsymbol{\phi}^*}(\cdot, \cdot)$ (see Appendix G for more details), we obtain the diffusion model parameter $(\boldsymbol{\mu}_\ell, \boldsymbol{\Sigma}_\ell)_{\ell \in [L+1]}^1$ for unconditional sampling in each reverse step: $p(\boldsymbol{\theta}_{\ell-1}|\boldsymbol{\theta}_\ell) = \mathcal{N}(\boldsymbol{\theta}_{\ell-1}; \boldsymbol{\mu}_\ell, \boldsymbol{\Sigma}_\ell)$, where $\boldsymbol{\mu}_\ell = \frac{1}{\sqrt{\alpha_\ell}}(\boldsymbol{\theta}_\ell - \frac{1-\alpha_\ell}{\sqrt{1-\bar{\alpha}_\ell}}\varepsilon_{\boldsymbol{\phi}^*}(\boldsymbol{\theta}_\ell, \ell))$, $\boldsymbol{\Sigma}_\ell = \frac{1-\bar{\alpha}_{\ell-1}}{1-\bar{\alpha}_\ell}\beta_\ell \mathbf{I}$. Note that at step $t$, the nonlinear reward model is defined as $y_t = f(\boldsymbol{\theta}^*; \boldsymbol{x}_t) + \varepsilon_t$. When we assume Gaussian noise $\varepsilon_t \sim \mathcal{N}(0, \sigma^2)$, then the likelihood becomes $p(\mathcal{H}_t|\boldsymbol{\theta}) \propto \exp(-\sigma^{-2}\sum_{i=1}^t (f(\boldsymbol{\theta}; \boldsymbol{x}_i) - y_i)^2) = \exp(-L_t(\boldsymbol{\theta}))$, where $L_t(\boldsymbol{\theta}) = \sigma^{-2}\sum_{i=1}^t (f(\boldsymbol{\theta}; \boldsymbol{x}_i) - y_i)^2$. Therefore, we can approximate the posterior in each reverse step for conditional sampling as follows (refer to Appendix C.1 for details of derivation),

$$p(\boldsymbol{\theta}_{\ell-1}|\boldsymbol{\theta}_\ell, \mathcal{H}_t) \propto \exp(-L_t(\boldsymbol{\theta}_{\ell-1}/\sqrt{\bar{\alpha}_{\ell-1}})) \cdot \mathcal{N}(\boldsymbol{\theta}_{\ell-1}; \boldsymbol{\mu}_\ell, \boldsymbol{\Sigma}_\ell).$$

To sample from $p(\boldsymbol{\theta}_{\ell-1}|\boldsymbol{\theta}_\ell, \mathcal{H}_t)$, we apply Langevin Monte Carlo for approximate sampling. Specifically, at round $t$ and diffusion reverse step $\ell$, we iteratively conduct the Langevin update,

$$\boldsymbol{\theta}_{\ell,k+1} = \boldsymbol{\theta}_{\ell,k} - \eta_\ell\Big[\nabla_{\boldsymbol{\theta}}L_t\big(\boldsymbol{\theta}_{\ell,k}/\sqrt{\bar{\alpha}_{\ell-1}}\big) + \boldsymbol{\Sigma}_\ell^{-1}\big(\boldsymbol{\theta}_{\ell,k} - \boldsymbol{\mu}_\ell\big)\Big] + \sqrt{2\eta_\ell\zeta_\ell^{-1}}\boldsymbol{\xi}_k,$$

where $\boldsymbol{\xi}_k \sim \mathcal{N}(\mathbf{0}, \mathbf{I})$, $\eta_\ell$ is the step size, and $\zeta_\ell$ is the temperature. We initialize $\boldsymbol{\theta}_{\ell,0} = \boldsymbol{\mu}_\ell$, after $K_\ell$ iterations, we set $\boldsymbol{\theta}_{\ell-1} = \boldsymbol{\theta}_{\ell,K_\ell}$ as the sample from the reverse step.

Specially, when we only conduct one LMC update ($K_\ell = 1$), we have the reverse update

$$\boldsymbol{\theta}_{\ell-1} \leftarrow \big(1 - \tfrac{(1-\bar{\alpha}_\ell)\eta_\ell}{(1-\bar{\alpha}_{\ell-1})\beta_\ell}\big)\boldsymbol{\theta}_\ell + \tfrac{(1-\bar{\alpha}_\ell)\eta_\ell}{(1-\bar{\alpha}_{\ell-1})\beta_\ell}\boldsymbol{\mu}_\ell - \eta_\ell\nabla_{\boldsymbol{\theta}}L_t\big(\boldsymbol{\theta}_\ell/\sqrt{\bar{\alpha}_{\ell-1}}\big) + \sqrt{2\eta_\ell\zeta_\ell^{-1}}\boldsymbol{\xi}. \qquad (3.3)$$

Note that (3.3) aligns with our unified update (3.1). This confirms the rationality of our framework and the essence of diffusion-based posterior sampling. Intuitively, DLTS explicitly forms a conditional posterior by incorporating the history into the diffusion prior at each reverse step, then draws samples from it via approximate sampling.

## 3.3 DIFFUSION POSTERIOR SAMPLING WITH GUIDANCE

We now give a simpler and faster realization inspired by Diffusion Posterior Sampling (DPS) (Chung et al., 2023), a common approach for conditional sampling in inverse problems. DPS augments the unconditional reverse step with a likelihood score term and then runs the reverse chain, yielding a conditional sampler. With a pretrained unconditional process, drawing a parameter sample for arm selection is straightforward: run the unconditional reverse steps and add a likelihood drift for guidance. This requires only the unconditional score and a tractable proxy for the likelihood at the Tweedie estimate, so there is no need to construct a per-level conditional posterior and apply approximate sampling as in DLTS. Following this design, we propose **Diffusion Posterior Sampling with Guidance (DPSG)**, shown in Algorithm 2.

To derive diffusion reverse update (Line 8 in Algorithm 2), we follow Chung et al. (2023) and decompose the conditional score into two terms,

$$\nabla_{\boldsymbol{\theta}} \log p_s(\boldsymbol{\theta}_s|\mathcal{H}_t) = \underbrace{\nabla_{\boldsymbol{\theta}} \log p_s(\boldsymbol{\theta}_s)}_{\text{unconditional score}} + \underbrace{\nabla_{\boldsymbol{\theta}} \log p_s(\mathcal{H}_t|\boldsymbol{\theta}_s)}_{\text{likelihood score}}.$$

For the first term, we can use pretrained diffusion model score network $s_{\boldsymbol{\psi}^*}$ to approximate the unconditional score. For the second term, we approximate the likelihood score via Tweedie's formula. Since $p(\mathcal{H}_t|\boldsymbol{\theta}) \propto \exp(-L_t(\boldsymbol{\theta}))$, based on Tweedie's formula (Efron, 2011), we can obtain a tractable approximation for $\nabla_{\boldsymbol{\theta}} \log p_s(\mathcal{H}_t|\boldsymbol{\theta}_s)$ (refer to Appendix C.2 for details of derivation): $\nabla_{\boldsymbol{\theta}} \log p_s(\mathcal{H}_t|\boldsymbol{\theta}_s) \simeq -\nabla_{\boldsymbol{\theta}}L_t(\hat{\boldsymbol{\theta}}_0(\boldsymbol{\theta}_s, s))$, where $\hat{\boldsymbol{\theta}}_0(\boldsymbol{\theta}_s, s) \simeq 1/\sqrt{\bar{\alpha}_s}(\boldsymbol{\theta}_s + (1 - \bar{\alpha}_s)s_{\boldsymbol{\psi}^*}(\boldsymbol{\theta}_s, s))$. Therefore, we can further have the approximation of the conditional score of reverse dynamics,

$$\nabla_{\boldsymbol{\theta}} \log p_s(\boldsymbol{\theta}_s|\mathcal{H}_t) \simeq s_{\boldsymbol{\psi}^*}(\boldsymbol{\theta}_s, s) - \nabla_{\boldsymbol{\theta}}L_t\big(\hat{\boldsymbol{\theta}}_0(\boldsymbol{\theta}_s, s)\big). \qquad (3.4)$$

In practice, to obtain a discrete-time algorithm we use DDPM ancestral sampling (Ho et al., 2020) to implement conditional sampling via the conditional score (3.4). The reverse update is done by

---

[1]We define $(\boldsymbol{\mu}_{L+1}, \boldsymbol{\Sigma}_{L+1}) = (\mathbf{0}, \mathbf{I})$ for notation simplicity.

---

**Algorithm 2** Diffusion Posterior Sampling with Guidance (DPSG)

---

**Input:** score network $s_{\psi^*}(\cdot, \cdot)$, noise schedule $\{\beta_\ell\}_{\ell=1}^L$, learning rate $\{\eta_\ell\}_{\ell=1}^L$.

1: $\alpha_\ell \leftarrow 1 - \beta_\ell$ and $\bar{\alpha}_\ell \leftarrow \prod_{j=1}^\ell \alpha_j$ for all $\ell$.
2: **for** $t = 1, 2, \cdots, T$ **do**
3:     Receive contextual vector $\{x_t(a)\}_{a \in \mathcal{A}}$.
4:     $\theta_L \sim \mathcal{N}(0, I)$.
5:     **for** $\ell = L, L-1, \cdots, 1$ **do**
6:         $\hat{\theta}_0(\theta_\ell, \ell) \leftarrow \frac{1}{\sqrt{\bar{\alpha}_\ell}}(\theta_\ell + (1 - \bar{\alpha}_\ell)s_{\psi^*}(\theta_\ell, \ell))$.
7:         $z_\ell \sim \mathcal{N}(0, \beta_\ell I)$.
8:         $\theta_{\ell-1} \leftarrow \frac{1}{\sqrt{\alpha_\ell}}\theta_\ell + \frac{\beta_\ell}{\sqrt{\alpha_\ell}}s_{\psi^*}(\theta_\ell, \ell) - \eta_\ell \nabla_\theta L_t(\hat{\theta}_0(\theta_\ell, \ell)) + z_\ell$.
9:     **end for**
10:    Let $\widetilde{\theta}_t \leftarrow \theta_0$, take action $x_t = \operatorname{argmax}_{x_t(a)} f(\widetilde{\theta}_t; x_t(a))$.
11:    Receive reward $y_t$ and update history $\mathcal{H}_{t+1} = \{(x_i, y_i)\}_{i=1}^t$.
12: **end for**

---

two steps. First, we use the unconditional score $s_{\psi^*}(\theta_\ell, \ell)$ through Tweedie's estimate $\hat{\theta}_0(\theta_\ell, \ell)$ to form the Gaussian posterior mean, then we sample

$$\theta'_{\ell-1} \leftarrow \frac{\sqrt{\alpha_\ell}(1 - \bar{\alpha}_{\ell-1})}{1 - \bar{\alpha}_\ell}\theta_\ell + \frac{\sqrt{\bar{\alpha}_{\ell-1}}\beta_\ell}{1 - \bar{\alpha}_\ell}\hat{\theta}_0(\theta_\ell, \ell) + z_\ell, \quad z_\ell \sim \mathcal{N}(0, \beta_\ell I), \quad (3.5)$$

Based on (3.4), the second step is that we make this update conditional by subsequently adding one likelihood-based term. In practice, after sampling $\theta'_{\ell-1}$, we take a gradient step using the likelihood $L_t$ at the Tweedie estimate for guidance: $\theta_{\ell-1} \leftarrow \theta'_{\ell-1} - \eta_\ell \nabla_\theta L_t(\hat{\theta}_0(\theta_\ell, \ell))$. When we merge this with (3.5), we obtain line 8 in Algorithm 2,

$$\theta_{\ell-1} \leftarrow \frac{1}{\sqrt{\alpha_\ell}}\theta_\ell + \frac{\beta_\ell}{\sqrt{\alpha_\ell}}s_{\psi^*}(\theta_\ell, \ell) - \eta_\ell \nabla_\theta L_t(\hat{\theta}_0(\theta_\ell, \ell)) + z_\ell. \quad (3.6)$$

Note that (3.6) aligns with our unified update (3.1). This confirms the rationality of our framework and the essence of diffusion-based posterior sampling. However, from the intuition perspective, DPSG is to use history information as guidance after unconditional update based on the diffusion prior. This is different from DLTS, which directly integrates history information into the unconditional update based on the diffusion prior to achieve conditional sampling.

**Empirical Variant of DPSG** DPSG is simple and fast, but it relies on the accuracy of likelihood-score at the Tweedie estimate. A recent work Xu et al. (2025) observes that DPS has the properties of high bias and low diversity, thus behaving like an implicit, but unstable, MAP estimator. Therefore, multi-step projection is used to solve this MAP optimization problem. Motivated by Xu et al. (2025), to stabilize learning, we introduce **DPSG with Multi-step Projection (DPSG-MP)** (Algorithm 3 in Appendix C.3), which is a practical DPSG variant that replaces the single correction with a short inner loop of gradient ascent on the likelihood score, and after each step projects the iterate onto the sphere where the reverse transition $p_{\psi^*}(\theta_{\ell-1}|\theta_\ell)$ concentrates around. Specifically, we first initialize $\theta_{\ell-1,0} \leftarrow 1/\sqrt{\alpha_\ell}\theta_\ell + \beta_\ell/\sqrt{\alpha_\ell}s_{\psi^*}(\theta_\ell, \ell) + z_\ell$, then we iteratively update

$$\theta_{\ell-1,k+1} \leftarrow \operatorname{Proj}(\theta_{\ell-1,k} - \eta_\ell \nabla_\theta L_t(\hat{\theta}_0(\theta_\ell, \ell))), \quad k = 0, 1, \ldots, K_\ell,$$

where function $\operatorname{Proj}(\cdot)$ projects the input onto sphere surface $\mathcal{S}(p_{\psi^*}(\theta_{\ell-1}|\theta_\ell))$, with radius $r_\ell = \|z_\ell\|_2$ and center at $\mathbb{E}[\theta_{\ell-1}|\theta_\ell]$. Finally, we assign $\theta_{\ell-1,K_\ell}$ to $\theta_{\ell-1}$ and finish the update. We provide the complete algorithm and further explanation in Appendix C.3.

## 4 THEORETICAL RESULTS

In this section, we analyze the connection between Thompson sampling and diffusion Thompson Sampling. For convenience of analysis, we first define two oracle algorithms.

**Definition 4.1.** *We define the following oracle algorithm as* **Oracle Thompson Sampling (OTS)***, which applies exact posterior sampling and greedy policy according to the true prior and likelihood:*

*1) Use true prior $p_0$.*
*2) Maintain exact posterior in each round: $p_t(\theta) \equiv p(\theta|\mathcal{H}_t) \propto p_0(\theta) \prod_{i=1}^{t-1} p(y_i|\theta, x_i)$.*

*3) Sample $\boldsymbol{\theta}_t^{OTS} \sim p_t(\boldsymbol{\theta})$, then select arm $\boldsymbol{x}_t = \arg\max_{\boldsymbol{x} \in \mathcal{X}_t} f(\boldsymbol{\theta}_t^{OTS}; \boldsymbol{x})$.*

**Definition 4.2.** *We define the following oracle algorithm as **Oracle Diffusion Thompson Sampling (ODTS)**, which performs posterior sampling via a reverse-diffusion chain, using the same pretrained diffusion model as DLTS (Algorithm 1).*

*1) Train a diffusion reverse process to learn prior $p_0$ and obtain score/denoiser network $s_{\boldsymbol{\psi}^*}(\cdot, \cdot)$.*
*2) The per-level conditional reverse process is implemented exactly: $\widetilde{p}_t(\boldsymbol{\theta}_{0:L}) \equiv p(\boldsymbol{\theta}_{0:L}|\mathcal{H}_t) = p(\boldsymbol{\theta}_L|\mathcal{H}_t) \prod_{\ell=1}^{L} p(\boldsymbol{\theta}_{\ell-1}|\boldsymbol{\theta}_\ell, \mathcal{H}_t)$ and $\widetilde{p}_t(\boldsymbol{\theta}) = \int \widetilde{p}_t(\boldsymbol{\theta}_{0:L}) d\boldsymbol{\theta}_{1:L} = \int p(\boldsymbol{\theta}_{0:L}|\mathcal{H}_t) d\boldsymbol{\theta}_{1:L}$.*
*3) Sample $\boldsymbol{\theta}_t^{ODTS} \sim \widetilde{p}_t(\boldsymbol{\theta})$, then selects arm $\boldsymbol{x}_t = \arg\max_{\boldsymbol{x} \in \mathcal{X}_t} f(\boldsymbol{\theta}_t^{ODTS}; \boldsymbol{x})$.*

We make the following assumption to measure the accuracy of score estimation.

**Assumption 4.3.** *For any diffusion time $s \in [0, \mathcal{S}]$ and let $p_s$ denote the $s$-marginal distribution of the forward diffusion over parameters $\boldsymbol{\theta} \in \Theta \subseteq \mathbb{R}^d$. Let $s_{\boldsymbol{\psi}^*}(\boldsymbol{\theta}, s)$ be the learned score estimator. Assume there exists $\epsilon_{score}$ such that for all $s$ and $\boldsymbol{\theta}$, $\|s_{\boldsymbol{\psi}^*}(\boldsymbol{\theta}, s) - \nabla_{\boldsymbol{\theta}} \log p_s(\boldsymbol{\theta})\|_2 \leq \epsilon_{score}$.*

Note that the key distinction between OTS and ODTS is whether the reverse-diffusion process is trained to recover the true prior $p_0$. The theorem below shows that if ODTS learns the prior exactly (i.e $\epsilon_{\text{score}} = 0$ in Assumption 4.3), then OTS and ODTS are equivalent.

**Theorem 4.4.** *If ODTS perfectly train a diffusion reverse process to get true prior $p_0$ and the score/denoiser network matches the true score at every diffusion level $\ell$: $s_{\boldsymbol{\psi}^*}(\cdot, \ell) \equiv \nabla_{\boldsymbol{\theta}} \log p_\ell(\cdot)$, then oracle algorithms OTS and ODTS are equivalent in the sense that they induce the same posterior distribution $\widetilde{p}_t(\boldsymbol{\theta}) = p_t(\boldsymbol{\theta})$ and both algorithms induce the same arm selection distribution.*

**Theorem 4.5.** *For any round $t \in [T]$, denote $p_t(\boldsymbol{\theta})$ as the exact posterior used by OTS, and $\widetilde{p}_t(\boldsymbol{\theta})$ as the marginal posterior used by ODTS. Assume the reward is bounded: for all $(\boldsymbol{\theta}, \boldsymbol{x})$, $|f(\boldsymbol{\theta}; \boldsymbol{x})| \leq f_{max}$. Then the per-round expected regret gap between OTS and ODTS is bounded by $\mathbb{E}[r_t^{ODTS} - r_t^{OTS}] \leq \Delta_t^{Score} = 2f_{max} \sum_{\ell=1}^{L} \kappa_\ell \epsilon_{score}$, where coefficient $\kappa_\ell$ is determined by noise schedule.*

## 5 EXPERIMENTS

In this section, we conduct experiments to evaluate our proposed algorithms DLTS and a variant of DPSG named DPSG-MP (refer to Appendix C.3). We aim to demonstrate the effectiveness of our proposed algorithms in the different settings such as the linear and non-linear contextual bandit. We conduct these experiments on simulation experiments.

### 5.1 IMPLEMENTATIONS

Our diffusion prior is implemented within a Denoising Diffusion Probabilistic Model (DDPM) framework, using the standard $\epsilon$-prediction parameterization. The denoiser network is an MLP that incorporates time embeddings. For synthetic simulations, we use $L = 100$ diffusion steps and a linear noise schedule with a constant $\beta_\ell$. A full description of the architectures and hyperparameters is available in Appendix H.

### 5.2 SIMULATION

**Prior Distribution** We follow Kveton et al. (2024) to design the prior over task parameters. We consider six prior distributions, including the 'cross', 'rays', 'triangles', 'swirl', 'H' and 'corners' in our simulation experiments. Each prior distribution has 10000 samples with $d = 2$ parameters defined as $\boldsymbol{\theta}$. We use $80\%$ of the samples as the training samples, which refer to the previous task. For the remaining $20\%$ samples, we use them as the test samples, which refer to the new tasks. We illustrate the prior distribution in Figure 6 in Appendix H.2. Based on the prior distribution, we design the linear and non-linear contextual bandit on different reward models.

**Linear Bandit** We first evaluate our methods in the linear contextual bandit setting. For each task, the ground-truth parameter $\boldsymbol{\theta}$ is drawn from the test set of a given prior distribution, and rewards are generated according to the model $r = \boldsymbol{x}^\top \boldsymbol{\theta} + \varepsilon$ where the $\varepsilon \sim \mathcal{N}(0, 1)$. We compare our proposed DLTS and DPSG-MP which is a more stable version of DPSG with several baseline methods, including the LinTS (Agrawal & Goyal, 2013b), LinUCB (Chu et al., 2011), $\epsilon$-greedy and DiffTS (Kveton et al., 2024). For LinTS, LinUCB, and $\epsilon$-greedy algorithms, we directly conduct them in the test bandit task. Our methods and DiffTS leverage a diffusion prior trained on the $80\%$ training

partition of the parameter data. Performance is measured over 64 new tasks per trial, with each task running for a horizon of 200 steps. To ensure statistical robustness, all results are averaged over 8 independent trials. The quality of the learned diffusion prior is visualized in Figure 1, and the comparative regret performance is summarized in Figure 2.

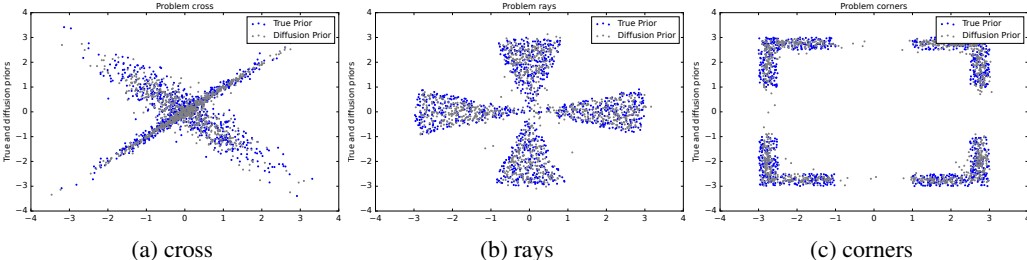

(a) cross          (b) rays          (c) corners

Figure 1: Visualization of the learned diffusion prior versus the true prior. We demonstrate three of them including cross, rays and corners. For each figure, samples from our trained model (in grey) are overlaid on the ground-truth samples (in blue). More results can be found in Appendix I.

First, Figure 1 validates the quality of our diffusion model, confirming that it accurately captures the complex structure of the true prior distributions. From the regret comparison in Figure 2, we observe that our methods DLTS and DPSG-MP achieve cumulative regret that is highly competitive with, and in some cases on par with, the specialized baseline algorithms. This is a noteworthy outcome, as methods like LinTS and DiffTS are designed specifically for the linear setting and leverage efficient closed-form posterior updates. In contrast, our framework relies on a more general, gradient-based sampling approach. The strong performance of our methods in a setting where they cannot exploit such closed-form solutions underscores the robustness of our framework and highlights its potential for nonlinear environments where those specialized updates are no longer applicable. The good performance of the algorithms also demonstrates the benefits of leveraging the diffusion prior learned on the previous task in solving the new tasks with the same prior distribution.

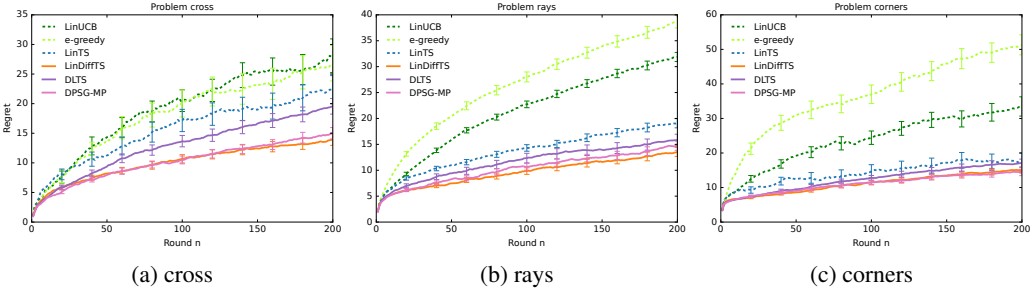

(a) cross          (b) rays          (c) corners

Figure 2: Performance of all algorithms on the linear contextual bandit tasks for the three prior distributions. Our proposed methods achieve cumulative regret comparable to LinTS and DiffTS. More results can be found in Appendix I.

**Nonlinear Bandit** We also evaluate our framework in scenarios where the reward function is inherently nonlinear. We construct three such environments where the reward is defined by $r = f(\boldsymbol{x}^\top \boldsymbol{\theta}) + \varepsilon$, with $\varepsilon \sim \mathcal{N}(0, 1)$. The nonlinear functions $f(\cdot)$ we consider are a cosine model ($f(z) = \cos(3z)$), a quadratic model ($f(z) = z^2$), and a sigmoid-gated model ($f(z) = 2z \cdot \mathrm{sigmoid}(z) + 1$). Due to the space limit, the results of the cosine model and the quadratic model in Appendix I.2.

In this challenging setting, our proposed methods DLTS and DPSG-MP, use a neural network (MLP) to represent the reward function. A key distinction from prior sections is that the diffusion model is now trained to learn a prior directly over the parameters of this MLP, using network weights from past tasks as training data. This setup tests the ability of our framework to handle priors in the high-dimensional weight space of neural networks. We compare our proposed algorithms with several baseline methods, including the LinTS, DiffTS, NeuralTS (Zhang et al., 2021) and NeuralUCB(Zhou et al., 2020). Visualizations of the learned diffusion priors over the MLP parameters for each reward model are provided in Figure 3.

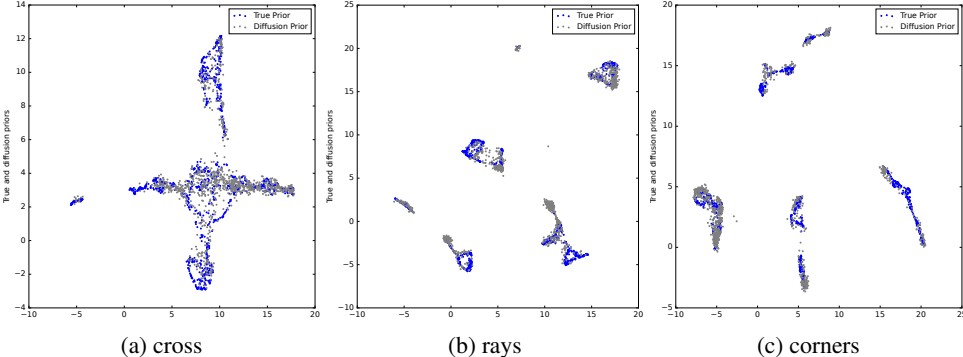

| (a) cross | (b) rays | (c) corners |

Figure 3: Visualization of the learned diffusion prior versus the true prior (under sigmoid-gated reward model). The true prior is the MLP parameters learned from the previous task. Samples from our trained model (grey) are overlaid on the ground-truth samples (blue). More results can be found in Appendix I.

When tested on the more complex sigmoid-gated reward model, our proposed methods demonstrate a clear performance advantage, as illustrated in Figure 4. Both DLTS and DPSG-MP consistently outperform the strong NeuralTS and NeuralUCB baselines, achieving the lowest cumulative regret. This superior performance underscores the primary benefit of our framework. By leveraging a powerful, pretrained diffusion prior, our algorithms can explore the parameter space more effectively and adapt more quickly in challenging nonlinear environments. This stands in contrast to the baseline methods, which must learn from scratch on each new task. The plotted results, which show the mean cumulative regret and standard error averaged over 64 tasks across 8 independent trials, confirm this significant performance gain.

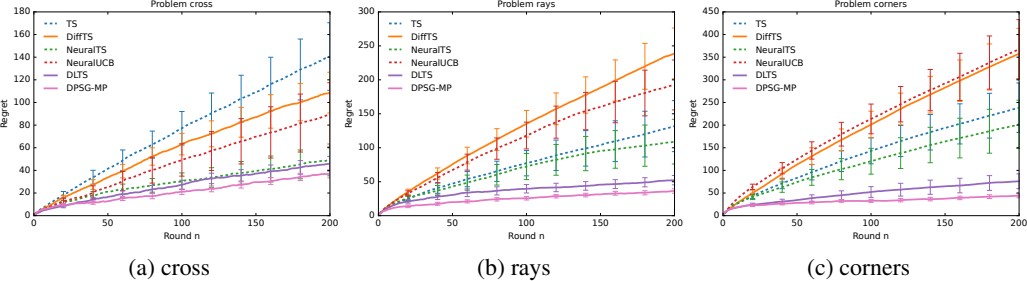

| (a) cross | (b) rays | (c) corners |

Figure 4: Performance of all algorithms on the sigmoid-gated nonlinear bandit tasks for the three prior distributions. Our proposed methods outperform strong baselines like NeuralTS and NeuralUCB, achieving the lowest cumulative regret. More results can be found in Appendix I.

Table 1: Results of ablation study on Diffusion Steps $L$. Reported values are Cumulative Regret at $T = 200$. The regrets are averaged over 64 tasks across 8 independent runs.

| Environment | Algorithm | Diffusion Steps ($L$) | | |
|---|---|---|---|---|
| | | $L = 20$ | $L = 50$ | $L = 100$ |
| Rays | DLTS | $136.08 \pm 3.18$ | $53.51 \pm 4.42$ | $\mathbf{51.20 \pm 2.90}$ |
| | DPSG-MP | $63.90 \pm 0.72$ | $50.28 \pm 1.09$ | $\mathbf{36.31 \pm 1.09}$ |
| Triangles | DLTS | $433.16 \pm 7.51$ | $97.68 \pm 5.95$ | $\mathbf{47.29 \pm 1.79}$ |
| | DPSG-MP | $86.57 \pm 0.66$ | $73.09 \pm 1.19$ | $\mathbf{52.20 \pm 3.54}$ |
| Swirl | DLTS | $732.70 \pm 7.29$ | $125.55 \pm 4.06$ | $\mathbf{53.95 \pm 1.74}$ |
| | DPSG-MP | $75.45 \pm 0.59$ | $52.47 \pm 0.51$ | $\mathbf{38.34 \pm 1.63}$ |

### 5.3 ABLATION STUDY AND TIME ANLYSIS

**Sensitivity to Inner-Loop Updates** $K_\ell$. To analyze the sensitivity of our algorithms, we conduct ablation studies on the number of diffusion steps $L$ and the number of inner-loop updates $K_\ell$ in Appendix I.2. These experiments are performed using the sigmoid-gated nonlinear bandit benchmark, a challenging setting from Section 5.2. We report the cumulative regret at $T = 200$, averaged over 64 tasks across 8 independent runs.

We examined the impact of the diffusion depth $L$ while keeping other parameters fixed. As shown in Table 1, increasing $L$ consistently improves performance. A larger number of diffusion steps allows for a more refined reverse generation process, resulting in more accurate posterior sampling and significantly lower regret for both DLTS and DPSG-MP.

**Wall-Clock Time Analysis**. To rigorously assess the trade-off between computational cost and sample efficiency, we perform a time-to-accuracy analysis rather than relying solely on per-round inference latency. This metric effectively normalizes for sample efficiency, demonstrating that a computationally more intensive algorithm (such as ours) can achieve superior overall time efficiency if it requires significantly fewer interactions to reach a high-quality policy compared to faster but less data-efficient baselines.

We benchmark all algorithms in the nonlinear sigmoid-gated bandit setting using the 'triangles' prior distribution. We conduct our DLTS and DPSG-MP with inner-loop $K_\ell$ to be 10 and 1 respectively. We define a target performance threshold of 0.5 average regret (calculated as cumulative regret divided by the number of rounds). For each algorithm, we measure the total wall-clock time required to achieve this threshold. A maximum time budget of 3000 seconds is enforced. For fair comparison, all evaluations are conducted on an Intel Xeon E5-2640 v4 CPU. The results, summarized in Table 2, show that our methods (DLTS and DPSG-MP) reach the performance threshold significantly faster than the baselines, despite having a higher per-step computational cost, due to their superior sample efficiency.

Table 2: Analysis of time to achieve the same average regret which is 0.5 in the sigmoid-gated nonlinear bandit with the Triangles problem in 64 task with 1 runs. The results show that our methods reach the performance threshold significantly faster than the baselines (NeuralTS and NeuralUCB).

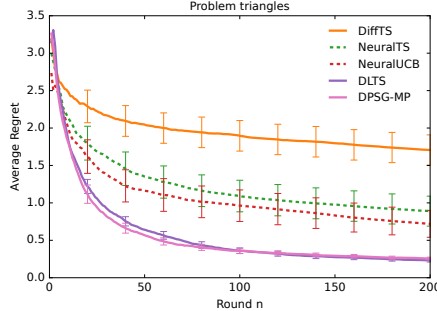

Figure 5: Results of average regret in Triangles with $T = 200$. It shows that our DLTS and DPSG-MP achieve the best performance.

| Algorithm | Round | Total Time (s) |
|---|---|---|
| DiffTS | $> 2000$ | $> 3000$ |
| NeuralTS | 1505 | $> 3000$ |
| NeuralUCB | 799 | $\sim 1967.71$ |
| **DLTS (Ours)** | 63 | $\sim \mathbf{1823}.81$ |
| **DPSG-MP (Ours)** | 54 | $\sim \mathbf{173}.63$ |

## 6 CONCLUSION

We address multi-task nonlinear contextual bandits by learning a flexible diffusion prior from past tasks and performing posterior sampling via a conditional reverse-diffusion process. We provide a unified update framework that, at each step, combines (i) an unconditional drift from the diffusion prior, (ii) a likelihood-driven drift from the interaction history, and (iii) a noise term enabling randomized exploration. Built on this view, we instantiate two variants: **DLTS**, which integrates history into the diffusion prior at every reverse step to form a conditional posterior and draw approximate samples; and **DPSG**, which first performs unconditional reverse sampling from the pretrained diffusion prior and then applies a single history-guided gradient correction. In theory part, we analyzed the connection and expected regret gap between Thompson Sampling and diffusion Thompson Sampling by formalizing corresponding oracle algorithms. In simulations across linear and non-linear contextual bandits, our methods are competitive with closed-form baselines in the linear regime (e.g., LinTS and DiffTS) and achieve consistent gains in neural settings. These demonstrate the potential of our algorithms to solve more realistic problems.

## REPRODUCIBILITY STATEMENT

Our code will be made available in the supplementary material and will be publicly released upon acceptance. The implementation of all our experiments is based on the PyTorch framework. Our simulation environment, particularly the design of the prior distributions, follows the benchmark described in Kveton et al. (2024).

We provide detailed pseudocode for our proposed algorithms, DLTS, DPSG, and DPSG-MP, in Section 3 in Algorithm 1, Algorithm 2 and Algorithm 3. We provide implementation details, including the network architectures for the diffusion models in Table 3 in Appendix H.1. Furthermore, all hyperparameters used for the nonlinear bandit settings are detailed in Table 4 and Table 5 in Appendix H.1.

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

## A  THE USE OF LARGE LANGUAGE MODELS (LLMs)

We utilized large language models (LLMs) to assist with language editing and improve the clarity and readability of the manuscript. The authors reviewed and revised all LLM-generated suggestions and take full responsibility for the final content of this paper. The LLM's role was strictly limited to editing and did not contribute to the core scientific ideas.

## B  RELATED WORK

**Diffusion Models**  Generative models have achieved strong results in modeling complex, multimodal distributions (Kingma & Welling, 2013; Goodfellow et al., 2014; Papamakarios et al., 2021; Ho et al., 2020). Among them, diffusion models are a good fit for inverse problems and decision-making because their reverse-time iterative sampling naturally supports conditioning on observed data (Sohl-Dickstein et al., 2015; Ho et al., 2020; Song et al., 2021). An inverse problem is to seek unknown parameters or signals from indirect, noisy observations using a forward measurement model. In this area, Chung et al. (2023) is a cornerstone: it samples from a diffusion-model posterior by augmenting the score with the observation likelihood. Follow-up work improves stability (Xu et al., 2025), image quality (Dou & Song, 2024), learning efficiency (Li & Wang, 2025), and offers principled formulations with theoretical guarantees that avoid inaccurate approximations (Wu et al., 2024). While most of these focus on imaging, they provide useful insight for sequential decision making. In offline settings, conditional diffusion has been used to synthesize trajectories with strong results on standard benchmarks (Janner et al., 2022; Ajay et al., 2023). In online settings, Hsieh et al. (2023) studied multi-armed bandits, learn a flexible diffusion prior over task parameters, and couple it with Thompson Sampling, including variance calibration for reliable uncertainty. Kveton et al. (2024) adapted diffusion priors to linear and generalized linear contextual bandits via closed-form reverse updates. Aouali (2024) studied contextual bandits with a linear diffusion-model prior but the model reduces to a linear Gaussian case rather than a general diffusion model under this assumption. However, multi-task nonlinear contextual bandits remain underexplored. The core challenge is the absence of closed-form reverse updates for nonlinear reward models, which complicates algorithm design. Without closed forms, sampling error can be large; coupled with limited interactions per task and possible inaccuracies in the learned diffusion prior, posterior sampling can become unstable and exploration can be misled.

**Randomized Exploration**  In sequential decision making problem such as bandits and reinforcement learning (RL), randomized exploration often outperforms deterministic strategies by preventing early convergence to suboptimal actions (Jin et al., 2021; 2023). Among such methods, Thompson Sampling (TS) (Thompson, 1933) is a key approach for multi-armed bandits (Agrawal & Goyal, 2017), contextual bandits (Agrawal & Goyal, 2013a), and RL (Osband et al., 2013). Unlike Upper Confidence Bound (UCB) algorithms, which rely on deterministic confidence intervals (Chu et al., 2011; Lattimore & Szepesvári, 2020), TS samples from posterior distributions, enabling robust exploration (Chapelle & Li, 2011). However, exact posterior sampling in TS is computationally intensive, particularly for non-conjugate priors. To address this, approximate sampling methods such as Langevin Monte Carlo (LMC) (Xu et al., 2022; Hsu et al., 2024), Stochastic gradient Langevin dynamics (SGLD) variants (Mazumdar et al., 2020; Zheng et al., 2024) and variational inference (Clavier et al., 2024) have been developed and applied to various problem settings, including multiarmed bandits with non-conjugate or highly nonlinear rewards, nonlinear contextual bandits and RL (Ishfaq et al., 2024a;b; Hsu et al., 2024). Perturb History Exploration (PHE) improves efficiency by perturbing historical data to approximate posterior sampling, making it applicable to complex reward distributions (Kveton et al., 2020; Ishfaq et al., 2021). Ensemble sampling keeps a small set of independently perturbed model replicas and selects actions using a randomly chosen replica (Lu & Van Roy, 2017; Qin et al., 2022; Janz et al., 2024; Lee & Oh, 2024).

# C MORE DETAILS ON DIFFUSION POSTERIOR SAMPLING ALGORITHMS

## C.1 DETAILED ALGORITHM INTERPRETATION FOR DLTS

We run DLTS for $T$ rounds to solve the new bandit task $(N+1)$. At each time $t$, with a pretrained diffusion model for unconditional sampling, we modify the reverse diffusion update to perform conditional sampling by injecting the history at every reverse step. We implement this conditional sampling by applying approximate sampling via Langevin Monte Carlo (LMC). After the reverse process ends, we take the final state $\boldsymbol{\theta}_0$ (i.e., $\widetilde{\boldsymbol{\theta}}_t$ ) as the parameter sample for arm selection.

Next, we provide a detailed derivation of the conditional sampling used in the reverse diffusion update in Algorithm 1. After training the diffusion denoiser $\varepsilon_{\boldsymbol{\phi}^*}(\cdot, \cdot)$ (see Appendix G for more details), we obtain the diffusion model parameter $(\boldsymbol{\mu}_\ell, \boldsymbol{\Sigma}_\ell)_{\ell \in [L+1]}{}^2$ for unconditional sampling in each reverse step, which is formulated as follow,

$$p(\boldsymbol{\theta}_{\ell-1}|\boldsymbol{\theta}_\ell) = \mathcal{N}(\boldsymbol{\theta}_{\ell-1}; \boldsymbol{\mu}_\ell, \boldsymbol{\Sigma}_\ell), \tag{C.1}$$

where $\boldsymbol{\mu}_\ell = \frac{1}{\sqrt{\alpha_\ell}}\big(\boldsymbol{\theta}_\ell - \frac{1-\alpha_\ell}{\sqrt{1-\bar{\alpha}_\ell}}\varepsilon_{\boldsymbol{\phi}^*}(\boldsymbol{\theta}_\ell, \ell)\big)$, $\boldsymbol{\Sigma}_\ell = \frac{1-\bar{\alpha}_{\ell-1}}{1-\bar{\alpha}_\ell}\beta_\ell \mathbf{I}$. Note that in (3.2), we can factorize $p(\mathcal{H}_t|\boldsymbol{\theta}_{\ell-1})$ as follows,

$$p(\mathcal{H}_t|\boldsymbol{\theta}_{\ell-1}) = \int_{\boldsymbol{\theta}_0} p(\mathcal{H}_t|\boldsymbol{\theta}_0, \boldsymbol{\theta}_{\ell-1})p(\boldsymbol{\theta}_0|\boldsymbol{\theta}_{\ell-1})d\boldsymbol{\theta}_0 = \int_{\boldsymbol{\theta}_0} p(\mathcal{H}_t|\boldsymbol{\theta}_0)p(\boldsymbol{\theta}_0|\boldsymbol{\theta}_{\ell-1})d\boldsymbol{\theta}_0,$$

where the second equality comes from that $\mathcal{H}_t$ and $\boldsymbol{\theta}_{\ell-1}$ are conditionally independent on $\boldsymbol{\theta}_0$. To derive the conditional diffusion reverse update, we need to approximate the following term $\int_{\boldsymbol{\theta}_0} p(\mathcal{H}_t|\boldsymbol{\theta}_0)p(\boldsymbol{\theta}_0|\boldsymbol{\theta}_{\ell-1})d\boldsymbol{\theta}_0$. Based on Kveton et al. (2024), we approximate it by

$$\int_{\boldsymbol{\theta}_0} p(\mathcal{H}_t|\boldsymbol{\theta}_0)p(\boldsymbol{\theta}_0|\boldsymbol{\theta}_{\ell-1})d\boldsymbol{\theta}_0 \approx p(\mathcal{H}_t|\boldsymbol{\theta}_{\ell-1}/\sqrt{\bar{\alpha}_{\ell-1}}). \tag{C.2}$$

The motivation of approximation in (C.2) is as follows. Note that $\boldsymbol{\theta}_\ell = \sqrt{\bar{\alpha}_\ell}\boldsymbol{\theta}_0 + \sqrt{1-\bar{\alpha}_\ell}\widetilde{\boldsymbol{\xi}}_\ell$, where $\widetilde{\boldsymbol{\xi}}_\ell \sim \mathcal{N}(\mathbf{0}, \mathbf{I})$ is standard Gaussian noise. Rearranging gives $\boldsymbol{\theta}_0 = (\boldsymbol{\theta}_\ell - \sqrt{1-\bar{\alpha}_\ell}\widetilde{\boldsymbol{\xi}}_\ell)/\sqrt{\bar{\alpha}_\ell}$ , so $\boldsymbol{\theta}_0$ can be viewed as a random variable with mean $\boldsymbol{\theta}_\ell/\sqrt{\bar{\alpha}_\ell}$. Replacing $\boldsymbol{\theta}_0$ in the LHS of (C.2) by this mean yields the approximation.

Note that at step $t$, given a pre-determined $f$, the nonlinear reward model is defined as

$$y_t = f(\boldsymbol{\theta}^*; \boldsymbol{x}_t) + \varepsilon_t.$$

When we assume Gaussian noise $\varepsilon_t \sim \mathcal{N}(0, \sigma^2)$, then the likelihood becomes

$$p(\mathcal{H}_t|\boldsymbol{\theta}) \propto \exp\left(-\sigma^{-2}\sum_{i=1}^t \big(f(\boldsymbol{\theta}; \boldsymbol{x}_i) - y_i\big)^2\right) = \exp(-L_t(\boldsymbol{\theta})), \tag{C.3}$$

where $L_t(\boldsymbol{\theta}) = \sigma^{-2}\sum_{i=1}^t (f(\boldsymbol{\theta}; \boldsymbol{x}_i) - y_i)^2$. Therefore, based on (3.2), (C.2) and (C.3), we approximate the posterior in each reverse step for conditional sampling as follows,

$$p(\boldsymbol{\theta}_{\ell-1}|\boldsymbol{\theta}_\ell, \mathcal{H}_t) \propto \exp\big(-L_t(\boldsymbol{\theta}_{\ell-1}/\sqrt{\bar{\alpha}_{\ell-1}})\big) \cdot \mathcal{N}(\boldsymbol{\theta}_{\ell-1}; \boldsymbol{\mu}_\ell, \boldsymbol{\Sigma}_\ell). \tag{C.4}$$

To sample from $p(\boldsymbol{\theta}_{\ell-1}|\boldsymbol{\theta}_\ell, \mathcal{H}_t)$, we apply Langevin Monte Carlo for approximate sampling. Specifically, at bandit round $t$ and diffusion reverse step $\ell$, we iteratively conduct the Langevin update,

$$\boldsymbol{\theta}_{\ell,k+1} = \boldsymbol{\theta}_{\ell,k} - \eta_\ell\Big[\nabla_{\boldsymbol{\theta}}L_t\big(\boldsymbol{\theta}_{\ell,k}/\sqrt{\bar{\alpha}_{\ell-1}}\big) + \boldsymbol{\Sigma}_\ell^{-1}\big(\boldsymbol{\theta}_{\ell,k} - \boldsymbol{\mu}_\ell\big)\Big] + \sqrt{2\eta_\ell\zeta_\ell^{-1}}\boldsymbol{\xi}_k, \tag{C.5}$$

where $\boldsymbol{\xi}_k \sim \mathcal{N}(\mathbf{0}, \mathbf{I})$, $\eta_\ell$ is step-size, $\zeta_\ell$ is temperature and (C.5) is initialized at $\boldsymbol{\theta}_{\ell,0} = \boldsymbol{\mu}_\ell$ and after $K_\ell$ iterations return $\boldsymbol{\theta}_{\ell,K_\ell}$ to be the diffusion reverse process sample $\boldsymbol{\theta}_{\ell-1}$.

Specially, when we only conduct one LMC update, based on (C.5), we have the reverse update

$$\boldsymbol{\theta}_{\ell-1} \leftarrow \left(1 - \frac{(1-\bar{\alpha}_\ell)\eta_\ell}{(1-\bar{\alpha}_{\ell-1})\beta_\ell}\right)\boldsymbol{\theta}_\ell + \frac{(1-\bar{\alpha}_\ell)\eta_\ell}{(1-\bar{\alpha}_{\ell-1})\beta_\ell}\boldsymbol{\mu}_\ell - \eta_\ell\nabla_{\boldsymbol{\theta}}L_t\big(\boldsymbol{\theta}_\ell/\sqrt{\bar{\alpha}_{\ell-1}}\big) + \sqrt{2\eta_\ell\zeta_\ell^{-1}}\boldsymbol{\xi}.$$

$$\tag{C.6}$$

Note that (C.6) aligns with our unified update (3.1). This confirms the rationality of our framework and the essence of diffusion-based posterior sampling. Intuitively, DLTS forms a conditional posterior by incorporating the history into the diffusion prior at each reverse step, then draws samples from it via approximate sampling.

---

$^2$We define $(\boldsymbol{\mu}_{L+1}, \boldsymbol{\Sigma}_{L+1}) = (\mathbf{0}, \mathbf{I})$ for notation simplicity.

## C.2 DETAILED ALGORITHM INTERPRETATION FOR DPSG

We run DPSG for $T$ rounds to solve the new bandit task $(N+1)$. At each time $t$, after using a pretrained diffusion model for unconditional sampling, we then turn the reverse diffusion into conditional sampling by adding a likelihood drift at every reverse step: at level $\ell$ we form the Tweedie proxy $\hat{\boldsymbol{\theta}}_0$ from the current state and apply a single gradient update inside the reverse transition. The history interaction data enter as one likelihood-score step per level. After the reverse process ends, we take the final state $\boldsymbol{\theta}_0$ (i.e., $\widetilde{\boldsymbol{\theta}}_t$) as the parameter sample for arm selection.

To derive diffusion reverse update (Line 8 in Algorithm 2), we follow Chung et al. (2023) and decompose the conditional score into two terms,

$$\nabla_{\boldsymbol{\theta}} \log p_s(\boldsymbol{\theta}_s|y) = \underbrace{\nabla_{\boldsymbol{\theta}} \log p_s(\boldsymbol{\theta}_s)}_{\text{unconditional score}} + \underbrace{\nabla_{\boldsymbol{\theta}} \log p_s(y|\boldsymbol{\theta}_s)}_{\text{likelihood score}}.$$

For the first term, we can use pretrained diffusion model score network $s_{\boldsymbol{\psi}^*}$ to approximate the unconditional score. For the second term, we can use Tweedie's formula to provide a tractable approximation for $p_s(y|\boldsymbol{\theta}_s)$ such that one can use the surrogate function for approximate posterior sampling, which is formulated as,

$$\nabla_{\boldsymbol{\theta}} \log p_s(y|\boldsymbol{\theta}_s) \simeq \nabla_{\boldsymbol{\theta}} \log p_s\big(y|\hat{\boldsymbol{\theta}}_0\big), \tag{C.7}$$

where $\boldsymbol{\theta}$ is the conditional posterior mean of $\boldsymbol{\theta}_0$, which have the following close-form expression derived by Tweedie's formula (Lemma F.2), that is,

$$\hat{\boldsymbol{\theta}}_0 = \mathbb{E}_{\boldsymbol{\theta}_0 \sim p(\boldsymbol{\theta}_0|\boldsymbol{\theta}_s)}[\boldsymbol{\theta}_0] = \frac{1}{\sqrt{\bar{\alpha}_s}}\big(\boldsymbol{\theta}_s + (1-\bar{\alpha}_s)\nabla_{\boldsymbol{\theta}} \log p_s(\boldsymbol{\theta}_s)\big).$$

In practice, given learned score network $s_{\boldsymbol{\psi}^*}$, we can further approximate the posterior mean as

$$\hat{\boldsymbol{\theta}}_0(\boldsymbol{\theta}_s, s) \simeq \frac{1}{\sqrt{\bar{\alpha}_s}}\big(\boldsymbol{\theta}_s + (1-\bar{\alpha}_s)s_{\boldsymbol{\psi}^*}(\boldsymbol{\theta}_s, s)\big). \tag{C.8}$$

At timestep $t$, the nonlinear reward model is defined as $y_t = f(\boldsymbol{\theta}^*; \boldsymbol{x}_t) + \varepsilon_t$, where $f$ is a pre-determined nonlinear function. Same as (C.3), when we assume $\varepsilon_t \sim \mathcal{N}(0, \sigma^2)$, then the likelihood becomes $p(\mathcal{H}_t|\boldsymbol{\theta}) \propto \exp(-\sigma^{-2}\sum_{i=1}^t (f(\boldsymbol{\theta}; \boldsymbol{x}_i) - y_i)^2) = \exp(-L_t(\boldsymbol{\theta}))$ where $L_t(\boldsymbol{\theta}) = \sigma^{-2}\sum_{i=1}^t (f(\boldsymbol{\theta}; \boldsymbol{x}_i) - y_i)^2$. Based on (C.7) and (C.8), then we can have the following tractable approximation of likelihood score,

$$\nabla_{\boldsymbol{\theta}} \log p_s(\mathcal{H}_t|\boldsymbol{\theta}_s) \simeq -\nabla_{\boldsymbol{\theta}} L_t\big(\hat{\boldsymbol{\theta}}_0(\boldsymbol{\theta}_s, s)\big).$$

Therefore, we can further have the approximation of the conditional score of reverse dynamics,

$$\nabla_{\boldsymbol{\theta}} \log p_s(\boldsymbol{\theta}_s|\mathcal{H}_t) \simeq s_{\boldsymbol{\psi}^*}(\boldsymbol{\theta}_s, s) - \nabla_{\boldsymbol{\theta}} L_t\big(\hat{\boldsymbol{\theta}}_0(\boldsymbol{\theta}_s, s)\big). \tag{C.9}$$

In practice, to obtain a discrete-time algorithm we use DDPM ancestral sampling (Ho et al., 2020) to implement conditional sampling via the conditional score (C.9). The reverse update is done by two steps. First, we use the unconditional score $s_{\boldsymbol{\psi}^*}(\boldsymbol{\theta}_\ell, \ell)$ through Tweedie's estimate $\hat{\boldsymbol{\theta}}_0(\boldsymbol{\theta}_\ell, \ell)$ to form the Gaussian posterior mean, we then sample

$$\boldsymbol{\theta}'_{\ell-1} \leftarrow \frac{\sqrt{\alpha_\ell}(1-\bar{\alpha}_{\ell-1})}{1-\bar{\alpha}_\ell}\boldsymbol{\theta}_\ell + \frac{\sqrt{\bar{\alpha}_{\ell-1}}\beta_\ell}{1-\bar{\alpha}_\ell}\hat{\boldsymbol{\theta}}_0(\boldsymbol{\theta}_\ell, \ell) + \boldsymbol{z}_\ell, \quad \boldsymbol{z}_\ell \sim \mathcal{N}(\mathbf{0}, \beta_\ell \mathbf{I}), \tag{C.10}$$

Based on (C.9), the second step is that we make this update conditional by subsequently adding one likelihood term. In practice, after sampling $\boldsymbol{\theta}'_{\ell-1}$, we take a gradient step using the likelihood $L_t$ at the Tweedie estimate for guidance, that is

$$\boldsymbol{\theta}_{\ell-1} \leftarrow \boldsymbol{\theta}'_{\ell-1} - \eta_\ell \nabla_{\boldsymbol{\theta}} L_t\big(\hat{\boldsymbol{\theta}}_0(\boldsymbol{\theta}_\ell, \ell)\big). \tag{C.11}$$

When we merge (C.10) and (C.11) together, we obtain line 8 in Algorithm 2,

$$\boldsymbol{\theta}_{\ell-1} \leftarrow \frac{1}{\sqrt{\alpha_\ell}}\boldsymbol{\theta}_\ell + \frac{\beta_\ell}{\sqrt{\alpha_\ell}}s_{\boldsymbol{\psi}^*}(\boldsymbol{\theta}_\ell, \ell) - \eta_\ell \nabla_{\boldsymbol{\theta}} L_t\big(\hat{\boldsymbol{\theta}}_0(\boldsymbol{\theta}_\ell, \ell)\big) + \boldsymbol{z}_\ell. \tag{C.12}$$

Note that (C.12) aligns with our unified update (3.1). This confirms the rationality of our framework and the essence of diffusion-based posterior sampling. However, from the intuition perspective, DPSG is to use history information as guidance after unconditional update based on the diffusion prior. This is different from DLTS, which directly integrates history information into the unconditional update based on the diffusion prior to achieve conditional sampling.

### C.3 ALGORITHM INTERPRETATION FOR DPSG-MP

DPSG is simple and fast, but it relies on the likelihood-score at the Tweedie estimate. A recent work Xu et al. (2025) observes that DPS has the properties of high bias and low diversity, thus behaving like an implicit, but unstable, MAP estimator. Therefore, multi-step projection is used to solve this MAP optimization problem. Xu et al. (2025) proposed Diffusion Maximize a Posterior (DMAP), which keeps the standard reverse diffusion prior step and replaces the single likelihood drift with several small gradient-ascent steps on the log posterior at each noise level, projecting after each step onto a sphere around the unconditional mean to stay consistent with the diffusion dynamics.

Therefore, we introduce DPSG with Multi-step Projection (DPSG-MP) shown in Algorithm 3. Compared to Algorithm 2, it replaces the single correction with a short inner loop of gradient ascent on the likelihood score, and after each step projects the iterate onto the sphere where the reverse transition $p_{\boldsymbol{\psi}^*}(\boldsymbol{\theta}_{\ell-1}|\boldsymbol{\theta}_\ell)$ concentrates around. This avoids the need for an accurate conditional score, reduces drift bias, and yields stable, data-consistent parameter estimates for arm selection.

---

**Algorithm 3** DPSG with Multi-step Projection (DPSG-MP)

---

**Input:** score network $s_{\boldsymbol{\psi}^*}(\cdot, \cdot)$, noise schedule $\{\beta_\ell\}$, trajectory history $\mathcal{H}_t$, learning rate $\{\eta_{\ell,k}\}$.

1: $\alpha_\ell \leftarrow 1 - \beta_\ell$ and $\bar{\alpha}_\ell \leftarrow \prod_{j=1}^{\ell} \alpha_j$ for all $\ell$.
2: **for** $t = 1, 2, \cdots, T$ **do**
3:     Receive contextual vector $\{\boldsymbol{x}_t(a)\}_{a \in \mathcal{A}}$.
4:     $\boldsymbol{\theta}_L \sim \mathcal{N}(\mathbf{0}, \mathbf{I})$.
5:     **for** $\ell = L, L-1, \cdots, 1$ **do**
6:         $\hat{\boldsymbol{\theta}}_0(\boldsymbol{\theta}_\ell, \ell) \leftarrow \frac{1}{\sqrt{\bar{\alpha}_\ell}} \big( \boldsymbol{\theta}_\ell + (1 - \bar{\alpha}_\ell) s_{\boldsymbol{\psi}^*}(\boldsymbol{\theta}_\ell, \ell) \big)$.
7:         $\boldsymbol{z}_\ell \sim \mathcal{N}(\mathbf{0}, \beta_\ell \mathbf{I})$, $r_\ell \leftarrow \|\boldsymbol{z}_\ell\|_2$.
8:         $\boldsymbol{m}_\ell \leftarrow \frac{\sqrt{\alpha_\ell}(1 - \bar{\alpha}_{\ell-1})}{1 - \bar{\alpha}_\ell} \boldsymbol{\theta}_\ell + \frac{\sqrt{\bar{\alpha}_{\ell-1}} \beta_\ell}{1 - \bar{\alpha}_\ell} \hat{\boldsymbol{\theta}}_0(\boldsymbol{\theta}_\ell, \ell)$.
9:         $\boldsymbol{\theta}_{\ell-1,0} \leftarrow \boldsymbol{m}_\ell + \boldsymbol{z}_\ell$.
10:       **for** $k = 1, 2, \cdots, K_\ell$ **do**
11:           $\boldsymbol{\theta}'_{\ell-1,k} \leftarrow \boldsymbol{\theta}_{\ell-1,k-1} - \eta_{\ell,k} \nabla_{\boldsymbol{\theta}} L_t\big(\hat{\boldsymbol{\theta}}_0(\boldsymbol{\theta}_\ell, \ell)\big)$.
12:           $\boldsymbol{\theta}_{\ell-1,k} \leftarrow \boldsymbol{m}_\ell + r_\ell \frac{\boldsymbol{\theta}'_{\ell-1,k} - \boldsymbol{m}_\ell}{\|\boldsymbol{\theta}'_{\ell-1,k} - \boldsymbol{m}_\ell\|_2}$.
13:       **end for**
14:       $\boldsymbol{\theta}_{\ell-1} \leftarrow \boldsymbol{\theta}_{\ell-1,K_\ell}$.
15:     **end for**
16:     Let $\widetilde{\boldsymbol{\theta}}_t \leftarrow \boldsymbol{\theta}_0$, take action $\boldsymbol{x}_t = \text{argmax}_{\boldsymbol{x}_t(a)} f\big(\widetilde{\boldsymbol{\theta}}_t; \boldsymbol{x}_t(a)\big)$.
17:     Receive reward $y_t$ and update history $\mathcal{H}_{t+1} = \{(\boldsymbol{x}_i, y_i)\}_{i=1}^t$.
18: **end for**

---

## D PROOF IN SECTION 4

### D.1 PROOF OF THEOREM 4.4

*Proof of Theorem 4.4.* We show that, under the ideal ODTS assumptions, the ODTS marginal posterior over the clean parameter is same as the OTS posterior.

Let the diffusion-path variable be $\boldsymbol{\theta}_{0:L} = (\boldsymbol{\theta}_0, \boldsymbol{\theta}_1, \ldots, \boldsymbol{\theta}_L)$, where $\boldsymbol{\theta}_0$ is the clean parameter used for decisions and $\boldsymbol{\theta}_L$ is the noisiest level. The diffusion prior has the the Markov factorization

$$p(\boldsymbol{\theta}_{0:L}) = p(\boldsymbol{\theta}_L) \prod_{\ell=1}^{L} p(\boldsymbol{\theta}_{\ell-1}|\boldsymbol{\theta}_\ell), \tag{D.1}$$

so that its marginal posterior is as follows,

$$p_0(\boldsymbol{\theta}_0) = \int p(\boldsymbol{\theta}_{0:L}) \, d\boldsymbol{\theta}_{1:L}, \tag{D.2}$$

which is the true prior used by OTS according to the condition. Note that the history $\mathcal{H}_t$ depends only on $\boldsymbol{\theta}_0$, then we have

$$p(\mathcal{H}_t|\boldsymbol{\theta}_{0:L}) = p(\mathcal{H}_t|\boldsymbol{\theta}_0). \tag{D.3}$$

ODTS is assumed ideal: (i) the learned score/denoiser equals the true score so that (D.1) holds for the implemented prior, and (ii) each per-level conditional reverse transition kernel is implemented exactly, thus representing the true joint posterior over the path.

By Bayes' rule and (D.3), the joint posterior over the path satisfies

$$\widetilde{p}_t(\boldsymbol{\theta}_{0:L}) \equiv p(\boldsymbol{\theta}_{0:L}|\mathcal{H}_t) \propto p(\mathcal{H}_t|\boldsymbol{\theta}_0)p(\boldsymbol{\theta}_{0:L}). \tag{D.4}$$

Integrating (D.4) over the noisy layers $\boldsymbol{\theta}_{1:L}$ and using (D.2) gives

$$
\begin{aligned}
\int p(\boldsymbol{\theta}_{0:L}|\mathcal{H}_t)d\boldsymbol{\theta}_{1:L} &= \frac{1}{p(\mathcal{H}_t)} \int p(\mathcal{H}_t|\boldsymbol{\theta}_0)p(\boldsymbol{\theta}_{0:L})d\boldsymbol{\theta}_{1:L} \\
&= \frac{1}{p(\mathcal{H}_t)}p(\mathcal{H}_t|\boldsymbol{\theta}_0)\underbrace{\int p(\boldsymbol{\theta}_{0:L})d\boldsymbol{\theta}_{1:L}}_{=p_0(\boldsymbol{\theta}_0)} \\
&= \frac{p(\mathcal{H}_t|\boldsymbol{\theta}_0)p_0(\boldsymbol{\theta}_0)}{p(\mathcal{H}_t)} = p(\boldsymbol{\theta}_0|\mathcal{H}_t). \tag{D.5}
\end{aligned}
$$

Note that $\widetilde{p}_t(\boldsymbol{\theta}) = \int \widetilde{p}_t(\boldsymbol{\theta}_{0:L})d\boldsymbol{\theta}_{1:L} = \int p(\boldsymbol{\theta}_{0:L}|\mathcal{H}_t)d\boldsymbol{\theta}_{1:L}$, then we have

$$\widetilde{p}_t(\boldsymbol{\theta}) = p(\boldsymbol{\theta}|\mathcal{H}_t) \equiv p_t(\boldsymbol{\theta}). \tag{D.6}$$

Therefore the ODTS posterior marginal $\widetilde{p}_t(\boldsymbol{\theta})$ and the OTS posterior $p_t(\boldsymbol{\theta})$ are same. Note that both oracle algorithms use greedy policy to select arm, thus share the same arm selection distribution. $\qquad\square$

### D.2  PROOF OF THEOREM 4.5

*Proof of Theorem 4.5.* Based on the definition of cumulative expected regret, we focus on the per-round expected regret in round $t$,

$$r_t = f(\boldsymbol{\theta}^*; \boldsymbol{x}^*) - f(\boldsymbol{\theta}^*; \boldsymbol{x}_t).$$

We then focus on the regret difference between ODTS and OTS,

$$
\begin{aligned}
\mathbb{E}\big[r_t^{\text{ODTS}} - r_t^{\text{OTS}}\big] &= \mathbb{E}\big[f(\boldsymbol{\theta}^*; \boldsymbol{x}^*) - f(\boldsymbol{\theta}^*; \boldsymbol{x}_t^{\text{ODTS}})\big] - \mathbb{E}\big[f(\boldsymbol{\theta}^*; \boldsymbol{x}^*) - f(\boldsymbol{\theta}^*; \boldsymbol{x}_t^{\text{OTS}})\big] \\
&= \mathbb{E}_{\boldsymbol{\theta} \sim p_t}\big[f(\boldsymbol{\theta}^*; \pi(\boldsymbol{\theta}))\big] - \mathbb{E}_{\boldsymbol{\theta} \sim \widetilde{p}_t}\big[f(\boldsymbol{\theta}^*; \pi(\boldsymbol{\theta}))\big] \\
&\leq 2f_{\max}\text{TV}(p_t, \widetilde{p}_t),
\end{aligned}
$$

where the inequality comes from the dual definition of TV divergence. $\text{TV}(p_t, \widetilde{p}_t)$ contains the score estimation error, we then further bound this term based on Assumption 4.3.

**Bounding $\text{TV}(p_t, \widetilde{p}_t)$.** At reverse level $\ell$, the true and learned reverse Gaussian transition share covariance $\boldsymbol{\Sigma}_\ell$ while their means differ because of the accuracy of denoiser. Denote that $\mu_\ell^*$ and $\varepsilon^*$ is the mean and noise of the true transition, then we have

$$\|\boldsymbol{\mu}_\ell - \boldsymbol{\mu}_\ell^*\|_2 \leq \frac{1 - \alpha_\ell}{\sqrt{\alpha_\ell \bar{\alpha}_\ell}}\|\varepsilon_{\boldsymbol{\phi}^*} - \varepsilon^*\|_2 = c_\ell\big\|s_{\boldsymbol{\psi}^*}(\boldsymbol{\theta}_\ell, \ell) - \nabla_{\boldsymbol{\theta}}\log p_\ell(\boldsymbol{\theta}_\ell)\big\|_2 \leq c_\ell\epsilon_{\text{score}},$$

where coefficient $c_\ell$ is determined by noise schedule, the second equality holds due to Tweedie's formula, the last inequality holds because of Assumption 4.3. For Gaussian transition with equal covariance, we have

$$\text{TV}\big(\mathcal{N}(\boldsymbol{\mu}_\ell^*, \boldsymbol{\Sigma}_\ell), \mathcal{N}(\boldsymbol{\mu}_\ell, \boldsymbol{\Sigma}_\ell)\big) \leq \kappa_\ell\epsilon_{\text{score}},$$

with $\kappa_\ell = \frac{c_\ell}{2\sqrt{\lambda_{\min}(\boldsymbol{\Sigma}_\ell)}} = \frac{1 - \alpha_\ell}{2\sqrt{\alpha_\ell}\sqrt{\lambda_{\min}(\boldsymbol{\Sigma}_\ell)}}$. Propagating this perturbation through the reverse Markov chain yields

$$\text{TV}(p_t, \widetilde{p}_t) \leq \sum_{\ell=1}^{L} \kappa_\ell\epsilon_{\text{score}}.$$

Therefore, we have the score estimation error,

$$\Delta_t^{\text{Score}} = 2f_{\max}\sum_{\ell=1}^{L} \kappa_\ell\epsilon_{\text{score}}.$$

This completes the proof. $\qquad\square$

# E  INSTANTIATION UNDER SPECIFIC SETTINGS

In this section, we describe the bandits settings where we instantiate our proposed algorithms. The general frameworks for DLTS and DPSG can be readily adapted to various specific models by defining the reward function $f(\boldsymbol{\theta}; \boldsymbol{x})$ and the corresponding loss function $L_t(\boldsymbol{\theta})$. The gradient of this loss, $\nabla_{\boldsymbol{\theta}} L_t(\boldsymbol{\theta})$, is the crucial model-dependent component that is plugged into the reverse diffusion update rules for both DLTS and DPSG. We now provide the concrete formulations for three widely-studied settings: linear, generalized linear, and neural contextual bandits.

**Linear Bandit**  We first consider the linear contextual bandit settings. In linear contextual bandits, the reward is given by a linear function of the context and an unknown parameter $\boldsymbol{\theta}^* \in \mathbb{R}^d$ plus some noise, that is, $y_t = \boldsymbol{\theta}^{*\top} \boldsymbol{x}_t + \varepsilon_t$, where $\boldsymbol{x}_t \in \mathbb{R}^d$ is the context at time $t$ and $\varepsilon_t$ is the noise term. We assume that $\varepsilon_t$ is i.i.d. Gaussian noise with mean zero and variance $\sigma^2$. The goal of the agent is to learn the unknown parameter $\boldsymbol{\theta}^*$ and select actions that maximize the cumulative reward over time. The reward function is defined as follows

$$f(\boldsymbol{\theta}; \boldsymbol{x}) = \boldsymbol{\theta}^{\top} \boldsymbol{x}.$$

In our proposed algorithms, we define the loss function over the observed history $\mathcal{H}_t$ as follows

$$L_t(\boldsymbol{\theta}) = \sigma^{-2} \sum_{i=1}^{t} (\boldsymbol{\theta}^{\top} \boldsymbol{x}_i - y_i)^2,$$

where $\sigma^2$ is the variance, $\boldsymbol{x}_i$ is the context at time $i$ and $y_i$ is the corresponding reward. For the linear model, the crucial gradient term $\nabla_{\boldsymbol{\theta}} L_t(\boldsymbol{\theta})$ required for our diffusion updates has a convenient closed-form expression. It can be computed as:

$$\nabla_{\boldsymbol{\theta}} L_t(\boldsymbol{\theta}) = 2\sigma^{-2} \sum_{i=1}^{t} (\boldsymbol{\theta}^{\top} \boldsymbol{x}_i - y_i) \boldsymbol{x}_i.$$

For DLTS, the Langevin Monte Carlo update from (C.6), which is iterated within each reverse diffusion step $\ell$. We have

$$\boldsymbol{\theta}_{\ell-1} \leftarrow \left(1 - \frac{(1-\bar{\alpha}_\ell)\eta_\ell}{(1-\bar{\alpha}_{\ell-1})\beta_\ell}\right)\boldsymbol{\theta}_\ell + \frac{(1-\bar{\alpha}_\ell)\eta_\ell}{(1-\bar{\alpha}_{\ell-1})\beta_\ell}\boldsymbol{\mu}_\ell - 2\eta_\ell\sigma^{-2}\sum_{i=1}^{t}\left(\frac{\boldsymbol{\theta}_\ell^{\top}\boldsymbol{x}_i}{\sqrt{\bar{\alpha}_{\ell-1}}} - y_i\right)\frac{\boldsymbol{x}_i}{\sqrt{\bar{\alpha}_{\ell-1}}} + \sqrt{2\eta_\ell\zeta_\ell^{-1}}\boldsymbol{\xi}.$$

For DPSG, the reverse update from Line 8 in Algorithm 2 is instantiated as:

$$\boldsymbol{\theta}_{\ell-1} \leftarrow \frac{1}{\sqrt{\alpha_\ell}}\boldsymbol{\theta}_\ell + \frac{\beta_\ell}{\sqrt{\alpha_\ell}}s_{\boldsymbol{\psi}^*}(\boldsymbol{\theta}_\ell, \ell) - 2\eta_\ell\sigma^{-2}\sum_{i=1}^{t}\left((\hat{\boldsymbol{\theta}}_0(\boldsymbol{\theta}_\ell, \ell))^{\top}\boldsymbol{x}_i - y_i\right)\boldsymbol{x}_i + \boldsymbol{z}_\ell.$$

**Generalized Linear Bandit**  Then we extend our proposed algorithms to generalized linear bandit settings. In generalized linear bandits, the reward is given by a generalized linear function of the context and an unknown parameter $\boldsymbol{\theta}^* \in \mathbb{R}^d$ plus some noise, that is, $y_t = g(\boldsymbol{\theta}^{*\top}\boldsymbol{x}_t) + \varepsilon_t$, where $\boldsymbol{x}_t \in \mathbb{R}^d$ is the context at time $t$, $g$ is a known link function and $\varepsilon_t$ is the noise term. We assume that $\varepsilon_t$ is i.i.d. Gaussian noise with mean zero and variance $\sigma^2$. The link function $g$ is a known function that maps the linear combination of the context and parameter to the expected reward. We have the following formulations for the reward function

$$f(\boldsymbol{\theta}; \boldsymbol{x}) = g(\boldsymbol{\theta}^{\top}\boldsymbol{x}).$$

The goal of the agent is to learn the unknown parameter $\boldsymbol{\theta}^*$ and select actions that maximize the cumulative reward over time. In our proposed algorithms, we define the loss function over the observed history $\mathcal{H}_t$ as follows

$$L_t(\boldsymbol{\theta}) = \sigma^{-2} \sum_{i=1}^{t} (g(\boldsymbol{\theta}^{\top}\boldsymbol{x}_i) - y_i)^2,$$

where $\sigma^2$ is the variance, $\boldsymbol{x}_i$ is the context at time $i$ and $y_i$ is the corresponding reward. We use the logistic function as the example in our experiments. The logistic function is defined as $g(z) = 1/(1 + e^{-z})$. For the gradient, we have:

$$\nabla_{\boldsymbol{\theta}} L_t(\boldsymbol{\theta}) = 2\sigma^{-2} \sum_{i=1}^{t} (g(\boldsymbol{\theta}^{\top} \boldsymbol{x}_i) - y_i) g'(\boldsymbol{\theta}^{\top} \boldsymbol{x}_i) \boldsymbol{x}_i,$$

where $g'(\cdot)$ is the derivative of the link function with respect to its input.

For DLTS, the single-step Langevin update (C.6) becomes:

$$\boldsymbol{\theta}_{\ell-1} \leftarrow \left(1 - \frac{(1 - \bar{\alpha}_{\ell})\eta_{\ell}}{(1 - \bar{\alpha}_{\ell-1})\beta_{\ell}}\right) \boldsymbol{\theta}_{\ell} + \frac{(1 - \bar{\alpha}_{\ell})\eta_{\ell}}{(1 - \bar{\alpha}_{\ell-1})\beta_{\ell}} \boldsymbol{\mu}_{\ell}$$

$$- 2\eta_{\ell}\sigma^{-2} \sum_{i=1}^{t} \left(g\left(\frac{\boldsymbol{\theta}_{\ell}^{\top} \boldsymbol{x}_i}{\sqrt{\bar{\alpha}_{\ell-1}}}\right) - y_i\right) g'\left(\frac{\boldsymbol{\theta}_{\ell}^{\top} \boldsymbol{x}_i}{\sqrt{\bar{\alpha}_{\ell-1}}}\right) \boldsymbol{x}_i + \sqrt{2\eta_{\ell}\zeta_{\ell}^{-1}} \boldsymbol{\xi}.$$

For DPSG, the reverse update (Line 8 in Algorithm 2) is instantiated as:

$$\boldsymbol{\theta}_{\ell-1} \leftarrow \frac{1}{\sqrt{\alpha_{\ell}}} \boldsymbol{\theta}_{\ell} + \frac{\beta_{\ell}}{\sqrt{\alpha_{\ell}}} s_{\boldsymbol{\psi}^*}(\boldsymbol{\theta}_{\ell}, \ell)$$

$$- 2\eta_{\ell}\sigma^{-2} \sum_{i=1}^{t} \left(g((\hat{\boldsymbol{\theta}}_0(\boldsymbol{\theta}_{\ell}, \ell))^{\top} \boldsymbol{x}_i) - y_i\right) g'\left((\hat{\boldsymbol{\theta}}_0(\boldsymbol{\theta}_{\ell}, \ell))^{\top} \boldsymbol{x}_i\right) \boldsymbol{x}_i + \boldsymbol{z}_{\ell}.$$

**Neural Bandit** Finally, we consider the neural contextual bandit settings. In neural contextual bandits, the reward is given by a neural network function of the context and an unknown parameter $\boldsymbol{\theta}^* \in \mathbb{R}^p$ plus some noise, that is, $y_t = f(\boldsymbol{\theta}^*; \boldsymbol{x}_t) + \varepsilon_t$, where $\boldsymbol{x}_t \in \mathbb{R}^d$ is the context at time $t$ and $\varepsilon_t$ is the noise term. We assume that $\varepsilon_t$ is i.i.d. Gaussian noise with mean zero and variance $\sigma^2$. The goal of the agent is to learn the unknown parameter $\boldsymbol{\theta}^*$ and select actions that maximize the cumulative reward over time. The reward function is defined as follows

$$f(\boldsymbol{\theta}; \boldsymbol{x}) = \sqrt{m} \mathbf{W}_L \sigma(\mathbf{W}_{L-1}\sigma(\cdots \sigma(\mathbf{W}_1 \mathbf{x}))),$$

where $\sigma(x) = \max\{x, 0\}$ is the rectified linear unit (ReLU) activation function, $\mathbf{W}_1 \in \mathbb{R}^{m \times d}$, $\mathbf{W}_i \in \mathbb{R}^{m \times m}, 2 \leq i \leq L - 1, \mathbf{W}_L \in \mathbb{R}^{m \times 1}$, and $\boldsymbol{\theta} = \left[\text{vec}(\mathbf{W}_1)^{\top}, \ldots, \text{vec}(\mathbf{W}_L)^{\top}\right]^{\top} \in \mathbb{R}^p$ with $p = m + md + m^2(L - 1)$. In our proposed algorithms, we define the loss function over the observed history $\mathcal{H}_t$ as follows

$$L_t(\boldsymbol{\theta}) = \sigma^{-2} \sum_{i=1}^{t} (f(\boldsymbol{\theta}; \boldsymbol{x}_i) - y_i)^2,$$

where $\sigma^2$ is the variance, $\boldsymbol{x}_i$ is the context at time $i$ and $y_i$ is the corresponding reward. The gradient of this loss function is required for our diffusion updates. We obtain:

$$\nabla_{\boldsymbol{\theta}} L_t(\boldsymbol{\theta}) = 2\sigma^{-2} \sum_{i=1}^{t} (f(\boldsymbol{\theta}; \boldsymbol{x}_i) - y_i) \nabla_{\boldsymbol{\theta}} f(\boldsymbol{\theta}; \boldsymbol{x}_i).$$

For DLTS, the single-step Langevin update (C.6) becomes:

$$\boldsymbol{\theta}_{\ell-1} \leftarrow \left(1 - \frac{(1 - \bar{\alpha}_{\ell})\eta_{\ell}}{(1 - \bar{\alpha}_{\ell-1})\beta_{\ell}}\right) \boldsymbol{\theta}_{\ell} + \frac{(1 - \bar{\alpha}_{\ell})\eta_{\ell}}{(1 - \bar{\alpha}_{\ell-1})\beta_{\ell}} \boldsymbol{\mu}_{\ell}$$

$$- 2\eta_{\ell}\sigma^{-2} \sum_{i=1}^{t} \left(f\left(\frac{\boldsymbol{\theta}_{\ell}}{\sqrt{\bar{\alpha}_{\ell-1}}}; \boldsymbol{x}_i\right) - y_i\right) \nabla_{\boldsymbol{\theta}} f\left(\frac{\boldsymbol{\theta}_{\ell}}{\sqrt{\bar{\alpha}_{\ell-1}}}; \boldsymbol{x}_i\right) + \sqrt{2\eta_{\ell}\zeta_{\ell}^{-1}} \boldsymbol{\xi}.$$

For DPSG, the reverse update (Line 8 in Algorithm 2) is instantiated as:

$$\boldsymbol{\theta}_{\ell-1} \leftarrow \frac{1}{\sqrt{\alpha_{\ell}}} \boldsymbol{\theta}_{\ell} + \frac{\beta_{\ell}}{\sqrt{\alpha_{\ell}}} s_{\boldsymbol{\psi}^*}(\boldsymbol{\theta}_{\ell}, \ell) - 2\eta_{\ell}\sigma^{-2} \sum_{i=1}^{t} \left(f(\hat{\boldsymbol{\theta}}_0(\boldsymbol{\theta}_{\ell}, \ell); \boldsymbol{x}_i) - y_i\right) \nabla_{\boldsymbol{\theta}} f(\hat{\boldsymbol{\theta}}_0(\boldsymbol{\theta}_{\ell}, \ell); \boldsymbol{x}_i) + \boldsymbol{z}_{\ell}.$$

# F    AUXILIARY LEMMAS

The following lemma provides the conditional probability distributions for diffusion reverse process based on history $\mathcal{H}_t = \{(\boldsymbol{x}_i, y_i)\}_{i=1}^{t-1}$ at timestep $t$.

**Lemma F.1** ((Kveton et al., 2024)). *For probability measure $p$ over the reverse process, we have*

$$p(\boldsymbol{\theta}_L | \mathcal{H}_t) \propto \left( \int_{\boldsymbol{\theta}_0} p(\mathcal{H}_t | \boldsymbol{\theta}_0) p(\boldsymbol{\theta}_0 | \boldsymbol{\theta}_L) d\boldsymbol{\theta}_0 \right) \cdot p(\boldsymbol{\theta}_L),$$

$$p(\boldsymbol{\theta}_{\ell-1} | \boldsymbol{\theta}_\ell, \mathcal{H}_t) \propto \left( \int_{\boldsymbol{\theta}_0} p(\mathcal{H}_t | \boldsymbol{\theta}_0) p(\boldsymbol{\theta}_0 | \boldsymbol{\theta}_{\ell-1}) d\boldsymbol{\theta}_0 \right) \cdot p(\boldsymbol{\theta}_{\ell-1} | \boldsymbol{\theta}_\ell), \quad for\ \ell = 2, \ldots, L,$$

$$p(\boldsymbol{\theta}_0 | \boldsymbol{\theta}_1, \mathcal{H}_t) \propto p(\mathcal{H}_t | \boldsymbol{\theta}_0) \cdot p(\boldsymbol{\theta}_0 | \boldsymbol{\theta}_1).$$

**Lemma F.2** (Tweedie's formula (Efron, 2011)). *Let $p(\boldsymbol{y}|\boldsymbol{\eta})$ belong to the exponential family distribution $p(\boldsymbol{y}|\boldsymbol{\eta}) = p_0(\boldsymbol{y}) \exp\left(\boldsymbol{\eta}^\top T(\boldsymbol{y}) - \varphi(\boldsymbol{\eta})\right)$, where $\boldsymbol{\eta}$ is the canonical vector of the family, $T(\boldsymbol{y})$ is some function of $\boldsymbol{y}$, and $\varphi(\boldsymbol{\eta})$ is the cumulant generation function which normalizes the density, and $p_0(\boldsymbol{y})$ is the density up to the scale factor when $\boldsymbol{\eta} = \boldsymbol{0}$. Then, the posterior mean $\hat{\boldsymbol{\eta}} := \mathbb{E}[\boldsymbol{\eta}|\boldsymbol{y}]$ should satisfy*

$$\left(\nabla_{\boldsymbol{y}} T(\boldsymbol{y})\right)^\top \hat{\boldsymbol{\eta}} = \nabla_{\boldsymbol{y}} \log p(\boldsymbol{y}) - \nabla_{\boldsymbol{y}} \log p_0(\boldsymbol{y}).$$

# G    DIFFUSION MODEL TRAINING

In this section, we introduce the training of diffusion model for bandit with nonlinear reward model, which is shown in Algorithm 4. The training process is a standard DDPM training process (Ho et al., 2020).

---

**Algorithm 4** Diffusion Model Training

---

**Input:** Training dataset $\mathcal{D} = \{\boldsymbol{\theta}^{(i)}\}_{i=1}^N$, total diffusion steps $L$, noise schedule $\{\beta_\ell\}_{\ell=1}^L$, learning rate $\eta$, mini–batch size $B$, epochs $E$.

1: $\alpha_\ell \leftarrow 1 - \beta_\ell$ and $\bar{\alpha}_\ell \leftarrow \prod_{j=1}^\ell \alpha_j$ for all $\ell$.
2: **Initialization:** denoiser parameters $\phi \leftarrow$ `randn()`.
3: **for** epoch $= 1, \ldots, E$ **do**
4:     **for all** mini–batches $\mathcal{B} \subset \mathcal{D}$ of size $B$ **do**
5:         Sample time indices $\ell_1, \ldots, \ell_B \sim \mathrm{Unif}\{1, \ldots, L\}$.
6:         Sample noises $\varepsilon_1, \ldots, \varepsilon_B \sim \mathcal{N}(\boldsymbol{0}, \mathbf{I})$.
7:         **for** $b = 1, \ldots, B$ **do**
8:             $X_0^{(b)} \leftarrow \boldsymbol{\theta}^{(b)}$.
9:             $X_{\ell_b}^{(b)} \leftarrow \sqrt{\bar{\alpha}_{\ell_b}} X_0^{(b)} + \sqrt{1 - \bar{\alpha}_{\ell_b}} \varepsilon_b.$          {forward diffusion}
10:            $\hat{\varepsilon}_b \leftarrow \varepsilon_\psi\left(X_{\ell_b}^{(b)}, \ell_b\right).$          {network prediction}
11:        **end for**
12:        $\mathcal{L} \leftarrow \dfrac{1}{B} \sum_{b=1}^B \|\hat{\varepsilon}_b - \varepsilon_b\|_2^2.$
13:        **Gradient decent:** $\phi \leftarrow \phi - \eta \nabla_\phi \mathcal{L}.$
14:    **end for**
15: **end for**

**Output:** Trained diffusion model denoiser $\varepsilon_{\phi^*}(\cdot, \cdot)$.

---

# H    EXPERIMENT DETAILS

In this section, we provide details about our experiment setups and implementations.

## H.1 Implement Details

**Diffusion Prior Implementation** Our diffusion prior is implemented within a Denoising Diffusion Probabilistic Model (DDPM) framework, using the standard $\epsilon$-prediction parameterization. The core of this model is a denoiser network responsible for predicting the noise at each diffusion step. We implement this denoiser using a Multi-Layer Perceptron (MLP) conditioned on a sinusoidal time embedding. The key architectural details of the MLP used at each diffusion step are summarized in Table 3.

Table 3: MLP Denoiser Configuration.

| Parameter | Value |
|-----------|-------|
| Hidden Layers | (32, 32) |
| Activation | ReLU |
| Optimizer | Adam |

**Hyperparameters** We provide the hyperparameters for our proposed algorithm in Table 4 and Table 5.

Table 4: Hyperparameters of DLTS

| Reward Model | Prior Distribution | Update Steps | Step size | Noise Scale |
|--------------|--------------------|--------------|-----------|-------------|
| Cosine | cross | 1 | 0.05 | 0.005 |
| | rays | 10 | 0.01 | 0.005 |
| | triangles | 1 | 0.005 | 0.05 |
| | swirl | 1 | 0.01 | 0.01 |
| | H | 1 | 0.05 | 0.01 |
| | corners | 1 | 0.05 | 0.05 |
| Quadratic | cross | 1 | 0.01 | 0.005 |
| | rays | 10 | 0.05 | 0.005 |
| | triangles | 10 | 0.05 | 0.01 |
| | swirl | 1 | 0.05 | 0.005 |
| | H | 1 | 0.005 | 0.1 |
| | corners | 10 | 0.01 | 0.05 |
| Sigmoid-gated | cross | 1 | 0.05 | 0.05 |
| | rays | 1 | 0.05 | 0.05 |
| | triangles | 10 | 0.05 | 0.005 |
| | swirl | 10 | 0.01 | 0.1 |
| | H | 1 | 0.1 | 0.01 |
| | corners | 1 | 0.05 | 0.1 |

## H.2 Prior Distribution

The 'cross' problem is generated from a mixture of two highly correlated 2D Gaussian distributions, creating a distinct cross shape. The 'rays' distribution is formed using rejection sampling, where accepted points are constrained to lie close to the four cardinal axes, resembling rays emanating from the origin. The 'triangles' distribution, also generated via rejection sampling, consists of points within two triangular regions. The 'swirl' distribution is generated parametrically, creating a spiral pattern with added noise. The 'H' distribution uses rejection sampling to accept points within three rectangular regions that form the letter 'H'. Finally, the 'corners' distribution consists of points sampled from four distinct rectangular areas located in the corners of the sampling domain.

## I Additional Experimental Results

In this section, we present additional experimental results to complement those in Section 5.

Table 5: Hyperparameters of DPSG-MP

| Reward Model | Prior Distribution | Update Steps | Step Size |
|---|---|---|---|
| Cosine | cross | 1 | 0.05 |
| | rays | 10 | 0.005 |
| | triangles | 10 | 0.01 |
| | swirl | 10 | 0.005 |
| | H | 1 | 0.05 |
| | corners | 1 | 0.05 |
| Quadratic | cross | 10 | 0.01 |
| | rays | 1 | 0.05 |
| | triangles | 1 | 0.1 |
| | swirl | 10 | 0.05 |
| | H | 10 | 0.005 |
| | corners | 10 | 0.001 |
| Sigmoid-gated | cross | 1 | 0.01 |
| | rays | 10 | 0.01 |
| | triangles | 10 | 0.1 |
| | swirl | 10 | 0.01 |
| | H | 10 | 0.05 |
| | corners | 10 | 0.05 |

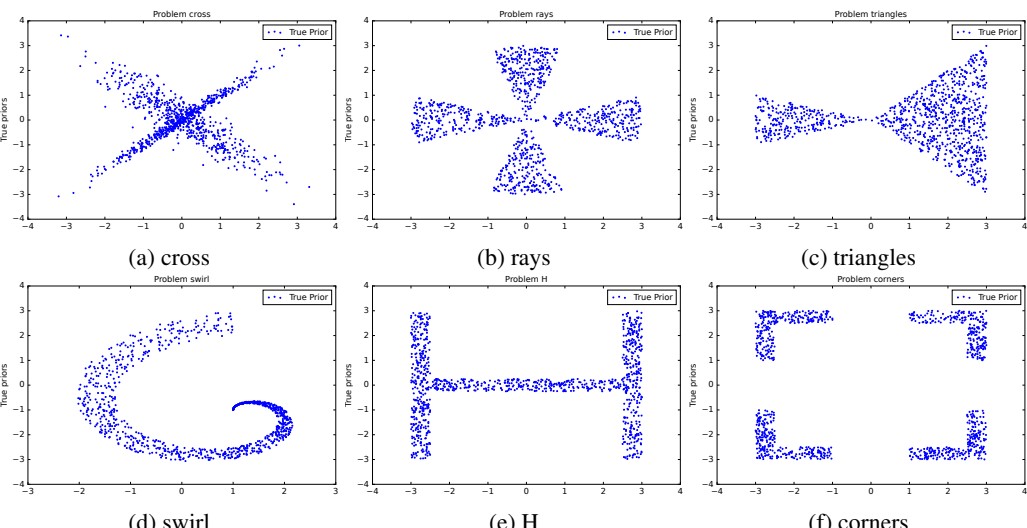

(a) cross  (b) rays  (c) triangles

(d) swirl  (e) H  (f) corners

Figure 6: Prior Distribution. Each panel visualizes a prior over $\theta \in \mathbb{R}^2$. For each prior, we draw $N = 10000$ parameter vectors; $80\%$ are used as training tasks (previously seen tasks) and $20\%$ as test tasks (newly encountered task). These parameters determine the true reward functions in our simulated bandit environments.

## I.1 RESULTS FOR THE TRIANGLES, SWIRL AND H PRIORS IN LINEAR BANDITS AND SIGMOID-GATED BANDITS

We present the triangles, swirl and H results in Linear bandits in Figure 7 and Figure 8. We also put the non-linear bandit results in Figure 9 and Figure 10.

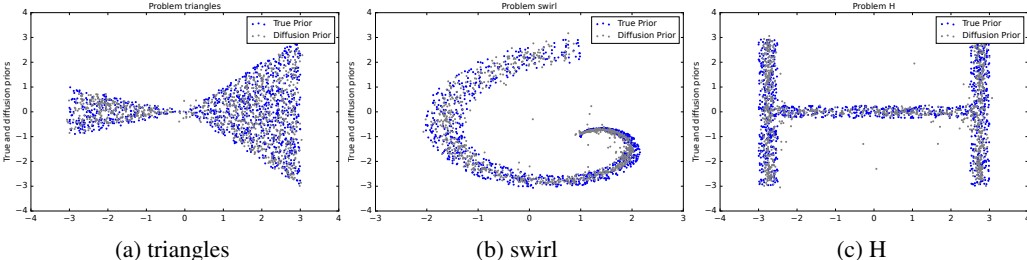

(a) triangles     (b) swirl     (c) H

Figure 7: Visualization of the learned diffusion prior versus the true prior. We demonstrate three of them. For each figure, samples from our trained model (in grey) are overlaid on the ground-truth samples (in blue).

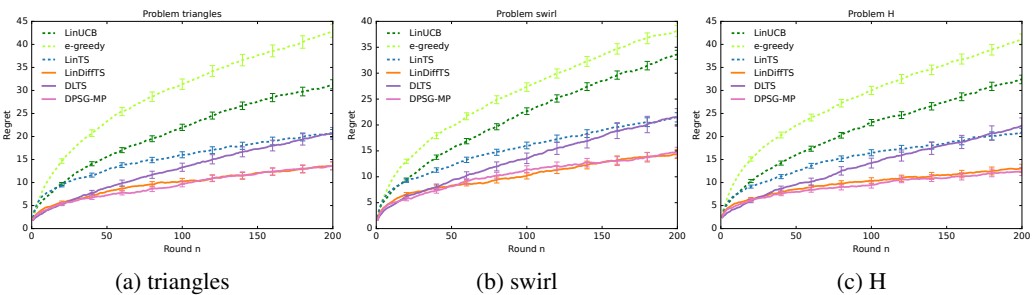

(a) triangles     (b) swirl     (c) H

Figure 8: Performance of all algorithms on the linear contextual bandit tasks for the three prior distributions. Our proposed methods achieve cumulative regret comparable to LinTS and DiffTS.

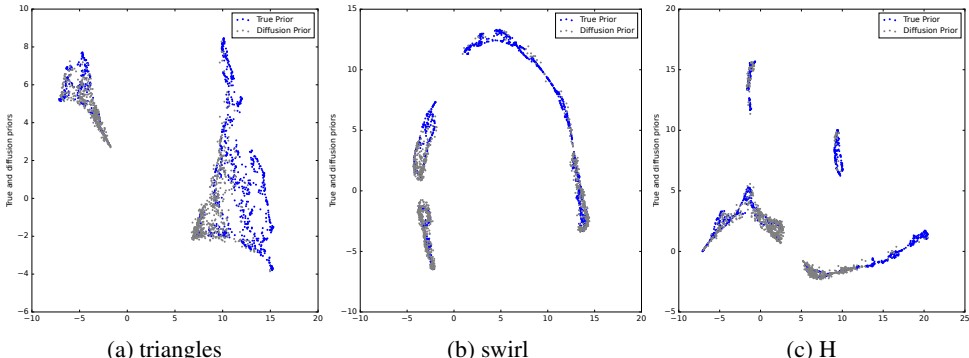

(a) triangles     (b) swirl     (c) H

Figure 9: Visualization of the learned diffusion prior versus the true prior (under sigmoid-gated reward model). The true prior is the MLP parameters learned from the previous task. Samples from our trained model (grey) are overlaid on the ground-truth samples (blue).

## I.2   RESULTS FOR COSINE AND QUADRATIC NONLINEAR BANDITS

In the cosine reward setting, our proposed algorithms demonstrate strong performance, as shown in Figure 12. Both DLTS and the stabilized DPSG-MP achieve cumulative regret comparable to the state-of-the-art NeuralTS and NeuralUCB baselines. This result validates that a diffusion prior over neural network weights can effectively guide exploration in a complex, nonlinear environment. The reported results show the mean cumulative regret and standard error, averaged across 64 new tasks over 8 independent trials.

Similarly, for the quadratic reward model in Figure 14, our methods perform competitively against the strong neural baselines. Notably, for the challenging 'corners' prior, our algorithms outperform both NeuralTS and NeuralUCB. This highlights a key advantage of our approach: when the underlying parameter distribution has a complex structure, the guidance from the learned diffusion prior

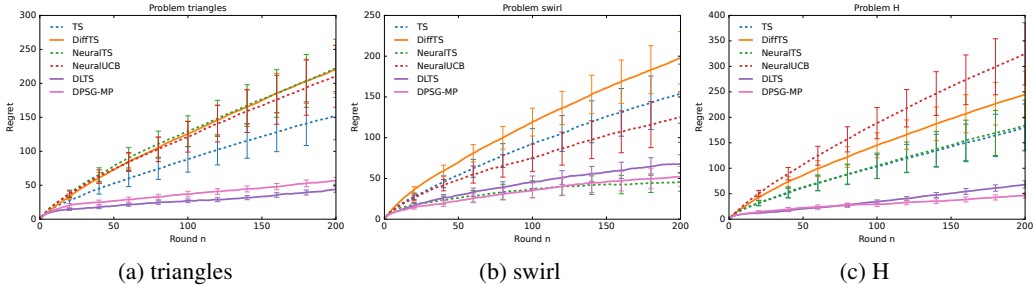

(a) triangles  (b) swirl  (c) H

Figure 10: Performance of all algorithms on the sigmoid-gated nonlinear bandit tasks for the three prior distributions. Our proposed methods outperform strong baselines like NeuralTS and NeuralUCB, achieving the lowest cumulative regret.

becomes particularly beneficial, leading to more efficient exploration and lower regret. The evaluation protocol remains the same, with results averaged over 64 tasks across 8 independent trials.

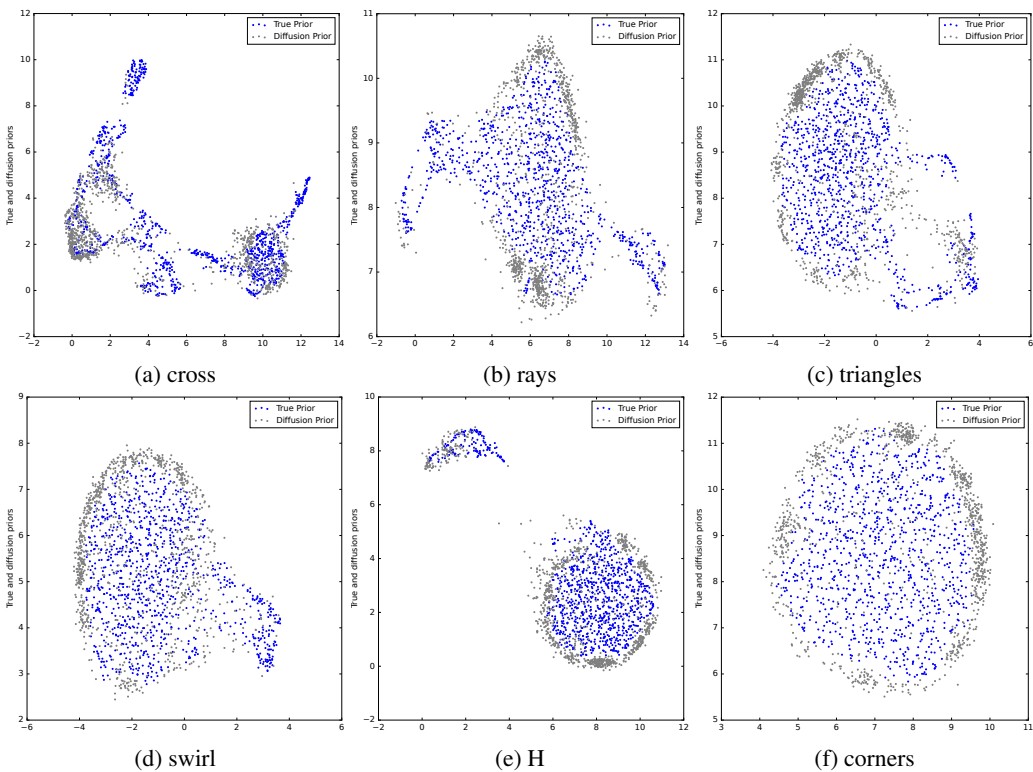

(a) cross  (b) rays  (c) triangles

(d) swirl  (e) H  (f) corners

Figure 11: Visualization of the learned diffusion prior versus the true prior (under cosine reward model). The true prior is the MLP parameters learned from the previous task. Samples from our trained model (grey) are overlaid on the ground-truth samples (blue).

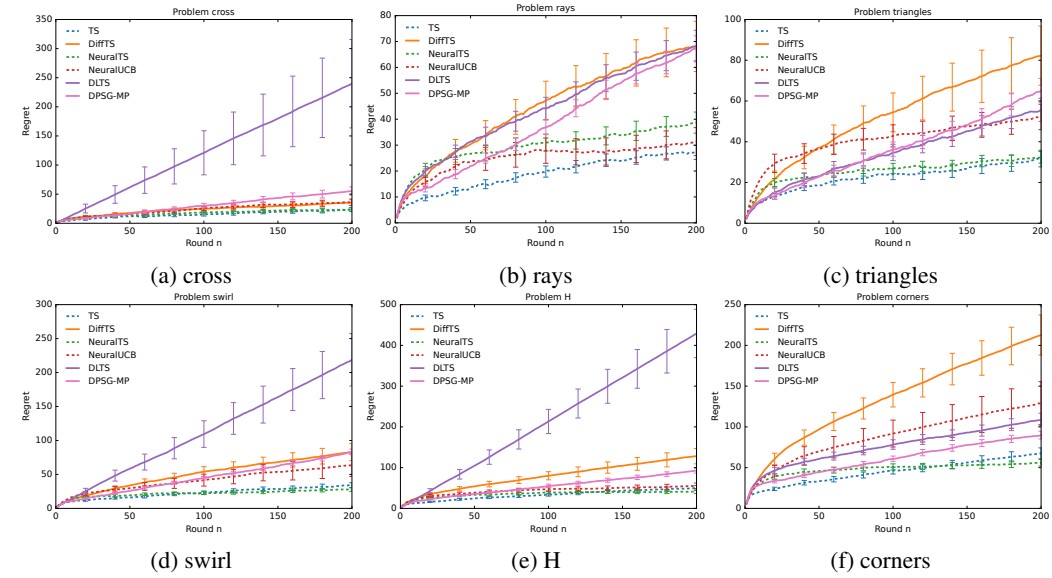

Figure 12: Performance of all algorithms on the cosine nonlinear bandit tasks for the six prior distributions.

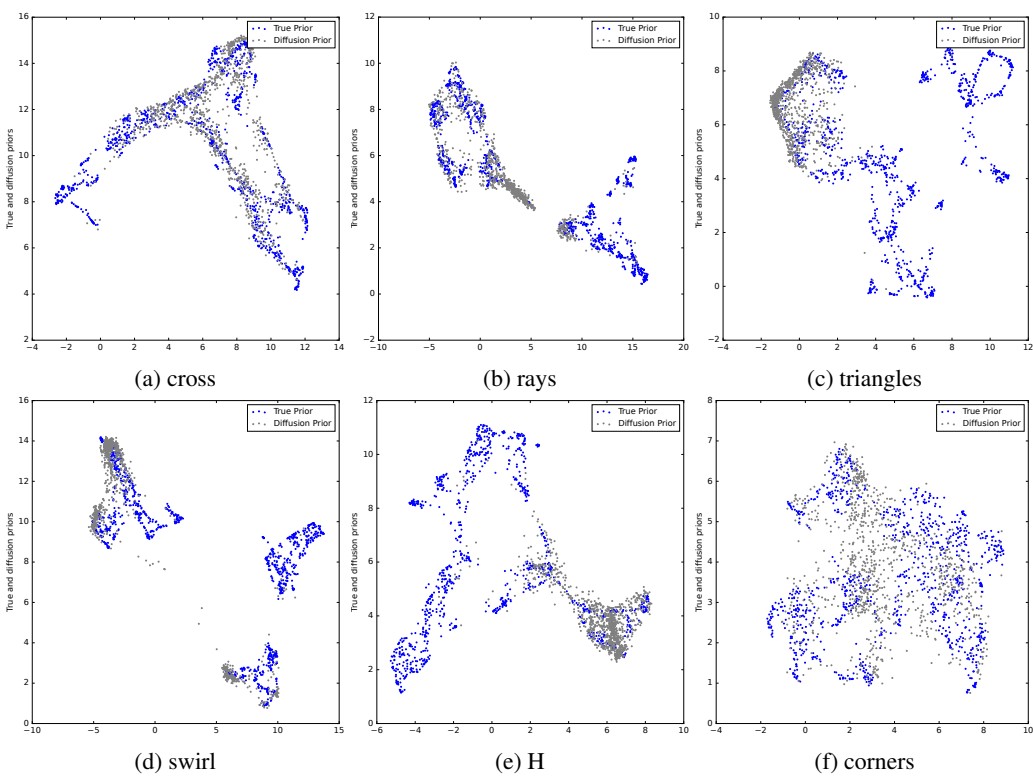

Figure 13: Visualization of the learned diffusion prior versus the true prior (under quadratic reward model). The true prior is the MLP parameters learned from the previous task. Samples from our trained model (grey) are overlaid on the ground-truth samples (blue).

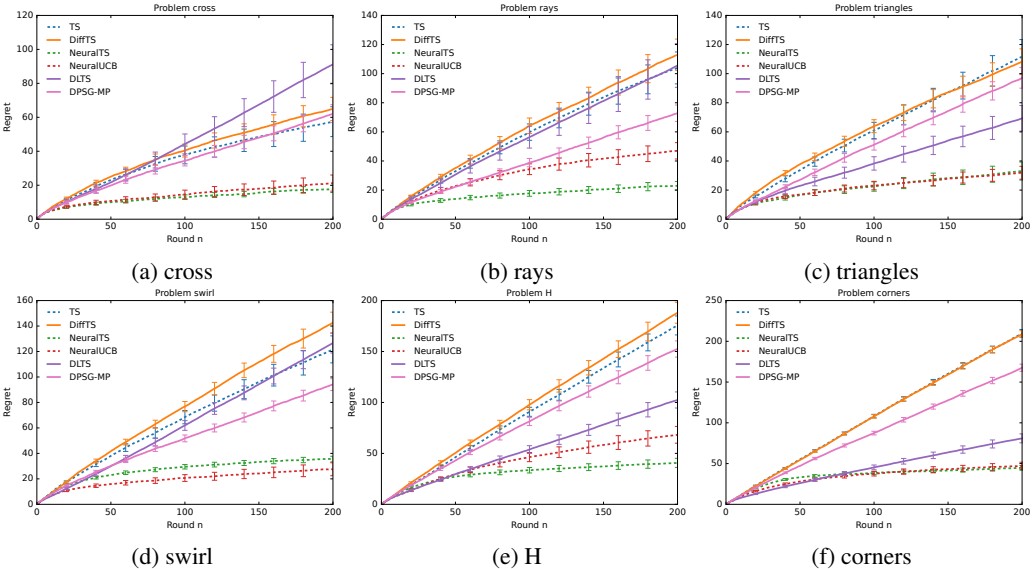

Figure 14: Performance of all algorithms on the quadratic nonlinear bandit tasks for the six prior distributions.

Table 6: Results of ablation study on Inner-Loop Updates $K_\ell$ with fixed Diffusion Steps $L = 100$. Reported values are Cumulative Regret at $T = 200$. The regrets are averaged over 64 tasks across 8 independent runs.

| Environment | Algorithm | Inner-Loop Steps ($K_\ell$) | | | |
|---|---|---|---|---|---|
| | | $K_\ell = 1$ | $K_\ell = 2$ | $K_\ell = 5$ | $K_\ell = 10$ |
| Rays | DLTS | $51.20 \pm 2.90$ | $\mathbf{43.72 \pm 1.58}$ | $63.24 \pm 3.48$ | $55.87 \pm 3.04$ |
| | DPSG-MP | $39.59 \pm 0.80$ | $39.47 \pm 0.85$ | $37.36 \pm 0.33$ | $\mathbf{36.31 \pm 1.09}$ |
| Triangles | DLTS | $79.60 \pm 2.49$ | $52.65 \pm 5.32$ | $160.99 \pm 25.42$ | $\mathbf{47.29 \pm 1.79}$ |
| | DPSG-MP | $54.94 \pm 6.35$ | $51.28 \pm 2.44$ | $\mathbf{49.51 \pm 1.96}$ | $52.20 \pm 3.54$ |
| Swirl | DLTS | $90.20 \pm 3.63$ | $78.81 \pm 0.77$ | $66.49 \pm 2.34$ | $\mathbf{53.95 \pm 1.74}$ |
| | DPSG-MP | $\mathbf{34.80 \pm 1.43}$ | $36.46 \pm 0.87$ | $34.81 \pm 0.93$ | $38.34 \pm 1.63$ |

### I.3 SENSITIVITY TO INNER-LOOP UPDATES $K_\ell$

We then investigate the effect of the number of inner-loop updates $K_\ell$ which are LMC steps for DLTS and projection steps for DPSG-MP, fixing the Diffusion Steps $L = 100$. The results, summarized in Table 6, indicate that a small number of steps (e.g., $K_\ell \in \{1, 2\}$) is generally sufficient to achieve strong performance. Further increasing $K_\ell$ yields diminishing returns and may even degrade performance in certain cases (e.g., DLTS on Triangles with $K_\ell = 5$), likely due to overfitting the local likelihood approximation. Consequently, we recommend a small $K_\ell$ as it offers the best trade-off between regret minimization and computational efficiency.

### I.4 LONG HORIZON EVALUATION

To validate the long-term robustness of our approach, we extend the evaluation of our most challenging nonlinear experiment, the sigmoid-gated reward model, to a horizon of $T = 2000$ rounds

Table 7: Cumulative Regret at $T = 2000$ on Sigmoid-Gated Nonlinear Bandits

| Environment | LinTS | DiffTS | NeuralTS | NeuralUCB | DLTS (Ours) | DPSG-MP (Ours) |
|---|---|---|---|---|---|---|
| Cross | $719.30 \pm 329.99$ | $899.33 \pm 622.08$ | $441.20 \pm 264.00$ | $413.51 \pm 383.25$ | $380.51 \pm 343.59$ | $\mathbf{252.91 \pm 7.50}$ |
| Rays | $1392.70 \pm 545.47$ | $1753.93 \pm 883.84$ | $501.88 \pm 403.60$ | $396.61 \pm 304.10$ | $\mathbf{301.98 \pm 21.53}$ | $355.43 \pm 98.89$ |
| Triangles | $942.77 \pm 505.50$ | $2873.07 \pm 530.49$ | $151.44 \pm 110.73$ | $1717.32 \pm 532.37$ | $513.40 \pm 137.74$ | $272.09 \pm 82.04$ |
| Swirl | $1388.62 \pm 723.92$ | $1012.04 \pm 231.75$ | $295.03 \pm 304.34$ | $997.91 \pm 597.07$ | $515.18 \pm 72.08$ | $\mathbf{176.44 \pm 33.48}$ |
| H | $1702.86 \pm 1043.99$ | $3120.56 \pm 827.30$ | $\mathbf{114.73 \pm 25.09}$ | $1303.26 \pm 603.24$ | $1293.67 \pm 136.93$ | $323.96 \pm 36.16$ |
| Corners | $1539.22 \pm 635.22$ | $2057.89 \pm 1181.21$ | $\mathbf{244.09 \pm 105.71}$ | $3515.52 \pm 953.39$ | $474.54 \pm 56.57$ | $258.20 \pm 24.41$ |

(increased from $T = 200$). The cumulative regret results, summarized in the table below across various prior distributions.

The extended horizon experiments confirm the robustness of our framework. In the 'Cross' and 'Rays' environments, both our methods (DLTS and DPSG-MP) clearly outperform all baselines, achieving the lowest cumulative regret. In 'Swirl' environment, DPSG-MP clearly outperform all baselines. In the remaining three scenarios, our approach consistently achieves top-two performance. Specifically, in 'Corners', NeuralTS yields regret comparable to DPSG-MP (244.09 vs. 258.20) but suffers from significantly higher instability ($\pm 105.71$ vs. $\pm 24.41$). In general, the neural baselines exhibit much higher variance across tasks, whereas our diffusion-based methods provide a more stable and reliable exploration strategy over long horizons.

