# OpenReview forum: "Diffusion Posterior Sampling for Nonlinear Contextual Bandits"
_ICLR.cc/2026/Conference — Submitted to ICLR 2026_

### Official Review · Reviewer_eTKd · 2025-10-29

**Soundness:** 3
**Presentation:** 3
**Contribution:** 2
**Rating:** 4
**Confidence:** 4

**Summary:**

This paper proposes a diffusion-based posterior sampling framework for multi-task nonlinear contextual bandits. The key idea is to replace the standard conjugate prior in Thompson Sampling (TS) with a learned diffusion prior trained on past tasks. Two algorithms are introduced: **DLTS** and **DPSG** and its variant **DPSG-MP**. Theory shows that oracle diffusion TS coincides with standard oracle TS when the learned prior is exact, and bounds the regret gap (when the learned prior is not exact) via cumulative score-estimation error. Experiments on synthetic bandit settings indicate that DLTS and DPSG-MP perform well.

**Strengths:**

**S1.** Clear and well-structured exposition; algorithms are easy to follow.

**S2.** First attempt to apply diffusion posterior sampling beyond the generalized linear bandit case.

**S3.** Provides a unified formulation connecting diffusion priors, Langevin updates, and TS exploration.

**S4.** Empirical results show competitive regret versus specialized baselines in both linear and nonlinear regimes.

**Weaknesses:**

**W1. Novelty.** The paper mainly extends prior work on diffusion-based Thompson sampling for generalized linear bandits [1, 2] to more general nonlinear reward models. As a side note, [2] already considers nonlinear diffusion priors: the linear assumption is made only for theory. Similar to [1], both works rely on the generalized linear models to enable their approximations. The present contribution thus extends those to any form of non-linear rewards.

**W2. Limited theory.**
Theorem 4.4 is a sanity check, confirming equivalence under an exact prior. Theorem 4.5 provides a useful per-round regret bound that highlights an implicit trade-off: smaller diffusion depth $L$ reduces the per-round regret, but might increase the score-estimation error \$\epsilon_{\rm{score}}\$ due to lower expressiveness. However, I think the per-round bound is loose: it does not depend on $t$, relies on TV distance, and a worst-case $f_{\max}$. In particular, it implies linear regret under prior misspecification \$\epsilon_{\rm{score}} \neq 0\$. In addition, the coefficient $\kappa_\ell$ in the bound is never explicitly defined.

**W3. Computational efficiency.**
Proposed approximations are computationally heavy (compared to closed-form approximations), which is undesirable in online decision-making. While the claimed efficiency of DPSG (no inner $K_\ell$ loop) is valid in principle, that variant is never evaluated; the reported DPSG-MP reintroduces a $K_\ell$ loop. It might indeed be faster than DLTS as it might need fewer inner updates due to improved stability, but this should be demonstrated empirically through runtime comparisons. Prior works [1, 2] were explicitly motivated by computational efficiency aspect, avoiding costly approximate-posterior methods such as LMC; hence a fair evaluation should compare not only regret but also wall-clock time to assess the regret–efficiency trade-off.

**W4. Experimental scope.**
All experiments use synthetic priors, short horizons (200 rounds), and only 8 trials. Longer-horizon evaluations (as in [1]) is important. In particular, [1] results showed some interesting behavior of DSP after round 300, given that DPSG relies on a similar idea, it is important to showcase what happens with longer horizons. Moreover, real-world or high-dimensional tasks (e.g., MovieLens as in [1]) would further validate the approach.

**W4-bis. Missing ablations.**
Ablations are absent. In particular, the paper does not compare DPSG and DPSG-MP, nor analyze sensitivity to the number of diffusion steps $L$ or inner iterations $K_\ell$.


**References.**

[1] https://arxiv.org/pdf/2410.03919 (Neurips 2024)

[2] https://arxiv.org/pdf/2402.10028 (Neurips 2025)

**Questions:**

**Q1.** Can the authors report computational/space complexities as well as empirical runtime comparisons between all baselines?

**Q2.** Why was plain DPSG omitted from experiments? Does projection alone explain the observed gains?

**Q3.** How sensitive are results to hyperparameters like the number of diffusion steps $L$, inner updates $K_\ell$, etc?

**Q4.** How sensitive are results to diffusion prior quality (authors can vary the number of tasks used to train the prior, vary $L$, etc.)?

**Q5.** Can the authors share real-world (e.g., MovieLens) semi-synthetic experiments with longer horizons (e.g., 500 for 50 trials for example)?

---

> ### Author Response · Authors · 2025-11-26
>
> ## Response to Reviewer eTKd
>
>
> We thank the reviewer for your valuable time and effort in providing detailed feedback on our work. We hope our response will fully address all your questions.
>
> ---
> ### **Q1:** Explanation for novelty over prior works [1,2]
>
> **A1:** We thank the reviewer for this summary. Our primary contribution is enabling diffusion-based Thompson Sampling for general nonlinear reward models. We would like to clarify the core technical innovation beyond the mechanisms in [1,2]. The reviewer is correct that both [1] and [2] fundamentally rely on a linear or generalized linear model (GLM) structure for the reward. This is because their posterior update mechanisms are built on closed-form, conjugate-like approximations that are only tractable in the linear or GLM setting.
> - In [1] (Kveton et al., 2024), the key idea is to sample from a chain of approximate conditional posteriors which are estimated in a closed form using the Laplace approximation. This closed-form update (Theorems 2 and 4 in [1]) is a product of two Gaussians, which is only possible by applying the Laplace approximation to the likelihood.
> - In [2] (Aouali et al., 2025), while the prior $f_l$ can be non-linear, the posterior update mechanism is similarly restricted. It relies on a Laplace-style approximation of the GLM likelihood to derive closed-form Gaussian posterior expressions (e.g., Eq. 9, 10, 13, 14 in [2]).
>
> This closed-form/conjugate-like update approach, which includes methods like IRLS for logistic models, mathematically breaks down for general nonlinear reward models, such as the sigmoid-gated function we use in our experiments (Section 5.2).
>
> We would like to emphasize that extending diffusion posterior sampling to general nonlinear bandits is far from straightforward. Unlike prior works [1, 2] that rely on convenient closed-form conjugate updates (which are only possible in linear/GLM settings), the nonlinear setting presents a fundamentally intractable posterior inference problem. Our core technical novelty lies in designing the first frameworks to break this reliance through approximate, gradient-based sampling to handle the non-conjugate posterior:
>
> 1. DLTS (Algorithm 1) explicitly constructs the conditional posterior at each reverse step and then draws approximate samples using Langevin Monte Carlo (LMC) , a gradient-based MCMC method.
> 2. DPSG (Algorithm 2) incorporates history by adding a gradient-based guidance step (derived from the likelihood and Tweedie's formula) to the unconditional reverse diffusion step.
>
> This shift from the closed-form updates of [1,2] to our novel gradient-based sampling paradigm is the key contribution that unlocks diffusion priors for the far more general and challenging nonlinear reward setting.

---

> ### Author Response · Authors · 2025-11-26
>
> ---
> ### **Q2:** Explanation for theoretical results
>
> **A2:** We clarify that our primary focus is on algorithm design rather than theoretical analysis, proposing a framework that fundamentally integrates posterior sampling into the iterative diffusion reverse process. Rather than treating the diffusion model as a mere plug-in prior, our approach leverages its multi-step structure to enable stable, history-guided exploration. We appreciate the reviewer's feedback and argue that our theoretical results provide essential foundational guarantees for this new class of non-linear algorithms. Theorem 4.4 is a necessary sanity check: it confirms that our proposed diffusion-based framework is mathematically consistent with standard Thompson Sampling in the ideal limit, ensuring that any performance gaps observed in practice are due to approximation errors (e.g., score estimation, discretization), not a flaw in the underlying methodology itself. Theorem 4.5 establishes the formal link between generative modeling quality (score matching error $\epsilon_{\text {score}}$) and decision-making performance (regret) in the general nonlinear setting. It bounds the per-round regret gap between the ideal Thompson Sampling (OTS) and our diffusion oracle (ODTS) by the cumulative score error, $2 f_{\max} \sum_{\ell=1}^L \kappa_\ell \epsilon_{\text{score}}$. The definition $\kappa_\ell = \frac{c_\ell}{2\sqrt{\lambda_{\text{min}}(\Sigma_\ell)}} = \frac{1- \alpha_\ell}{2\sqrt{\alpha_\ell}\sqrt{\lambda_{\text{min}}(\Sigma_\ell)}}$ is shown in Section D.2 in revised version. As derived in Appendix D.2, it is a coefficient determined by the noise schedule that bounds the Total Variation distance induced by the score error at level $\ell$. Regarding the trade-off highlighted by the reviewer, our bound explicitly captures the tension between model complexity and approximation error: while reducing the diffusion depth $L$ decreases the number of terms in the regret bound summation (potentially lowering the bound), a smaller $L$ limits the expressiveness of the diffusion prior, which inevitably leads to a larger score-matching error $\epsilon_{\text{score}}$. Finally, while a fixed prior misspecification $(\epsilon_{\text{score}}>0)$ logically implies linear cumulative regret, which is an expected result for any Bayesian algorithm with an imperfect prior, our theorem's contribution is to quantify precisely how this misspecification accumulates over the reverse diffusion chain, isolating the source of regret to the learnability of the prior.
>
> ---
> ### **Q3:** Evaluation on wall-clock time
>
> **A3:** We thank the reviewer for highlighting the importance of computational efficiency in online decision-making. To rigorously evaluate the trade-off between computational cost and sample efficiency, we conducted a time-to-accuracy analysis rather than relying solely on per-round inference comparisons. This metric normalizes for sample efficiency: it demonstrates that a computationally heavier algorithm (like ours) can be more time-efficient overall if it requires significantly fewer interactions to reach a high-quality policy compared to faster but less data-efficient baselines.
>
> Experimental Setup: We benchmarked all algorithms in the nonlinear sigmoid-gated bandit setting (using the Triangles prior) with $K_{\ell}=1$. We defined a target performance threshold of **0.5 average regret** (cumulative regret divided by the number of rounds). For each algorithm, we calculated the total wall-clock time required to reach this threshold (computed as the average per-round inference latency multiplied by the number of rounds to convergence). We enforced a maximum time budget of 3000 seconds and performed all evaluations on an Intel Xeon E5-2640 v4 CPU. We include these results (Table 2 and Figure 5) in our revision.
>
>
> | Algorithm | Round | Total Time (s) |
> | :--- | :--- |:--- |
> | DiffTS | >2000 |> 3000 |
> | NeuralTS | 1505 |> 3000 |
> | NeuralUCB | 799  |~1967.71 |
> | **DLTS (Ours)** | **63** |**~1823.81** |
> | **DPSG-MP (Ours)** | **54** |**~ 173.63** |
>
> **Results Analysis:** The results show that our methods (DLTS and DPSG-MP) reach the performance threshold significantly faster than the baselines, despite having a higher per-step computational cost, due to their superior sample efficiency.

---

> ### Author Response · Authors · 2025-11-26
>
> ---
> ### **Q4:** Additional experiments of long-horizon evaluation
>
> **A4:** We thank the reviewer for this excellent suggestion. This is a crucial point, as robust long-term performance is essential.
>
> We have re-run our most challenging nonlinear experiment, the **sigmoid-gated** reward model, with an extended horizon of **T=2000** rounds (up from 200). We summarize the results in the following table and add to Appendix I in our revised manuscript.
>
>
>
> | Environment | LinTS | DiffTS | NeuralTS | NeuralUCB | DLTS | DPSG-MP |
> | :--- | :--- | :--- | :--- | :--- | :--- | :--- |
> | **Cross** | 719.30 ± 329.99 | 899.33 ± 622.08 | 441.20 ± 264.00 | 413.51 ± 383.25 | 380.51 ± 343.59 | 252.91 ± 7.50 |
> | **Rays** | 1392.70 ± 545.47 | 1753.93 ± 883.84 | 501.88 ± 403.60 | 396.61 ± 304.10 | 301.98 ± 21.53 | 355.43 ± 98.89 |
> | **Triangles** | 942.77 ± 505.50 | 2873.07 ± 530.49 | 151.44 ± 110.73 | 1717.32 ± 532.37 | 513.40 ± 137.74 | 272.09 ± 82.04 |
> | **Swirl** | 1388.62 ± 723.92 | 1012.04 ± 231.75 | 295.03 ± 304.34 | 997.91 ± 597.07 | 515.18 ± 72.08 | 176.44 ± 33.48 |
> | **H** | 1702.86 ± 1043.99 | 3120.56 ± 827.30 | 114.73 ± 25.09 | 1303.26 ± 603.24 | 1293.67 ± 136.93 | 323.96 ± 36.16 |
> | **Corners** | 1539.22 ± 635.22 | 2057.89 ± 1181.21 | 244.09 ± 105.71 | 3515.52 ± 953.39 | 474.54 ± 56.57 | 258.20 ± 24.41 |
>
> **Results Analysis:** The extended horizon experiments confirm the robustness of our framework. In the **'Cross' and 'Rays'** environments, both our methods (DLTS and DPSG-MP) clearly outperform all baselines, achieving the lowest cumulative regret. In **'Swirl'** environment, DPSG-MP clearly outperform all baselines. In the remaining three scenarios, our approach consistently achieves top-two performance. Specifically, in **'Corners'**, NeuralTS yields regret comparable to DPSG-MP (244.09 vs. 258.20) but suffers from significantly higher instability ($\pm 105.71$ vs. $\pm 24.41$). **In general, the neural baselines exhibit much higher variance across tasks, whereas our diffusion-based methods provide a more stable and reliable exploration strategy over long horizons.**
>
> Furthermore, **our additional wall-clock time evaluation (Q3)** shows that DLTS and DPSG-MP reach the same performance threshold using substantially fewer rounds (DLTS: 63; DPSG-MP: 54). Their superior sample efficiency also results in shorter overall inference time compared to the baselines.

---

> ### Author Response · Authors · 2025-11-26
>
> ---
> ### **Q5:** Additional experiment of real-world high-dimensional task
>
> **A5:** We have added additional experiments on real-world, high-dimensional tasks to demonstrate the generalization and effectiveness of our proposed DLTS and DPSG-MP algorithms. We selected MovieLens as representative dataset.
>
> While prior works such as DiffTS [1] and dTS [2] also utilize MovieLens for experiments, they primarily focus on synthetic reward functions that are linear or logistic rather than real-world user ratings. Specifically, these methods use the MovieLens-1M dataset (6,040 users and 3,952 movies) and perform low-rank factorization on the sparse rating matrix to obtain 5-dimensional representations for users $U$ and movies $V$.
>
> In DiffTS [1], to construct the contextual bandit, a movie embedding $V_i$ is selected as the task, and in each round, actions are randomly sampled from user embeddings $U$. In dTS [2], the setup is slightly different: a user is randomly selected to define the bandit, and in each round, the algorithm is presented with 100 candidate movie actions. Both methods define their reward functions using either linear $UV$ or logistic $\sigma(UV)$ forms. Consequently, the reward function is explicitly designed rather than derived from real rating labels. In contrast, our setting targets a more complex and realistic environment that requires constructing nonlinear rewards directly from real user data.
>
>
> In MovieLens, we formulate a contextual bandit task using the MovieLens-25M dataset (approx. $1.6 \times 10^5$ users and $6 \times 10^4$ movies). We construct the bandit problem following previous work[6]. We first construct a dense rating matrix by selecting the top 2,000 users and top 10,000 movies, and then apply SVD to extract 10-dimensional feature vectors for each user and movie. We treat each user as a distinct task within the multi-task bandit framework. The objective is to identify specific user preferences (specifically, movies with low ratings). In each task, we select a user embedding. In each round, the agent is presented with $K=10$ arms: 1 "target" arm (a movie with a rating $< 2$, yielding reward $r=1$) and 9 "non-target" arms (ratings $\geq 2$, yielding reward $r=0$). To construct the prior, we train MLPs on the offline dataset to do regression on these movie ratings. A key challenge in diffusing over MLP weights is permutation invariance. To address this and ensure the parameter distribution remains single-modal and learnable, we first train a single "anchor" MLP and then for all subsequent tasks in the prior dataset, we initialize the MLP weights from this anchor before fine-tuning on the specific task data. This ensures the parameters remain aligned in the weight space. We conduct the experiments for our proposed methods compared with NeuralTS and NeuralUCB with $T=500$ with 5 runs. We summarize the results in the following table. We conclude that our proposed algorithm DLTS shows better performance compared to NeuralTS, which shows the effectiveness of our methods. The failure in DPSG-MP may be raised by the bias in estimating the MLP prior and lead to the low-quality posterior synthetic parameters.
>
>
> | Algorithm | Regret |
> | :--- | :--- |
> |**DiffTS**    |444.80 ± 2.18|
> | **NeuralTS** | 364.20 ± 31.08 |
> | **NeuralUCB** |197.80 ± 82.57  |
> | **DLTS** | 245.60 ± 68.31 |
> | **DPSG-MP** | 448.20 ± 1.45 |
>
> **Results Analysis:** The results on the real-world MovieLens dataset emphasize the challenge of modeling complex, non-linear user preferences compared to synthetic settings. DLTS significantly outperforms the neural baseline NeuralTS (245.60 vs. 364.20), demonstrating that our learned diffusion prior provides more effective inductive bias than a Gaussian prior for neural networks. While NeuralUCB achieves a lower mean regret, it exhibits higher variance $(\pm 82.57)$ compared to DLTS, which indicates that our method offers a more stable and robust exploration strategy. We note that DPSG-MP performs similarly to DiffTS in this specific high-dimensional setting. We attribute this to the high sensitivity of the guidance mechanism (based on Tweedie's estimate) when applied to the complex, high-dimensional parameter space of MLP weights. In such cases, DLTS, which directly approximate posterior via Langevin dynamics rather than relying on the likelihood-based guidance, proves to be the more robust and effective choice.

---

> ### Author Response · Authors · 2025-11-26
>
> ---
> ### **Q6:** Ablation study and sensitivity of $L$, $K_{\ell}$
>
> **A6:** We thank the reviewer for raising this important question about hyperparameter sensitivity. We conducted new ablation studies to analyze the impact of the number of diffusion steps ($L$) and the number of inner-loop updates ($K_{\ell}$) on our algorithms' performance.
>
> We first quantitatively analyze the sensitivity of diffusion steps $L$. We used the sigmoid-gated nonlinear bandit as our testbed, as it represents the challenging nonlinear setting from Section 5.2. All results are cumulative regret at T=200, averaged over 64 tasks with 8 runs. For this experiment, we fixed the other parameters to the values used in our main experiments and varied the total number of diffusion steps $L$. **We can conclude that larger diffusion steps yield better performance.**
>
>
>
> ### Rays
> | L | DLTS | DPSG-MP |
> | :--- | :--- | :--- |
> | 20 | 136.0792 ± 3.1829 | 63.9006 ± 0.7246 |
> | 50 | 53.5112 ± 4.4162 | 50.2798 ± 1.0924 |
> | 100 | 51.1978 ± 2.9018 | 36.3051 ± 1.0933 |
>
> ### Triangles
> | L | DLTS | DPSG-MP |
> | :--- | :--- | :--- |
> | 20 | 433.1590 ± 7.5096 | 86.5653 ± 0.6570 |
> | 50 | 97.6766 ± 5.9548 | 73.0886 ± 1.1895 |
> | 100 | 47.2908 ± 1.7911 | 52.1988 ± 3.5373 |
>
> ### Swirl
> | L | DLTS | DPSG-MP |
> | :--- | :--- | :--- |
> | 20 | 732.7003 ± 7.2932 | 75.4511 ± 0.5929 |
> | 50 | 125.5525 ± 4.0634 | 52.4696 ± 0.5126 |
> | 100 | 53.9466 ± 1.7429 | 38.3384 ± 1.6278 |
>
>
>
> We then analyzed the sensitivity to the number of inner-loop updates $K_{\ell}$. For this experiment, we fixed $L=100$ and varied the number of inner-loop updates $K_{\ell}$ (the LMC steps in DLTS and projection steps in DPSG-MP). **Our analysis suggests that a small $K_{\ell}$ (e.g., 1 or 2) often provides the best trade-off between performance and computational cost.**
>
>
>
> ### Rays
> | Inner-Loop Steps | DLTS | DPSG-MP |
> | :--- | :--- | :--- |
> | 1 | 51.1978 ± 2.9018 | 39.5867 ± 0.8039 |
> | 2 | 43.7158 ± 1.5759 | 39.4732 ± 0.8546 |
> | 5 | 63.2363 ± 3.4842 | 37.3604 ± 0.3303 |
> | 10 | 55.8693 ± 3.0374 | 36.3051 ± 1.0933 |
>
>
>
> ### Triangles
> | Inner-Loop Steps | DLTS | DPSG-MP |
> | :--- | :--- | :--- |
> | 1 | 79.5997 ± 2.4942 | 54.9418 ± 6.3533 |
> | 2 | 52.6539 ± 5.3189 | 51.2819 ± 2.4380 |
> | 5 | 160.9929 ± 25.4214 | 49.5074 ± 1.9641 |
> | 10 | 47.2908 ± 1.7911 | 52.1988 ± 3.5373 |
>
> ### Swirl
> | Inner-Loop Steps | DLTS | DPSG-MP |
> | :--- | :--- | :--- |
> | 1 | 90.1970 ± 3.6337 | 34.7951 ± 1.4286 |
> | 2 | 78.8142 ± 0.7685 | 36.4597 ± 0.8702 |
> | 5 | 66.4889 ± 2.3441 | 34.8075 ± 0.9337 |
> | 10 | 53.9466 ± 1.7429 | 38.3384 ± 1.6278 |
>
>
> We include these detailed ablation studies in our revision.
>
>
> ---
> ### **Q7:** Clarification about DPSG performance in experiments
>
> **A7:** We did not include the results for DPSG (Algorithm 2) in the main paper's figures because its empirical performance is unstable and substantially worse than both DLTS and our proposed variant, DPSG-MP.
>
> The core issue, as we noted in Section 3.3, is that plain DPSG relies on a single likelihood-gradient correction at each reverse step.
>
> To demonstrate this empirically, we run an experiment comparing DPSG against DLTS and DPSG-MP on the sigmoid-gated nonlinear bandit tasks from Figure 4. The results, shown in the table below.
>
>
>
> | Prior Distribution | DPSG-MP | DLTS | DPSG |
> | :--- | :--- | :--- | :--- |
> | Rays | 36.3051 ± 1.0933 | 51.1978 ± 2.9018 | 326.5208 ± 0.9431 |
> | Triangles | 52.1988 ± 3.5373 | 47.2908 ± 1.7911 | 375.4471 ± 1.1921 |
> | Swirl | 38.3384 ± 1.6278 | 53.9466 ± 1.7429 | 387.2442 ± 1.9992 |

---

> ### Author Response · Authors · 2025-11-26
>
> ---
> ### **Q8:** How sensitive are results to diffusion prior quality?
>
> **A8:** We thank the reviewer for this important question. The quality of the learned prior is indeed a crucial factor.  To quantitatively measure this sensitivity, we conducted a new ablation study by training the diffusion prior on varying subsets of the training dataset. We trained three separate diffusion priors using 10%, 50%, and 100% of the dataset, respectively. We then evaluated the performance of our DLTS and DPSG-MP algorithms on the nonlinear contextual bandit task described in Section 5.2.
>
> The results, measured in Cumulative Regret at T=200, are summarized in the table below. **We conclude that the diffusion prior quality will affect the performance. With smaller dataset like 10\% of the dataset, the performance drops compared to the full dataset.**
>
> ### Problem: Swirl
> | Prior Training Data | DLTS | DPSG-MP |
> | :--- | :--- | :--- |
> | **10%** | 60.4547 ± 0.3335 | 41.5447 ± 0.3013 |
> | **50%** | 60.0962 ± 0.5088 | 38.3751 ± 0.3469 |
> | **100%** | 53.9466 ± 1.7429 | 38.3384 ± 1.6278 |
>
> ### Problem: Triangles
> | Prior Training Data | DLTS | DPSG-MP |
> | :--- | :--- | :--- |
> | **10%** | 47.7303 ± 0.7707 | 57.6593 ± 0.6928 |
> | **50%** | 47.4867 ± 0.6109 | 50.9737 ± 0.9494 |
> | **100%** | 47.2908 ± 1.7911 | 52.1988 ± 3.5373 |
>
>
> ---
> We hope we have addressed all of your questions. If you have any further questions, we would be happy to answer them and if you don’t, would you kindly consider increasing your score?
>
>
>
> ### References:
>
> [1] Kveton, Branislav, et al. "Online posterior sampling with a diffusion prior." Advances in Neural Information Processing Systems 37 (2024): 130463-130484.
>
> [2] Aouali, Imad. "Diffusion models meet contextual bandits." Advances in Neural Information Processing Systems (2025).
>
> [3] Ban, Yikun, et al. "EE-Net: Exploitation-Exploration Neural Networks in Contextual Bandits." International Conference on Learning Representations (2022).

---

### Official Review · Reviewer_J2mU · 2025-11-01

**Soundness:** 3
**Presentation:** 3
**Contribution:** 3
**Rating:** 6
**Confidence:** 4

**Summary:**

Thompson sampling (TS) is a widely used bandit algorithm that maintains a Bayesian posterior over the parameters of a task reward function. A challenge for TS is that it requires a prior distribution over the parameters that admits a tractable update. This presents a difficulty for domains with complex prior distributions. The main contribution of this paper is to show how TS can be implemented with diffusion models so as to provide a flexible prior. The paper considers a multi-task learning setting in which multiple samples of true task reward parameters are available to train a prior distribution. The method introduced uses a diffusion model to train this prior. The model is constructed such that it can be updated as more actions are taken (and rewards observed) while running TS. The paper shows empirically that this approach lowers cumulative regret compared to a well-chosen set of baselines.

**Strengths:**

- Thomson Sampling is a very widely used algorithm but stuggles with complex priors. This paper shows how modern generative models can be used to address this limitation.
- The empirical study convincingly supports the central claims of the paper.
- The paper is generally well-written and the authors have made an effort to provide both intuitive high-level descriptions and precise description of details.

**Weaknesses:**

- The technical novelty of the theoretical results is not clearly highlighted. It seems that the assumption is that the oracle diffusion approach can learn the true prior and posterior update. Under this assumption, it seems trivial to claim that the diffusion approach matches the oracle.
- Why does "Can we design diffusion posterior sampling algorithms for general nonlinear contextual bandits?" matter as a research question? I buy that we care about solving general nonlinear contextual bandits. I am less sure why diffusion posterior sampling is an essential part of doing that.
- Equation 3.1: while I appreciate the effort to make equation understandable, there do not appear to be any formal definition of the terms in this equation. This makes it difficult to understand precisely what each term means.
- The experiments only consider synthetic, low-dimensional domains. There are two sets of experiments: the first using linear reward functions and the second using non-linear reward functions. While the second one uses neural network reward approximations, the underlying domain is still fairly simple. Evaluation on higher dimensional or real-world domains would further solidify the paper's contribution.

**Questions:**

The following set of questions are supplemental to the weaknesses mentioned above. Addressing the weaknesses is the most important issue for the discussion phase.
- What is the prior for LinTS?
- Figure 3 shows samples from the prior which is over neural network parameters. How is this just shown in two dimensions?
- Can you elaborate on the key technical novelty of the theoretical results?
- How does the learning history affect the posterior in Algorithm 1? I don’t see the learning history being used, only updated.
- Same question for Algorithm 2.
- This work seems closely related to works that learn algorithms in multi-task settings. For instance: "Supervised Pretraining Can Learn In-Context Reinforcement Learning," ""In-context Reinforcement Learning with Algorithm Distillation," and "Pretraining decision transformers with reward prediction for in-context multi-task structured bandit learning" (among many others). It could be interesting to benchmark the proposed method against these methods.

---

> ### Author Response · Authors · 2025-11-26
>
> ## Response to Reviewer J2mU
>
> We thank the reviewer for your valuable time and effort in providing detailed feedback on our work. We hope our response will fully address all your questions.
>
> ---
> ### **Q1:** Explanation for theoretical results
>
> **A1:** We clarify that our primary focus is on algorithm design rather than theoretical analysis, proposing a framework that fundamentally integrates posterior sampling into the iterative diffusion reverse process. Rather than treating the diffusion model as a mere plug-in prior, our approach leverages its multi-step structure to enable stable, history-guided exploration. We appreciate the reviewer's feedback and argue that our theoretical results provide essential foundational guarantees. We view Theorem 4.4 as a necessary sanity check rather than a trivial claim. It confirms that our proposed diffusion-based framework is mathematically consistent with standard Thompson Sampling in the ideal limit, ensuring that any performance gaps observed in practice are due to approximation errors (e.g., score estimation, discretization), not a flaw in the underlying methodology itself. Theorem 4.5 provides the formal link between generative modeling quality (score matching error $\epsilon_{\text {score}}$) and decision-making performance (regret) in the general non-linear setting. It bounds the gap from the learned prior (OTS vs. ideal ODTS), which depends on the cumulative score-matching error.
>
> Extending this analysis to our practical algorithms, DLTS and DPSG, is a significant theoretical challenge precisely because they rely on inexact (approximate) posterior updates. A full analysis would require introducing a new, distinct error term, denoted as $\Delta^{\text {Approx}}$, to capture the gap between the ideal ODTS and our practical implementation. This $\Delta^{\text {Approx}}$ term would need to quantify the specific approximation error of the posterior at each reverse step: for DLTS (Algorithm 1), this arises from the use of finite-step Langevin Monte Carlo (LMC) to sample from the per-level posterior $p(\theta_ {\ell-1} | \theta_ {\ell}, \mathcal{H}_ t)$; for DPSG (Algorithm 2), this arises from the Tweedie's formula approximation used to estimate the intractable likelihood score $\nabla_\theta \log p_s(\mathcal{H}_ t | \theta_ s)$. The main theoretical challenge, which we leave as an important direction for future work, is in formally bounding how this per-step approximation error $\Delta^{\text {Approx}}$ accumulates over the $L$ steps of the reverse-diffusion chain and interacts with the prior score-matching error $\epsilon_{\text {score}}$.
>
>
> ---
> ### **Q2:** The advantage of diffusion posterior sampling in solving nonlinear contextual bandits
>
> **A2:** Thank you for this excellent question. It highlights the core motivation of our work. Diffusion posterior sampling is essential because it provides a computationally tractable way to use rich, complex priors for solving nonlinear contextual bandits, which is a key challenge in the multi-task setting. In this setting, the prior distribution over task parameters is often highly complex, such as being multi-modal or heavy-tailed. A simple prior, like a Gaussian, would be misspecified and lead to suboptimal exploration. A diffusion model is powerful enough to learn and represent this complex prior from past tasks. However, just having a complex prior is not enough. We need to perform posterior sampling at every round. The diffusion model's iterative reverse process is uniquely suited for this: it provides a natural framework (our Unified Update, Eq. 3.1) to inject the new task's history (as the likelihood-driven drift) at each sampling step. This allows the agent to efficiently adapt the complex learned prior to the new task's data, which is not straightforward with other generative models.

---

> ### Author Response · Authors · 2025-11-26
>
> ---
> ### **Q3:** Detailed explanation of terms in eq(3.1)
>
> **A3:** We thank the reviewer for this valuable feedback. Equation 3.1 is presented as a high-level conceptual framework to unify our two proposed algorithms. We now provide a formal mapping from this conceptual equation to the specific update rules of DLTS and DPSG.
> 1. Unconditional Prior Drift: This term represents the mean of the standard, pre-trained reverse diffusion step $p(\theta_{\ell-1} | \theta_\ell)$, which pulls the current state $\theta_\ell$ back toward the learned prior distribution.
> - In DLTS (Eq. 3.3): This is represented by the LMC drift components that pull $\theta_\ell$ toward the unconditional mean $\mu_\ell$ (i.e., the terms $(1-\ldots) \theta_\ell + (\ldots) \mu_\ell$).
> - In DPSG (Eq. 3.6): This is the explicit unconditional posterior mean $\frac{1}{\sqrt{\alpha_\ell}} \theta_\ell+ \frac{\beta_\ell}{\sqrt{\alpha_\ell}} s_{\psi^*}(\theta_\ell, \ell)$.
> 2. Likelihood-Driven Drift: This term incorporates the online learning history $\mathcal{H}_t$ via the gradient of the negative log-likelihood loss $L_t(\theta)$.
> - In DLTS (Eq. 3.3): This is the explicit LMC gradient term $-\eta_\ell \nabla_\theta L_\ell (\theta_\ell / \sqrt{\bar{\alpha}_{\ell -1}})$.
> - In DPSG (Eq. 3.6): This is the explicit guidance term $-\eta_\ell \nabla_\theta L_\ell (\widehat{\theta}_0(\theta_\ell, \ell))$.
> 3. Randomized Exploration Noise: This is the stochastic component required for sampling and exploration.
> - In DLTS (Eq. 3.3): This is the Langevin noise term $\sqrt{2 \eta_\ell \zeta_\ell^{-1}} \xi_k$.
> - In DPSG (Eq. 3.6): This is the standard ancestral sampling noise $z_\ell$.

---

> ### Author Response · Authors · 2025-11-26
>
> ---
> ### **Q4:**  Additional experiment of real-world high-dimensional task
>
> **A4:** We have added additional experiments on real-world, high-dimensional tasks to demonstrate the generalization and effectiveness of our proposed DLTS and DPSG-MP algorithms. We selected MovieLens as representative dataset.
>
>
> While prior works such as DiffTS [1] and dTS [2] also utilize MovieLens for experiments, they primarily focus on synthetic reward functions that are linear or logistic rather than real-world user ratings. Specifically, these methods use the MovieLens-1M dataset (6,040 users and 3,952 movies) and perform low-rank factorization on the sparse rating matrix to obtain 5-dimensional representations for users $U$ and movies $V$.
>
> In DiffTS [1], to construct the contextual bandit, a movie embedding $V_i$ is selected as the task, and in each round, actions are randomly sampled from user embeddings $U$. In dTS [2], the setup is slightly different: a user is randomly selected to define the bandit, and in each round, the algorithm is presented with 100 candidate movie actions. Both methods define their reward functions using either linear $UV$ or logistic $\sigma(UV)$ forms. **Consequently, the reward function is explicitly designed rather than derived from real rating labels. In contrast, our setting targets a more complex and realistic environment that requires constructing nonlinear rewards directly from real user data.**
>
>
> In MovieLens, we formulate a contextual bandit task using the MovieLens-25M dataset ($1.6 \times 10^5$ users and $6 \times 10^4$ movies). We construct the bandit problem following previous work [6]. We first construct a dense rating matrix by selecting the top 2,000 users and top 10,000 movies, and then apply SVD to extract 10-dimensional feature vectors for each user and movie. We treat each user as a distinct task within the multi-task bandit framework. The objective is to identify specific user preferences (specifically, movies with low ratings). In each task, we select a user embedding. In each round, the agent is presented with $K=10$ arms: 1 "target" arm (a movie with a rating $< 2$, yielding reward $r=1$) and 9 "non-target" arms (ratings $\geq 2$, yielding reward $r=0$). To construct the prior, we train MLPs on the offline dataset to do regression on these movie ratings. A key challenge in diffusing over MLP weights is permutation invariance. To address this and ensure the parameter distribution remains single-modal and learnable, we first train a single "anchor" MLP and then for all subsequent tasks in the prior dataset, we initialize the MLP weights from this anchor before fine-tuning on the specific task data. This ensures the parameters remain aligned in the weight space. We conduct the experiments for our proposed methods compared with NeuralTS and NeuralUCB with $T=500$ with 5 runs. We summarize the results in the following table.
>
>
>
> | Algorithm | Regret |
> | :--- | :--- |
> |**DiffTS**    |444.80 ± 2.18|
> | **NeuralTS** | 364.20 ± 31.08 |
> | **NeuralUCB** |197.80 ± 82.57  |
> | **DLTS** | 245.60 ± 68.31 |
> | **DPSG-MP** | 448.20 ± 1.45 |
>
>
> **Results Analysis:** The results on the real-world MovieLens dataset emphasize the challenge of modeling complex, non-linear user preferences compared to synthetic settings. DLTS significantly outperforms the neural baseline NeuralTS (245.60 vs. 364.20), demonstrating that our learned diffusion prior provides more effective inductive bias than a Gaussian prior for neural networks. While NeuralUCB achieves a lower mean regret, it exhibits higher variance $(\pm 82.57)$ compared to DLTS, which indicates that our method offers a more stable and robust exploration strategy. We note that DPSG-MP performs similarly to DiffTS in this specific high-dimensional setting. We attribute this to the high sensitivity of the guidance mechanism (based on Tweedie's estimate) when applied to the complex, high-dimensional parameter space of MLP weights. In such cases, DLTS, which directly approximate posterior via Langevin dynamics rather than relying on the likelihood-based guidance, proves to be the more robust and effective choice.
>
> ---
> ### **Q5:** What is the prior for LinTS?
>
> **A5:** The standard prior for LinTS (Linear Thompson Sampling) is a multivariate Gaussian prior. This is a conjugate prior for the linear reward model with Gaussian noise, which is why it's used: it allows the posterior to also be a Gaussian, which can be computed efficiently in closed form. In our experiments, we follow this standard choice and use a Gaussian prior, specifically $\mathcal{N}(0, I)$, for the LinTS baseline.

---

> ### Author Response · Authors · 2025-11-26
>
> ---
> ### **Q6:** How is figure 3 (priors over neural network parameters) shown in two dimensions?
>
> **A6:** Since the neural network parameters reside in a high-dimensional space, we employ Uniform Manifold Approximation and Projection (UMAP) to reduce their dimensionality for visualization. This technique projects the high-dimensional weight vectors onto a 2D plane while preserving the local and global structure of the data, allowing us to visually compare the distribution of our learned diffusion prior against the ground truth.
>
>
> ---
> ### **Q7:** How does the learning history affect the posterior in Algorithm 1&2?
>
> **A7:** The learning history $\mathcal{H}_ t=\{(x_i, y_i)\}_ {i=1}^t$ is used to compute the negative log-likelihood loss function, $L_ t(\theta)$. As defined in Section 3.2 (line 224) and Appendix C.1 (line 792), this loss is the sum of squared errors over all past observations: $L_t(\theta)=\sigma^{-2} \sum_{i=1}^{t-1}(f(\theta ; x_i)-y_i)^2$. This $L_t(\theta)$ term is the likelihood term in our unified update framework (Eq. 3.1).
>
> Algorithm 1 (DLTS): In DLTS, this history-dependent loss function is central to the posterior sampling. The LMC update in line 9 of Algorithm 1 explicitly includes the gradient of this loss, $\nabla_ \theta L_t$. As derived in Appendix C.1 (Eq. C.5), this gradient term $\nabla_ \theta L_ t(\theta_ {\ell, k} / \sqrt{\bar{\alpha}_ {\ell-1}})$ is precisely what makes the update an approximate sampler for the true per-level posterior $p(\theta_{\ell-1} | \theta_\ell, \mathcal{H}_t)$. It effectively incorporates the history $\mathcal{H}_t$ at every reverse step.
>
> Algorithm 2 (DPSG): Similarly, in DPSG, this same gradient $\nabla_\theta L_t$ is the guidance term in line 8 of Algorithm 2. As derived in Appendix C.2 (Eq. 3.4, C.9), this gradient (evaluated at the Tweedie estimate $\hat{\theta}_ 0$) serves as a tractable approximation for the likelihood score $\nabla_\theta \log p_s(\mathcal{H}_t | \theta_s)$. This gradient term is added to the unconditional reverse step to guide the sample toward the posterior defined by the history $\mathcal{H}_t$.
>
> In both cases, the history $\mathcal{H}_ t$ is the source of the crucial $\nabla_ \theta L_ t$ term that allows our algorithms to perform conditional posterior sampling. We provide a full, step-by-step derivation and interpretation of how this history is used in both algorithms in Appendix C.

---

> ### Author Response · Authors · 2025-11-26
>
> ---
> ### **Q8:** Discussion on mentioned literature and benchmark
>
> **A8:** We thank the reviewer for highlighting these important works. We agree that they represent a significant parallel approach to multi-task decision-making: In-Context Learning (ICL) via Transformers. In contrast, our work establishes a framework for Diffusion-based Posterior Sampling.
>
> We have added DPT (as a representative ICL method) to our baselines in the additional experiments to empirically benchmark these approaches. Theoretically, we distinguish our method from ICL methods (DPT [3], AD [4], PreDeToR [5]) based on the following key advantages:
> 1. ICL methods must ingest the entire interaction history ($x_1, a_1, r_1, \ldots$) as a sequence. Their inference cost and memory complexity scale with the task horizon $T$. In contrast, our method operates in the parameter space; the history is compressed into the likelihood gradient term $\nabla_\theta L_t$. This makes our inference cost independent of the horizon length, offering superior scalability for long-horizon tasks.
> 2. DPT explicitly requires optimal action labels (oracle) during pretraining, which are often unavailable in real-world offline datasets. Our diffusion method learns the prior distribution of parameters $\theta$ directly from data (which can be suboptimal), without needing such oracle supervision.
> 3. Methods like AD work by cloning a source algorithm (e.g., UCB), meaning their performance is upper-bounded by that source algorithm. Our method performs explicit Bayesian posterior sampling via the diffusion reverse process, allowing it to recover the true posterior structure from the data distribution itself, rather than imitating a specific source algorithm.
>
> Our additional experiments on sigmoid-gated bandit settings demonstrate superior performance compared to DPT. We summarize the cumulative regret with 64 tasks in 8 runs in the following table.
>
>
> | Problem | DLTS  | DPSG-MP  | DPT   |
> | :--- | :--- | :--- | :--- |
> | **Cross** | 44.3341 ± 3.0742 | 34.8733 ± 0.4400 | 320.8006 ± 13.9658 |
> | **Rays** | 51.1978 ± 2.9018 | 36.3051 ± 1.0933 | 587.1248 ± 16.0626 |
> | **Triangles** | 47.2908 ± 1.7911 | 52.1988 ± 3.5373 | 597.3580 ± 38.9246 |
> | **Swirl** | 53.9466 ± 1.7429 | 38.3384 ± 1.6278 | 562.0886 ± 15.7313 |
> | **H** | 121.3069 ± 5.9771 | 52.4910 ± 3.4348 | 733.2195 ± 42.0738 |
> | **Corners** | 57.9246 ± 2.3493 | 38.1240 ± 1.2652 | 955.3090 ± 16.5104 |
>
> **Results Analysis:** The empirical results reveal a substantial performance gap: DPT incurs drastically higher cumulative regret across all environments, indicating a failure to effectively adapt to the complex nonlinear reward settings. In contrast, our diffusion-based methods (DLTS and DPSG-MP) successfully leverage the diffusion prior to guide stable exploration. Although ICL-style approaches such as DPT can perform well in multi-task bandit settings, they generally require far more data to learn a reliable prior and to learn optimal policy during training.
>
>
> ---
> We hope we have addressed all of your questions. If you have any further questions, we would be happy to answer them and if you don’t, would you kindly consider increasing your score?
>
>
>
> ### References:
>
> [1] Kveton, Branislav, et al. "Online posterior sampling with a diffusion prior." Advances in Neural Information Processing Systems 37 (2024): 130463-130484.
>
> [2] Aouali, Imad. "Diffusion models meet contextual bandits." Advances in Neural Information Processing Systems (2025).
>
>
> [3] Lee, Jonathan, et al. "Supervised pretraining can learn in-context reinforcement learning." Advances in Neural Information Processing Systems 36 (2023): 43057-43083.
>
> [4] Laskin, Michael, et al. "In-context reinforcement learning with algorithm distillation." arXiv preprint arXiv:2210.14215 (2022).
>
> [5] Mukherjee, Subhojyoti, et al. "Pretraining decision transformers with reward prediction for in-context multi-task structured bandit learning." arXiv preprint arXiv:2406.05064 (2024).
>
> [6] Ban, Yikun, et al. "EE-Net: Exploitation-Exploration Neural Networks in Contextual Bandits." International Conference on Learning Representations (2022).

---

### Official Review · Reviewer_5DY8 · 2025-11-02

**Soundness:** 3
**Presentation:** 3
**Contribution:** 2
**Rating:** 4
**Confidence:** 3

**Summary:**

This paper studies multi-task nonlinear contextual bandits where task parameters are drawn from a shared unknown prior. The authors propose an approach that learns the prior with a diffusion model trained on past tasks, then performs posterior sampling on new tasks via a conditional reverse-diffusion process that combines an unconditional drift from the diffusion prior, a likelihood-driven drift from the interaction history, and a noise term for randomized exploration. They offer two concrete algorithms: DLTS, which runs LMC at each diffusion level, and DPSG, which takes unconditional DDPM steps plus a single Tweedie-guided likelihood step (with a stabilized DPSG-MP variant). The paper argues that oracle diffusion Thompson sampling matches standard oracle Thompson sampling when the learned score is exact, and it bounds the per-round regret gap by the cumulative score-estimation error across diffusion levels. Experiments on synthetic data show competitive results in linear settings and gains in nonlinear ones.

**Strengths:**

I think combining multi-task bandits with a learned generative prior is a reasonable idea. The per-level decomposition is a clean way to view conditional diffusion for decision making, and both instantiations of the algorithm follow naturally from this view.
The experiments also empirically reveal the effectiveness of the approach. The visualizations make sense and help illustrate how the diffusion prior behaves in practice.

**Weaknesses:**

- The methods treat the diffusion prior largely as a plug-in prior/score oracle with path-wise conditioning. The per-level update separates an unconditional diffusion drift from a likelihood drift, and the theory hinges only on score accuracy. Thus, while the algorithms do use diffusion-time dynamics, the benefits appear model-agnostic and could likely be obtained with other generative oracles offering similar score or denoising interfaces. It would be more important to see how the diffusion denoiser can be trained or adapted online on the new task.

- The justification of DPSG-MP is mainly empirical. The multi-step projection variant is introduced to avoid collapse toward a MAP-like update, but there is little quantitative analysis of its bias or diversity trade-offs.

**Questions:**

- Can the authors elaborate on why a diffusion prior is preferred compared to other generative-prior choices (for example, normalizing flows or score-only EBMs) in this Thompson Sampling setting?

---

> ### Author Response · Authors · 2025-11-26
>
> ## Response to Reviewer 5DY8
>
> We thank the reviewer for your valuable time and effort in providing detailed feedback on our work. We hope our response will fully address all your questions.
>
> ---
> ### **Q1:** Why is a diffusion prior preferred compared to other generative-prior choices?
>
>
> **A1:** The central challenge in Thompson Sampling is the need to draw a sample from the posterior distribution $p(\theta | \mathcal{H}_t)$ at every single round $t$. This posterior combines the complex learned prior $p_0(\theta)$ and the new likelihood $p(\mathcal{H}_t | \theta)$.
>
> The diffusion model is preferred because it offers a computationally sound trade-off between sampling stability and algorithmic flexibility, which is essential for Thompson Sampling in *online* nonlinear multi-task bandits. Our approach relies on the unique ability of the diffusion model's iterative, $L$-step reverse process to enable stable, guided sampling by combining the complex learned prior drift with new likelihood information at every step. This iterative guidance is not feasible with other generative models:
>
> - Tractability and Guidance (vs. Normalizing Flows): Normalizing Flows (NFs) define distributions via a single, invertible transformation $\mathbf{x}=T(\mathbf{u})$ [1]. While NFs offer fast prior sampling, this one-shot deterministic structure lacks the intermediate stochastic steps necessary for our per-level conditioning. To use NFs in this context, one would have to perform complex inference (like MCMC or Variational Inference) on the entire posterior at once, which is computationally prohibitive to run at every single bandit round $t$ [2]. In contrast, diffusion models allow us to inject the likelihood drift gradually at each of the $L$ steps (Eq. 3.1), making the conditional sampling tractable.
>
> - Stability and Efficiency (vs. Score-only EBMs): While score-only Energy-Based Models (EBMs) also provide a score function, sampling from them requires running MCMC algorithms (e.g., Langevin dynamics) in the original, complex parameter space $\theta_0$ [3]. This standard MCMC process is empirically known to be unstable (slow mixing) on high-dimensional, multi-modal distributions [3, 4]. Furthermore, in the online setting, as the number of observations increases, the likelihood score magnitude can grow unbounded, causing standard MCMC-based posterior sampling to diverge. Diffusion models overcome this by using annealed sampling, which starts from simple noise and gradually refines the sample. This approach is proven to be far more stable and robust for exploring complex modes [4].
>
>
>
> ---
> ### **Q2:** How can the diffusion denoiser be trained or adapted online on the new task?
>
> **A2:** This is an excellent point. First, we emphasize that after offline training, our diffusion model can easily adapt to a new task through our conditional sampling framework. By injecting the task-specific interaction history as a likelihood-driven drift (Eq. 3.1) into the reverse process, the algorithm flexibly combines the fixed offline prior knowledge with online interaction data to guide exploration, enabling fast adaptation without needing to retrain the model parameters immediately.
>
> Regarding the online training of the denoiser weights themselves:
>
> 1. *Within a single task:* We clarify that adapting the diffusion prior using the current task's data is undesirable. This would cause the prior to "double-count" the evidence that is already being used in the likelihood gradient term, leading to posterior collapse and failed exploration.
>
> 2. *Across tasks (Meta-learning):* The more valuable adaptation, as the reviewer suggests, is in an online meta-learning setting across a stream of multiple, sequential tasks. A practical formulation for this would be to add the inferred posterior estimate $\hat{\theta}_{N+1}^*$ from the completed $(N+1)$-th task to our original training dataset, and periodically fine-tune the diffusion denoiser to continually refine the learned prior for subsequent tasks. Developing efficient methods to fine-tune the diffusion model on such new task data remains an open and interesting direction for future work.

---

> ### Author Response · Authors · 2025-11-26
>
> ---
> ### **Q3:** Quantitative analysis of multi-step projection in DPSG-MP
>
> **A3:** We thank the reviewer for this sharp observation. First, we would like to clarify the motivation: the multi-step projection variant is **not** introduced to avoid collapsing toward a MAP-like update, but rather because the standard DPSG (based on DPS) already behaves like an implicit, but unstable, MAP estimator. This design is motivated by the quantitative analysis in [5], which demonstrates that DPS does not function as a true posterior sampler. Specifically, [5] provide the following quantitative evidence:
> 1. Low Diversity (Collapse): They show that DPS samples exhibit significantly lower diversity (e.g., per-pixel standard deviation of $0.0453$ vs $0.3939$ for the ground truth), indicating that the method naturally collapses toward a deterministic mode rather than sampling from the full posterior.
> 2. High Bias: The conditional score estimated by DPS has a large error and a non-zero mean (violating the zero-mean property of valid scores), which acts as a drift term pushing the process toward a MAP solution.
>
> Therefore, to address this inherent behavior, we use multi-step projection (DPSG-MP) to explicitly and more effectively solve this constrained MAP optimization problem. This explicit optimization leads to significant stability compared to the implicit, one-step approximation in standard DPS. In our decision-making framework, where the accuracy of parameter estimation is critical for effective exploration, this improved stability allows DPSG-MP to significantly outperform the unstable DPSG baseline.
>
> To demonstrate this empirically, we additionally show an experiment comparing DPSG against DLTS and DPSG-MP on the sigmoid-gated nonlinear bandit tasks. The results are shown in the table below. We evaluate them in 64 tasks with 8 runs and demonstrate the mean and variance of the cumulative regret at $T=200$. It shows that DPSG is not ideal in performance.
>
> ### Regret
> | Prior Distribution | DPSG-MP | DLTS | DPSG |
> | :--- | :--- | :--- | :--- |
> | Cross | 34.8733 ± 0.4400 | 44.3341 ± 3.0742 | 262.1455 ± 1.0730 |
> | Rays | 36.3051 ± 1.0933 | 51.1978 ± 2.9018 | 326.5208 ± 0.9431 |
> | Corners | 38.1240 ± 1.2652 | 57.9246 ± 2.3493 | 660.2565 ± 1.5724 |
> | Triangles | 52.1988 ± 3.5373 | 47.2908 ± 1.7911 | 375.4471 ± 1.1921 |
> | Swirl | 38.3384 ± 1.6278 | 53.9466 ± 1.7429 | 387.2442 ± 1.9992 |
> | H | 52.4910 ± 3.4348 | 121.3069 ± 5.9771 | 493.5792 ± 2.3435 |
>
>
> Finally, we conduct an additional experiment to analyze the bias and variance of the parameter estimates, aiming to investigate how the quality of the estimation impacts the decision-making framework. We conduct the following experiments in the linear bandit with six problems. We evaluate the MSE of our proposed algorithms at $T=200$ between the estimated $\theta$ and the true prior in the following table. It illustrates that DLTS and DPSG-MP can have an exact estimation of the true prior.
>
>
>
>
> ### MSE for $\theta$ in linear setting
> | Problem | DLTS | DPSG | DPSG-MP |
> | :--- | :--- | :--- | :--- |
> | **Cross** | 0.1616 ± 0.1504 | 10140.0119 ± 103.4680 | 0.0212 ± 0.0238 |
> | **Rays** | 0.2285 ± 0.2577 | 10228.5693 ± 80.5599 | 0.0394 ± 0.0399 |
> | **Triangles** | 0.0995 ± 0.0743 | 10037.3289 ± 278.5414 | 0.0370 ± 0.0370 |
> | **Swirl** | 0.0599 ± 0.1125 | 10257.8273 ± 157.2136 | 0.0388 ± 0.0439 |
> | **H** | 0.1462 ± 0.1461 | 10062.1304 ± 376.1012 | 0.0554 ± 0.0719 |
> | **Corners** | 0.0835 ± 0.1335 | 10477.0995 ± 65.7582 | 0.0574 ± 0.0493 |
>
>
> **Results Analysis:** The empirical results clearly demonstrate that while the standard DPSG baseline fails to converge due to severe parameter estimation bias (MSE > 10,000) and obtains high regret, our proposed methods successfully stabilize the inference process. Both DPSG-MP and DLTS achieve precise parameter estimation (MSE < 0.3) which translates directly into superior decision-making performance. **Specifically, DPSG-MP exhibits exceptional stability with the lowest estimation error and variance, achieving the best regret performance in environments like 'Cross' and 'H'. DLTS also proves robust, yielding the lowest cumulative regret in the 'Triangles' setting.** These findings confirm that the multi-step projection and explicit conditional sampling strategies are essential for accurate and efficient exploration in nonlinear bandits.
>
> ---
> We hope we have addressed all of your questions. If you have any further questions, we would be happy to answer them and if you don’t, would you kindly consider increasing your score?

---

> ### Author Response · Authors · 2025-11-26
>
> ### References:
>
> [1] Papamakarios, George, et al. "Normalizing flows for probabilistic modeling and inference." Journal of Machine Learning Research 22.57 (2021): 1-64.
>
> [2] Bond-Taylor, Sam, et al. "Deep generative modelling: A comparative review of vaes, gans, normalizing flows, energy-based and autoregressive models." IEEE transactions on pattern analysis and machine intelligence 44.11 (2021): 7327-7347.
>
> [3] Song, Yang, and Stefano Ermon. "Generative modeling by estimating gradients of the data distribution." Advances in neural information processing systems 32 (2019).
>
> [4] Song, Yang, et al. "Score-based generative modeling through stochastic differential equations." arXiv preprint arXiv:2011.13456 (2020).
>
> [5] Xu, Tongda, et al. "Rethinking diffusion posterior sampling: From conditional score estimator to maximizing a posterior." arXiv preprint arXiv:2501.18913 (2025).

---

### Official Review · Reviewer_x8AA · 2025-11-03

**Soundness:** 3
**Presentation:** 3
**Contribution:** 3
**Rating:** 6
**Confidence:** 3

**Summary:**

This paper studies using posterior sampling to solve contextual bandits with general function approximation. Specifically, it assumes that the prior of the reward function parameter is represented by a diffusion model, which is in turn represented by a diffusion denoiser \epsilon_{\phi^*} (this can be obtained by e.g., domain knowledge or meta-learning), and designs efficient posterior sampling methods for the posterior distribution after some (action, reward) interaction history is seen.

It proposes two methods, Diffusion Langevin Thomspon Sampling (DLTS), and Diffusion Posterior Sampling with Guidance (DPSG) for this. DLTS's main idea is to use Langevin Monte Carlo to sample p(\theta_{l-1} | \theta_l, H_t). For DPSG, it is motivated by DPS (Chung et al, 2023), which approximates the likelihood-based score; the paper also proposes the DPSG-MP algorithm using multi-step projection as a more stable heuristic.

The paper gives some theory that when the quality of the initial diffusion denoiser \epsilon_{\phi^*} is imperfect (Assumption 4.3), and when the per step conditional reverse process p(\theta_{l-1} | \theta_l, H_t) is sampled exactly, then the proposed approximate Thompson sampling algorithm has Bayesian regret close to the exact Thompson sampling algorithm.

Experiments show that the proposed algorithms, DLTS and DPSG-MP, have good performance in linear and nonlinear bandits problems.

**Strengths:**

- Using diffusion models to model general prior distributions on reward parameters is a natural idea that can capture multimodality of the prior

- The proposed algorithm can handle general function approximation, which is in contrast with (Kveton et al, 2024) that can only handle linear and generalized linear models using special closed-form updates

- The experimental results are impressive

**Weaknesses:**

- The quality of approximation in DLTS (line 227), and DPSG (line 263) is not very clear to me. Can the authors comment on this?

- Continuing the last comment, Theorem 4.5 is proved in the ideal setting where p(\theta_{l-1} | \theta_l, H_t) is exact. How easy can it be extended to the setting where it is inexact?

- Can the author provide ablation study on DPSG vs DPSG-MP? Also, how would you recommend choosing K_l in practice?

- I think it would be nice to experiment on a bandit setting with reward function beyond the generalized linear form (line 422), so DiffTS is not runnable, to show the versatility of the proposed algorithm.

**Questions:**

- Is figure 3 showing the prior distribution of the MLP parameters? Then it is a high-dimensional distribution, so it is showing a projection of the samples from the distribution? How is that related to the distribution of \theta in the model r = f(x^T theta) + \epsilon in line 422?

---

> ### Author Response · Authors · 2025-11-26
>
> ## Response to Reviewer x8AA
>
> We thank the reviewer for your valuable time and effort in providing detailed feedback on our work. We hope our response will fully address all your questions.
>
> ---
> ### **Q1:** Explanation for approximation in DLTS and DPSG
>
> **A1:** Thank you for this question. Both approximations are necessary to make the posterior tractable for a general nonlinear reward model. The detailed derivations are provided in Appendix C of our paper.
>
> For DLTS, the challenge is to sample from the per-level posterior $p(\theta_ {\ell-1} | \theta_ {\ell}, \mathcal{H}_ {t})$, which requires computing the intractable likelihood integral $p(\mathcal{H}_ t|\theta_ {\ell-1})$. Our approximation, $\int_ {\theta_ 0} p\big(\mathcal{H}_ t|\theta_ 0\big) p\big(\theta_ 0 \big|\theta_ {\ell-1}\big) d \theta_ 0 \approx p\big(\mathcal{H}_ t|\theta_ {\ell-1} / \sqrt{\bar{\alpha}_ {\ell-1}}\big)$, is motivated by replacing the clean parameter $\theta_ 0$ with its expected value given the noisy state $\theta_ {\ell-1}$ (i.e., $\mathbb{E}[\theta_ 0 | \theta_ {\ell-1}] \approx \theta_ {\ell-1} / \sqrt{\bar{\alpha}_ {\ell-1}}$). Note that $\theta_ \ell = \sqrt{\bar{\alpha}_ \ell} \theta_0 + \sqrt{1-\bar{\alpha}_ \ell} \widetilde{\xi}_ \ell$, where $\widetilde{\xi}_ \ell \sim \mathcal{N}(0, I)$ is standard Gaussian noise. Rearranging gives $\theta_ 0 =(\theta_ \ell - \sqrt{1-\bar{\alpha}_ \ell} \widetilde{\xi}_ \ell) / \sqrt{\bar{\alpha}_ \ell}$, so $\theta_ 0$ can be viewed as a random variable with mean $\theta_ \ell / \sqrt{\bar{\alpha}_ \ell}$. Note that $\sqrt{(1-\bar{\alpha}_ \ell) / \bar{\alpha}_ \ell} \rightarrow 0$ as $\ell \rightarrow 1$. Therefore, this approximation becomes more precise in later stages of the reverse process.
>
> For DPSG, the method is inspired by the standard Diffusion Posterior Sampling (DPS) framework and aims to compute the conditional score $\nabla_\theta \log p_ s(\theta_ s | \mathcal{H}_ t)$. This score is decomposed into an unconditional (prior) term and an intractable likelihood score $\nabla_ \theta \log p_ s(\mathcal{H}_ t | \theta_ s)$. We approximate this likelihood score by first using Tweedie's formula to get a tractable estimate of the clean parameter, $\hat{\theta}_ 0(\theta_ s, s)$, and then using the gradient of the loss $L_ t$ evaluated at this estimate $(-\nabla_\theta L_t(\hat{\theta}_0(\theta_s, s)))$ as a proxy for the true score. DPSG uses this Tweedie estimate (unlike the simpler rescaling in DLTS) because it incorporates the learned score function to provide a denoised estimate of $\theta_0$, which yields more accurate likelihood gradients for guidance, which pulls the unconditional sample from the diffusion prior toward regions of high likelihood fitting the current task's history $\mathcal{H}_t$.
>
>
> ---
> ### **Q2:** Discussion on extension of Thm 4.5 to inexact setting
>
> **A2:** Theorem 4.5 bounds the gap from the learned prior (OTS vs. ideal ODTS), which depends on the score-matching error $\epsilon_{\text{score}}$. Extending this analysis to our practical algorithms, DLTS and DPSG, is a significant theoretical challenge precisely because they rely on inexact (approximate) posterior updates. A full analysis would require introducing a new, distinct error term, denoted as $\Delta^{\text{Approx}}$, to capture the gap between the ideal ODTS and our practical implementation.
>
> This $\Delta^{\text {Approx}}$ term would need to quantify the specific approximation error of the posterior at each reverse step:
> - For DLTS (Algorithm 1), this error would arise from the use of finite-step Langevin Monte Carlo (LMC) to sample from the per-level posterior $p(\theta_{\ell-1} | \theta_\ell, \mathcal{H}_t)$.
> - For DPSG (Algorithm 2), this error would arise from the Tweedie's formula approximation used to estimate the intractable likelihood score $\nabla_\theta \log p_s(\mathcal{H}_t | \theta_s)$.
>
> The main theoretical challenge is in formally bounding how this per-step approximation error $\Delta^{\text {Approx}}$ accumulates over the $L$ steps of the reverse-diffusion chain and interacts with the prior score-matching error $\epsilon_{\text {score}}$.
>
> We clarify that the primary scope of this work is algorithm design, establishing a framework that fundamentally integrates posterior sampling into the iterative diffusion reverse process rather than treating it as a simple plug-in prior. We note that deriving precise regret bounds for diffusion-based bandits remains an open and non-trivial challenge even in linear settings, and thus falls outside the scope of the current paper.

---

> ### Author Response · Authors · 2025-11-26
>
> ---
> ### **Q3:** Ablation study and choice of $K_{\ell}$
>
> **A3:** We specify that the hyperparameter $K_{\ell}$ (inner-loop steps) is selected via grid search from the set $\{1,10\}$ in our main paper. While a larger $K_{\ell}$ could theoretically allow for more precise posterior sampling (in DLTS) or projection (in DPSG-MP), it linearly increases the inference cost. In practice, the optimal choice is also affected by the learning rate $\eta$. To empirically demonstrate this, we conducted new ablation studies on the challenging sigmoid-gated nonlinear bandit setting from Section 5.2, analyzing the sensitivity to $K_{\ell}$ (varying over 1, 2, 5, 10) with the diffusion steps fixed at $L=100$. Results are reported as cumulative regret at $T=200$, averaged over 64 tasks across 8 runs. We present the results in the Rays, Triangles and Swirl. **Our analysis suggests that a small $K_{\ell}$ (e.g., 1 or 2) often provides the best trade-off between performance and computational cost.** We include these detailed ablation studies (Appendix I) in our revision.
>
>
> ### Rays
>
> | Inner-Loop Steps | DLTS | DPSG-MP |
> | :--- | :--- | :--- |
> | 1 | 51.1978 ± 2.9018 | 39.5867 ± 0.8039 |
> | 2 | 43.7158 ± 1.5759 | 39.4732 ± 0.8546 |
> | 5 | 63.2363 ± 3.4842 | 37.3604 ± 0.3303 |
> | 10 | 55.8693 ± 3.0374 | 36.3051 ± 1.0933 |
>
>
> ### Triangles
>
> | Inner-Loop Steps | DLTS | DPSG-MP |
> | :--- | :--- | :--- |
> | 1 | 79.5997 ± 2.4942 | 54.9418 ± 6.3533 |
> | 2 | 52.6539 ± 5.3189 | 51.2819 ± 2.4380 |
> | 5 | 160.9929 ± 25.4214 | 49.5074 ± 1.9641 |
> | 10 | 47.2908 ± 1.7911 | 52.1988 ± 3.5373 |
>
> ### Swirl
>
> | Inner-Loop Steps | DLTS | DPSG-MP |
> | :--- | :--- | :--- |
> | 1 | 90.1970 ± 3.6337 | 34.7951 ± 1.4286 |
> | 2 | 78.8142 ± 0.7685 | 36.4597 ± 0.8702 |
> | 5 | 66.4889 ± 2.3441 | 34.8075 ± 0.9337 |
> | 10 | 53.9466 ± 1.7429 | 38.3384 ± 1.6278 |
>
>
>
>
>
> ---
> ### **Q4:** Additional experiments of reward function beyond the generalized linear form
>
> **A4:** We thank the reviewer for this excellent suggestion. It allows us to demonstrate the primary advantage of our framework: its ability to handle general non-linear reward functions where specialized baselines like DiffTS are inapplicable.
>
> **Limitations of Prior Work:** As the reviewer notes in *Strengths*, state-of-the-art diffusion bandit methods [1,2] are restricted to linear or generalized linear models (GLM) due to their reliance on closed-form, conjugate-like posterior updates.
> - DiffTS (Kveton et al., 2024) experiments exclusively with linear and logistic reward models.
> - dTS (Aouali et al., 2025) similarly evaluates only on linear and logistic (Bernoulli) rewards.
>
> To demonstrate the versatility of our gradient-based sampling framework (DLTS/DPSG), we propose experiments on the following distinct reward model that lie strictly outside the GLM family:
>
>
> **Generalized Additive Models (GAMs)** model the reward as a sum of nonlinear functions of individual features: $r(x)=w_0+f_1(x_1)+f_2(x_2)+\cdots+f_d(x_d)+\epsilon$, where we parameterize each $f_j$ using nonlinear basis function, where $\theta$ consists of the function coefficients. This provides a flexible, interpretable non-linear model that baselines like DiffTS cannot capture. Specifically, we use the following nonlinear reward: $r(x) = \theta_1 \cdot \sin (\pi x_1) + \theta_2 \cdot \cos (\pi x_2)+\epsilon$. We follow the same setting in nonlinear bandit to perform the experiments. We summarize the performance with 64 tasks in 8 runs as follows.
>
> | Dataset | DiffTS | NeuralTS | NeuralUCB | DLTS | DPSG-MP |
> | :--- | :--- | :--- | :--- | :--- | :--- |
> | **Cross** | 71.70 ± 2.04 | 15.58 ± 3.18 | 28.66 ± 5.24 | 49.64 ± 1.29 | 21.82 ± 0.13 |
> | **Rays** | 259.85 ± 14.00 | 79.04 ± 3.18 | 95.94 ± 14.66 | 86.75 ± 5.07 | 103.05 ± 0.49 |
> | **Triangles** | 90.87 ± 10.01 | 32.04 ± 12.59 | 47.91 ± 9.98 | 26.07 ± 0.32 | 29.62 ± 0.21 |
> | **Swirl** | 80.93 ± 4.80 | 18.81 ± 3.10 | 44.81 ± 9.69 | 78.14 ± 1.49 | 27.10 ± 0.29 |
> | **H** | 180.55 ± 1.52 | 42.49 ± 4.50 | 111.85 ± 10.14 | 59.39 ± 3.28 | 37.60 ± 0.28 |
> | **Corners** | 104.22 ± 6.01 | 39.96 ± 2.42 | 30.90 ± 4.34 | 245.65 ± 12.16 | 34.87 ± 0.39 |
>
>
> **Results Analysis:** As expected, the linear baseline DiffTS fails to capture the nonlinear reward structure, resulting in high regret across all datasets. In contrast, our methods successfully leverage the diffusion prior to solve these Generalized Additive Models. Specifically, **DPSG-MP** achieves the lowest cumulative regret in the **'H'** environment, and **DLTS** yields the best performance in **'Triangles'**. In other challenging settings like **'Cross', 'Swirl', and 'Corners'**, while DPSG-MP delivers competitive regret comparable to strong neural baselines (NeuralTS/NeuralUCB), it demonstrates significantly **higher stability with drastically lower variance** (e.g., standard error $\pm 0.13$ vs $\pm 3.18$ in 'Cross'). This suggests that our diffusion guided approach provides more robust exploration compared with other baselines.

---

> ### Author Response · Authors · 2025-11-26
>
> ---
> ### **Q5:** Is figure 3 showing the prior distribution of the MLP parameters? Is it showing a projection of the samples from the distribution? How is that related to the distribution of $\theta$?
>
> **A5:** Yes, Figure 3 shows the prior distribution of the MLP parameters. We use UMAP to project the high-dimensional parameters into two dimensions. We constructed the MLP prior by first sampling the $\theta$ from the problem and using the reward function to construct the $(x,y)$ pairs in one task. We then train an MLP to approximate the true $y$ and use the MLP parameter as one data point. We collected $N$ tasks as the prior training datapoint.
>
>
> ---
> We hope we have addressed all of your questions. If you have any further questions, we would be happy to answer them and if you don’t, would you kindly consider increasing your score?
>
>
> ### References:
>
> [1] Kveton, Branislav, et al. "Online posterior sampling with a diffusion prior." Advances in Neural Information Processing Systems 37 (2024): 130463-130484.
>
> [2] Aouali, Imad. "Diffusion models meet contextual bandits." Advances in Neural Information Processing Systems (2025).

---

### Author Response · Authors · 2025-11-26
**General response comment about additional experiments**

## General response comment:
We would like to express our sincere gratitude to all reviewers for their feedback and their overall appreciation of our work. Based on the reviewers' valuable feedback, we have conducted the following additional experiments to strengthen our empirical validation and demonstrate the broad applicability of our framework:


1. **Real-World MovieLens Recommendation:** We added a realistic recommendation experiment using the MovieLens-25M dataset to identify user preferences. This validates our approach on real-world data structures, where we trained an MLP-based diffusion prior on offline user-item interactions.
2. **Hyperparameter Sensitivity ($L$ and $K_{\ell}$):** We performed comprehensive ablation studies on the sigmoid-gated nonlinear bandit to analyze the impact of the number of diffusion steps $L$ and inner-loop updates $K_{\ell}$. These results quantify the trade-offs between computational complexity and regret performance.
3. **General Nonlinear Reward Models:** To demonstrate versatility beyond Generalized Linear Models (GLMs), we evaluated our algorithms on strictly non-linear reward functions: Generalized Additive Models (GAMs). The experiments confirm that our framework effectively handles complex reward landscapes where baselines like DiffTS are inapplicable.
4. **Validation of DPSG-MP Design:** We conducted comparative experiments and a bias-variance analysis to empirically validate the motivation for DPSG-MP. The results confirm that the multi-step projection strategy effectively mitigates high bias issues observed in the standard DPSG formulation.
5. **Comparison with In-Context RL:** We benchmarked our method against the Decision Pretrained Transformer (DPT). This comparison positions our diffusion-based approach against state-of-the-art in-context reinforcement learning baselines.

6. **Wall-Clock Time Analysis:** We conducted a time-to-accuracy analysis to better capture the practical trade-off between computational cost and sample efficiency. We define a target performance threshold based on average regret (cumulative regret divided by the number of rounds). We then measure the total wall-clock time each algorithm requires to achieve this target average regret level. This metric effectively normalizes for sample efficiency: a slower per-step algorithm like ours may still be more time-efficient overall if it requires significantly fewer interactions to reach a high-quality policy compared to faster but less sample-efficient baselines.

7. **Long-Horizon Evaluation:** We extended the horizon of our challenging sigmoid-gated experiment to $T=2000$ rounds. This confirms that the exploration advantages provided by our learned diffusion prior are robust and sustainable over long-term interactions.

8. **DPSG vs. DPSG-MP Stability:** We explicitly compared the performance of plain DPSG against DPSG-MP on sigmoid-gated tasks. The results highlight the instability of the single-step correction in standard DPSG and justify the necessity of the multi-step projection used in DPSG-MP.

9. **Sensitivity to Prior Data Quantity:** We performed an ablation study by training the diffusion prior on varying subsets of the offline dataset ($10 \%$ to $100 \%$). This analysis quantifies the robustness of our algorithms to the amount of data available for pre-training the prior.

We believe these additional results further substantiate the advantages of our proposed methods and provide deeper insights into their properties. We sincerely thank all reviewers for their constructive feedback, which has significantly improved the quality of our paper.

---

### Meta-Review · Area_Chair_Nxmh · 2026-01-05

**Summary:**

The paper tackles an interesting and timely topic: using diffusion models as learned priors for posterior sampling in nonlinear contextual bandits. The framework is technically solid and experiments are strong, with extra results added in the rebuttal like ablations and a MovieLens test. Reviewers appreciate the ambition of going beyond linear or GLM settings. But concerns remain about novelty and theory. The theoretical results are mostly sanity checks or loose bounds under ideal assumptions, and they don’t really describe the practical algorithms, which rely on heavy approximations. Regret bounds look weak and maybe pessimistic, so the gap between theory and practice is big. Some reviewers also question if diffusion is essential here or just a fancy prior, since other generative models might work too. Computational cost and online practicality are still concerns even after extra experiments. Overall, the paper feels solid but incremental compared to recent work, and doesn’t quite meet the ICLR bar.

**Reviewer Concerns:**

The rebuttal helped with clarifications and added lots of experiments, including ablations and comparisons with in-context RL methods. It also explained the role of diffusion and design choices like DPSG-MP. But the main issues remain: no strong theoretical guarantees for the practical algorithms, regret bounds are loose and not very informative, novelty beyond recent diffusion bandit papers is unclear, and doubts about whether diffusion is essential persist. Efficiency and scalability concerns are still only partly addressed.

**Reviewer Scores:**

Reviewers who were slightly positive might keep their scores or raise them a little after seeing the extra work. Those who were negative because of novelty and theory will likely stay the same. Overall, the scores remain mixed and probably below the acceptance threshold.

---

### Decision · Program_Chairs · 2026-01-26

Reject